# A Policy-Gradient Approach to Solving Imperfect-Information Games with Best-Iterate Convergence

**Mingyang Liu, Gabriele Farina & Asuman Ozdaglar**
LIDS, EECS
Massachusetts Institute of Technology
Cambridge, MA 02139, USA
{liumy19,gfarina,asuman}@mit.edu

## Abstract

Policy gradient methods have become a staple of any single-agent reinforcement learning toolbox, due to their combination of desirable properties: iterate convergence, efficient use of stochastic trajectory feedback, and theoretically-sound avoidance of importance sampling corrections. In multi-agent imperfect-information settings (extensive-form games), however, it is still unknown whether the same desiderata can be guaranteed while retaining theoretical guarantees. Instead, sound methods for extensive-form games rely on approximating *counter-factual* values (as opposed to Q values), which are incompatible with policy gradient methodologies. In this paper, we investigate whether policy gradient can be safely used in two-player zero-sum imperfect-information extensive-form games (EFGs). We establish positive results, showing for the first time that a policy gradient method leads to provable best-iterate convergence to a regularized Nash equilibrium in self-play.

## 1 Introduction

In recent years, deep reinforcement learning (DRL) has succeeded tremendously in many applications with large and complex environments, such as games (Mnih et al., 2013; Silver et al., 2017), autonomous driving (Kiran et al., 2021), robotics (Ibarz et al., 2021), and large language models (*e.g.* Ouyang et al. (2022) and ChatGPT). Much of these successes are due to the applicability of scalable algorithms, such as proximal policy optimization (PPO) (Schulman et al., 2017) and soft actor-critic (SAC) (Haarnoja et al., 2018). The success of these algorithms hinges on a few critical properties—these algorithms (**I**) only require value estimates obtained from repeated random rollouts which can be implemented efficiently; (**II**) converge in iterates (as opposed to in averages), removing the need for either training an average policy approximator or storing snapshots of past policies; and (**III**) soundly avoid importance sampling corrections, which can be detrimental in practice as they often lead to outsized reward estimates.

However, DRL is not applicable in multi-agent imperfect-information settings, such as Texas Hold'em poker, where they tend to end up trapped in cycles without making progress (Balduzzi et al., 2019). Constructing policy gradient algorithms that enjoy the same wide applicability as their DRL counterparts and yet retain theoretical guarantees in tabular settings even in imperfect-information games is a challenging, open direction of research. Current sound algorithms for competitive games typically see their scalability limited by two major obstacles: their lack of last- (or even best-) iterate convergence, and their reliance on counterfactual values. In what follows we illustrate both of these issues separately.

**Average vs iterate convergence.** In the last decade, most scalable techniques to solve Nash equilibrium strategies in two-player zero-sum imperfect-information extensive-form games (EFGs) have been based on Counterfactual Regret Minimization (CFR) (Zinkevich et al., 2007) and its modern variants (Tammelin et al., 2015; Farina et al., 2019a; Brown and Sandholm, 2019; Farina et al., 2021a). These algorithms guarantee that their *average* strategy converges to the set of Nash equi-

librium (NE) strategies. However, average-iterate convergence is not desirable within the regime of deep learning, where strategies are stored indirectly as a vector of neural network weights. To represent and use the average strategy, some authors have resorted to storing in memory multiple snapshots of the network (Steinberger, 2019; Steinberger et al., 2020), sampling one at random; this can quickly become expensive in large games (Liu et al., 2023)). Alternatively, some authors have included—as part of their pipeline—training a second network whose goal is approximating the average strategy (Brown et al., 2019). This approach is also undesirable, as it incurs an additional error in the approximation of the average strategy.

In light of the above discussion, a recent trend in the literature has focused on algorithms that do not require taking averages of strategies and instead *converge* to the set of Nash equilibrium strategies— in short, exhibit *iterate convergence*. A distinction between two forms of iterate convergence is often made: *best-iterate* convergence, meaning that at least one of the iterates produced by the algorithm is very close to equilibrium, and *last-iterate* convergence, meaning that the last iterate produced is. While this line of work has produced a wealth of algorithms with provable last-iterate convergence Daskalakis and Panageas (2019); Anagnostides et al. (2022); Wei et al. (2021); Lee et al. (2021); Cen et al. (2021); Liu et al. (2023), a major obstacle towards practical combination with function approximation has been the reliance of these algorithms on *counterfactual values*, which we discuss next.

**Counterfactual values vs Q-values.** Aside from the cycling effect, there is another reason that reinforcement learning, commonly based on Q-values in single-agent environments, is not applicable in EFGs, which is the fundamental *state* concept being replaced by the *information set*. Players' expected payoffs of taking action $a$ at an information set $s$ need to consider the opponent's reach probabilities to $s$, captured in the notion of *counterfactual values*. These are essentially Q-values multiplied by the probability of the opponent and environment reaching $s$ from the game's root when the opponent follows their strategy.

The downsides of counterfactual values become apparent when turning the attention onto *estimation* of values. Q-values are extremely practical to estimate by performing random rollouts. For counterfactual values, several techniques have been proposed. A popular technique in this space is using importance sampling to estimate the counterfactual values (Lanctot et al., 2009; Farina et al., 2020). However, importance sampling is not suitable for DRL in large-scale environments due to the high dispersion (*i.e.*, range of values) of the produced estimator—for instance, the estimated Q-value can be as large as the game size (Kozuno et al., 2021; Bai et al., 2022; Fiegel et al., 2023), thus hindering the stable training of neural networks. Certain variance reduction techniques, such as those by Schmid et al. (2019), still suffer a large dispersion even though the variance is reduced.

**External Sampling vs Trajectory Rollouts.** Algorithms that sidestep the need for importance sampling, such as External Monte Carlo CFR (Lanctot et al., 2009), do that at the expense of exploring about square-root of the EFG size in every iteration to reduce the variance of the estimator. This is in stark contrast with DRL, which simply samples a batch of trajectories at every iteration, each with a size proportional to the height of the tree, which is typically logarithmic in the size of the game. In large games such as Stratego, where the size of the game tree is approximately $10^{535}$ (Perolat et al., 2022), external sampling needs to visit more than $10^{200}$ infosets at each iteration while trajectory rollouts only visits no more than $4 \cdot 10^3$ infosets per iteration.

**Contributions.** Given the above discussion, a question is natural:

> *Is it possible to design a theoretically sound policy gradient method for solving two-player zero-sum extensive-form games that achieves the desiderata* (**I**), (**II**), (**III**) *listed in the introduction?*

Such an algorithm would enable estimating values via rollouts, without need of importance sampling, all while ensuring iterate convergence, bringing EFG technology more in line with modern DRL technology. Our aim in this paper is to show that a positive answer to the question is possible.

In this work, we develop the first principled policy gradient approach for solving imperfect information EFGs. Our approach builds on a particular notion of Q-values for EFGs, called *trajectory Q-values*, which admits efficient estimation through random rollouts without importance sampling. Our algorithm also introduces a new regularizer for EFGs, which we coin *bidilated* regularizer. When paired with trajectory Q-values, the bidilated regularizer enables iterate convergence with both full-information and stochastic feedback obtained through the sampled trajectory. To obtain

the results, we devised a novel learning rate schedule that increases with the depth of the game tree. It ensures that the strategies of ancestors are changing slower than those of the children, and are therefore stable when the children are updating their payoff estimates.

The discussion about related work is postponed to Appendix A, and summarized in Table 1.

| Algorithm | Iterate convergence | Q-values | Stochastic feedback |
|---|---|---|---|
| DREAM, DEEP-CFR (Steinberger et al., 2020; Brown et al., 2019) | ✗ | ✗ | ✓ |
| OOMD, REG-DOMD, MMD (Lee et al., 2021; Liu et al., 2023; Sokota et al., 2023) | ✓(last) | ✗ | ✗ |
| ADAPTIVE FTRL (Fiegel et al., 2023) | ✗ | ✗ | ✓ |
| ESCHER, LOCALOMD (McAleer et al., 2023; Fiegel et al., 2024) | ✗ | ✓ | ≈ (off-policy) |
| ARMAC, ACH (Gruslys et al., 2020; Fu et al., 2021) | ✗ | ✓ | ≈ (infinite samples) |
| QFR (this paper) | ✓(best) | ✓ | ✓ |

Table 1: The table above compares the related work in three aspects: convergence guarantee, feedback type, and supporting sampling or not. (last) and (best) denote last-iterate convergence and best-iterate convergence, respectively.

## 2 PRELIMINARIES

For any vector $\boldsymbol{x}$, we use $x_i$ as element $i$ of vector $\boldsymbol{x}$ and $\|\boldsymbol{x}\|_p$ as the $p$-norm. We let $\|\boldsymbol{x}\|$ denote the Euclidean norm $\|\boldsymbol{x}\|_2$. We use $\Delta^n$ to denote the $(n-1)$-dimensional probability simplex $\{\boldsymbol{x} \in [0,1]^n \colon \sum_{i=1}^n x_i = 1\}$. We also define the Bregman divergence $D_\psi(\boldsymbol{x}, \boldsymbol{y}) \coloneqq \psi(\boldsymbol{x}) - \psi(\boldsymbol{y}) - \langle \nabla \psi(\boldsymbol{y}), \boldsymbol{x} - \boldsymbol{y} \rangle$ with respect to the $c$-strongly convex function $\psi$. The $c$-strong convexity of $\psi$ implies the bound $D_\psi(\boldsymbol{x}, \boldsymbol{y}) \geq \frac{c}{2} \|\boldsymbol{x} - \boldsymbol{y}\|^2$. For any integer $n \geq 0$, we use $[n] \coloneqq \{1, 2, \cdots, n-1, n\}$. For any set $\mathcal{S}$, we denote with $|\mathcal{S}|$ as its cardinality.

**Extensive-Form Games.** EFGs are played on a rooted game tree. In this paper we focus on two-player zero-sum EFGs; hence, each node (also known as *history*) belongs to exactly one player out of the set $\{1, 2\} \cup \{c\}$. The special player $c$ is called the *chance player*, and models stochastic events (for example: a roll of the dice or dealing a card from a shuffled deck) sampled from a known distribution. We use $\mathcal{H}_1, \mathcal{H}_2, \mathcal{H}_c$ to denote the set of nodes belonging to each of the players. Terminal nodes (nodes without children) have an associated payoff for each player, *i.e.* $\mathcal{U}_1(h), \mathcal{U}_2(h)$ for player $1, 2$ individually and $\mathcal{U}_1(h) = -\mathcal{U}_2(h)$ for any terminal node $h$ since the game is zero-sum.

To model imperfect information, the set of nodes $\mathcal{H}_i$ of each player $i \in [2]$ is partitioned into *information sets* (or *infosets* for short) $s_1, s_2, \cdots, s_m$. Nodes in the same infoset are indistinguishable for the acting player of that infoset. For example, in poker player 1 cannot distinguish two nodes in the game tree that only differ on the private cards of player 2, since player 1 does not observe the hand of the opponent. We use $\mathcal{S}_i \coloneqq \{s_1, s_2, \cdots, s_m\}$ to denote the collection of all infosets of player $i$. Let $\mathcal{H} \coloneqq \mathcal{H}_1 \cup \mathcal{H}_2$ and $\mathcal{S} \coloneqq \mathcal{S}_1 \cup \mathcal{S}_2$ be the joint set of nodes and infosets of player $1, 2$ for convenience. Because nodes in the same infoset are indistinguishable from the acting player, they must all have the same action set, which we denote with $\mathcal{A}_s$ as the action set of infoset $s \in \mathcal{S}$. Furthermore, $p \colon \mathcal{S} \to \{1, 2\}$ denotes the player that an infoset $s$ belongs to.

We make the assumption that each player remembers all their past observations and actions; this assumption is standard and goes under the name of *perfect recall*. A direct corollary of this assumption is that nodes in the same infoset $s \in \mathcal{S}_i$ have the same past observation along the path from the root to the node in the view of player $i$. For any two nodes $h, h' \in \mathcal{H}$, we write $h \sqsubseteq h'$ if $h$ is on the path from the root of the game tree to $h'$. Suppose $h \in s$ and for any $a \in \mathcal{A}_s$, we write $(h, a) \sqsubseteq h'$ if the path from the root of the game tree to node $h'$ includes the edge corresponding to taking action $a$ at $s$. For any two infosets $s, s' \in \mathcal{S}_i$ that belong to the same player $i \in [2]$, whenever there exist two nodes $h \in s, h' \in s'$ such that $h \sqsubseteq h'$, we write $s \sqsubseteq s'$. Similarly, we write $(s, a) \sqsubseteq s'$ for any $a \in \mathcal{A}_s$ when there exist nodes $h \in s, h' \in s'$ such that $(h, a) \sqsubseteq h'$. Moreover, we can define $(s, a) \sqsubseteq (s', a')$ for $s, s' \in \mathcal{S}_i$ for some $i \in [2]$ and $a \in \mathcal{A}_s, a' \in \mathcal{A}_{s'}$. Furthermore, for any player $i \in [2]$, we define the *parent sequence* $\sigma(s)$ of an infoset $s \in \mathcal{S}_i$ as the last infoset-action pair $(s', a')$, where $s' \in \mathcal{S}_i, a' \in \mathcal{A}_{s'}$, encountered along the path from root to any nodes in $s$ (the choice of node

in $s$ is irrelevant and $\sigma(s)$ is either unique or non-existing due to the perfect-recall assumption ). If there does not exist such infoset-action pair, we let $\sigma(s) = \emptyset$. For any node $h \in \mathcal{H}$, we define $\sigma_i(h)$ as the last infoset-action pair $(s, a)$, where $s \in \mathcal{S}_i$, encountered along the path from root to $h$. Finally, we define the *depth* $\mathcal{D}(h)$ of a node $h \in \mathcal{H}$ as the number of actions (of all players) on the path from the root of the game tree to $h$. The depth $\mathcal{D}(s)$ of an infoset $s \in \mathcal{S}_i$ is the maximum depth of any node $h \in s$. Furthermore, $\mathcal{D} := \max_{h \in \mathcal{H}} \mathcal{D}(h)$ is the depth of the game.

**Strategies in EFGs.** Since players cannot differentiate nodes in the same infoset, their strategies must be the same at all of them. For player $i$ and infoset $s \in \mathcal{S}_i$ (*i.e.* $p(s) = i$), we use $\pi_i(a \,|\, s)$ to denote the probability of taking action $a \in \mathcal{A}_s$ at any node in infoset $s$. We use $\boldsymbol{\pi} := (\pi_1, \pi_2)$ to denote the strategy profile. For a player $i$, given an assignment of $\pi_i(a \,|\, s)$ for each $s \in \mathcal{S}_i, a \in \mathcal{A}_s$, then we can represent strategies for the EFG via their *sequence-form* representation (Von Stengel, 1996). This is a mapping $\mu_i^{\pi_i} \colon \bigcup_{s \in \mathcal{S}_i, a \in \mathcal{A}_s} \{(s, a)\} \to [0, 1]$ associated with strategy $\pi_i$ for each player $i \in [2]$, where $\mu_i^{\pi_i}(s, a) := \mu_i^{\pi_i}(\sigma(s)) \cdot \pi_i(a \,|\, s)$ for any $s \in \mathcal{S}_i, a \in \mathcal{A}_s$. Note that $\mu_i^{\pi_i}(\emptyset) = 1$. For each $h \in \mathcal{H}$, according to the definition above, $\mu_i^{\pi_i}(\sigma_1(h))$ is equal to the product of the probability of all of Player $i$'s actions from the root of the tree down to node $h$. We use $\mu_c(h)$ to denote the probability of reaching $h \in \mathcal{H}$ contributed by the chance player. We assume $\mu_c(h) > 0$ for any $h \in \mathcal{H}$, since otherwise $h$ will never be reached and thus can be removed from the game tree.

For simplicity, let $\mu_1^{\pi_1}$ to be a vector with index $(s, a)$, where $s \in \mathcal{S}_i, a \in \mathcal{A}_s$, and $(\mu_1^{\pi_1})_{(s,a)} = \mu_1^{\pi_1}(s, a)$. In this representation, the expected utility for Player 1 is the bilinear function $(\mu_1^{\pi_1})^\top \boldsymbol{A} \mu_2^{\pi_2}$ [1] (the utility for Player 2 is $-(\mu_1^{\pi_1})^\top \boldsymbol{A} \mu_2^{\pi_2}$ since the game is zero-sum). We define the convex polytope of all valid sequence-form strategies as $\Pi_1, \Pi_2$ for player $1, 2$ respectively, and $\Pi := \Pi_1 \times \Pi_2$. For simplicity, let $\boldsymbol{\mu^\pi} := (\mu_1^{\pi_1}, \mu_2^{\pi_2})$ as the concatenation of the sequence-form strategy of both players. Sometimes, we will omit the subscript of $\mu$ when it is clear from the context, such as writing $\mu_{p(s)}^{\pi_{p(s)}}(s, \cdot)$ as $\mu^{\boldsymbol{\pi}}(s, \cdot)$.

**Counterfactual and Q-Values.** In this section we recall some key notions of values for EFGs. Our exposition is mostly intuitive to avoid notational burden; all definitions can be found in Appendix B. We start from introducing these values for *nodes* and for Player 1 (the definitions are symmetric for Player 2), and will later extend the definition to infosets.

- The *Q-value* $Q_1^{\boldsymbol{\pi}}(h, a)$ associated with strategies $\pi_1, \pi_2$ for node-action pair $(h, a)$ is defined as Player 1's expected utility in the subtree rooted at $h$, when Player 1 follows $\pi_1$ and Player 2 follows $\pi_2$ after first selecting $a$ as their first action.
- The *counterfactual value* $\mathsf{CF}_1^{\boldsymbol{\pi}}(h, a)$ is defined as the product of the corresponding Q-value with the probability that Player 2 and the chance player reach $h$; in symbols,

$$\mathsf{CF}_1^{\boldsymbol{\pi}}(h, a) := \mu_2^{\pi_2}(\sigma_2(h)) \cdot \mu_c(h) \cdot Q_1^{\boldsymbol{\pi}}(h, a).$$

- The *trajectory Q-value* $\overline{Q}_1^{\boldsymbol{\pi}}(h, a)$ is defined as the product of the corresponding Q-value with the probability of the path of actions from the root of the game tree down to $h$. In symbols,

$$\overline{Q}_1^{\boldsymbol{\pi}}(h, a) := \mu_1^{\pi_1}(\sigma_1(h)) \cdot \mu_2^{\pi_2}(\sigma_2(h)) \cdot \mu_c(h) \cdot Q_1^{\boldsymbol{\pi}}(h, a).$$

We now extend these definitions from nodes to *infosets*. Consider infoset $s \in \mathcal{S}$ and action $a \in \mathcal{A}_s$. For trajectory Q-value and counterfactual value we simply have $\overline{Q}_1^{\boldsymbol{\pi}}(s, a) := \sum_{h \in s} \overline{Q}_1^{\boldsymbol{\pi}}(h, a)$ and $\mathsf{CF}_1^{\boldsymbol{\pi}}(s, a) := \sum_{h \in s} \mathsf{CF}_1^{\boldsymbol{\pi}}(h, a)$. For Q-value, $Q_1^{\boldsymbol{\pi}}(s, a) := \mathbb{E}_{h \sim d(\cdot \,|\, s)}[Q_1^{\boldsymbol{\pi}}(h, a)]$, where $d(h \,|\, s) \propto \mu_c(h)\mu_1^{\pi_1}(\sigma_1(h))\mu_2^{\pi_2}(\sigma_2(h)) \propto \mu_c(h)\mu_2^{\pi_2}(\sigma_2(h))$ [2] for any $h \in s$. All definitions are symmetric for Player 2.

**Regret and Equilibrium.** When players 1 and 2 play according to strategies $\mu_1^{\pi_1}, \mu_2^{\pi_2}$ respectively, the utility of Player 1 is $(\mu_1^{\pi_1})^\top \boldsymbol{A} \mu_2^{\pi_2}$ and that of Player 2 is $-(\mu_1^{\pi_1})^\top \boldsymbol{A} \mu_2^{\pi_2}$ since the game is zero-sum. An $\epsilon$-approximate Nash equilibrium (NE) of the game is then defined as follows.

**Definition 2.1** ($\epsilon$-approximate Nash Equilibrium)**.** For compact convex set $\mathcal{C}_1, \mathcal{C}_2$, a strategy profile $(\mu_1, \mu_2) \in \mathcal{C}_1 \times \mathcal{C}_2$ is an $\epsilon$-approximate NE if

$$\max_{\widehat{\mu}_1 \in \mathcal{C}_1} \widehat{\mu}_1^\top \boldsymbol{A} \mu_2 - \min_{\widehat{\mu}_2 \in \mathcal{C}_2} \mu_1^\top \boldsymbol{A} \widehat{\mu}_2 \le \epsilon. \tag{2.1}$$

---

[1] $\boldsymbol{A} \in [-1, 1]^{\sum_{s \in \mathcal{S}_1} |\mathcal{A}_s| \times \sum_{s' \in \mathcal{S}_2} |\mathcal{A}_{s'}|}$ is the utility matrix of the game.

[2] Because $\sigma_1(h) = \sigma(s)$ for any $s \in \mathcal{S}_1$ and $h \in s$ due to the perfect-recall assumption.

When $\epsilon = 0$, we also simply call the strategy profile a Nash equilibrium (NE). Equation (2.1) is also called the *exploitability* of the strategy profile $(\mu_1, \mu_2)$.

Given a sequence of $T$ strategy pairs $(\mu_1^{(t)}, \mu_2^{(t)}) \in \mathcal{C}_1 \times \mathcal{C}_2$, we can define the regret of Player 1 as (Player 2's is analogous),

$$R_1^{(T)} := \max_{\widehat{\mu}_1 \in \mathcal{C}_1} \sum_{t=1}^{T} \left( \widehat{\mu}_1 - \mu_1^{(t)} \right)^\top \boldsymbol{A} \mu_2^{(t)}. \tag{2.2}$$

A folklore result establishes a direct connection between regret minimization and approximate Nash equilibrium. Specifically, when the players' strategies incur regret $R_1^{(T)}$ and $R_2^{(T)}$ respectively, then the *average* strategies of the players form an $\epsilon$-approximate NE, where

$$\epsilon = (R_1^{(T)} + R_2^{(T)})/T. \tag{2.3}$$

## 3 Q-FUNCTION BASED REGRET MINIMIZATION (QFR)

In this section, we propose our policy gradient algorithm for EFGs, which we coin *Q-Function based Regret Minimization (QFR)*. In QFR, for each player $i \in [2]$ and state $s \in \mathcal{S}_i$, we enforce the strategy $\pi_i^{(t)}(\cdot \mid s)$ to explore with probability $\gamma_s$ using the exploration strategy $\boldsymbol{\nu}_s$, in order to ensure that each infoset will be reached with a positive probability $\gamma > 0$.

Then, we show that QFR converges in best iterate to the *regularized Nash equilibrium*. Specifically, QFR will converge to the solution $\boldsymbol{\mu}^{(\tau,\gamma),*} = (\mu_1^{(\tau,\gamma),*}, \mu_2^{(\tau,\gamma),*})$ of the original bilinear minimax objective plus additional regularization term $\psi_{\mathrm{bi}}^{\Pi_1}(\mu_1^{\pi_1}, \mu_2^{\pi_2})$ and $\psi_{\mathrm{bi}}^{\Pi_2}(\mu_1^{\pi_1}, \mu_2^{\pi_2})$, which we call *bidilated regularizer*. $\psi_{\mathrm{bi}}^{\Pi_1}(\mu_1^{\pi_1}, \mu_2^{\pi_2})$ is strongly convex with respect to $\mu_1^{\pi_1}$ and convex with respect to $\mu_2^{\pi_2}$. Conversely, $\psi_{\mathrm{bi}}^{\Pi_2}(\mu_1^{\pi_1}, \mu_2^{\pi_2})$ is strongly convex with respect to $\mu_2^{\pi_2}$ and convex with respect to $\mu_1^{\pi_1}$. In contrast to the original bilinear objective, optimizing the regularized objective will stabilize the training process, and result in better convergence results. Formally, the *regularized and perturbed* (perturb refers to the exploration) game is,

$$\max_{\substack{\mu_1^{\pi_1} \in \Pi_1: \\ \forall s \in \mathcal{S}_1, \, \pi_1(\cdot \mid s) \in \Delta_{\gamma_s, \boldsymbol{\nu}_s}^{|\mathcal{A}_s|}}} \min_{\substack{\mu_2^{\pi_2} \in \Pi_2: \\ \forall s \in \mathcal{S}_2, \, \pi_2(\cdot \mid s) \in \Delta_{\gamma_s, \boldsymbol{\nu}_s}^{|\mathcal{A}_s|}}} \left( \mu_1^{\pi_1} \right)^\top \boldsymbol{A} \mu_2^{\pi_2} - \tau \psi_{\mathrm{bi}}^{\Pi_1}(\mu_1^{\pi_1}, \mu_2^{\pi_2}) + \tau \psi_{\mathrm{bi}}^{\Pi_2}(\mu_1^{\pi_1}, \mu_2^{\pi_2})$$

$$\tag{3.1}$$

where $\Delta_{\gamma_s, \boldsymbol{\nu}_s}^{|\mathcal{A}_s|} := \left\{ \boldsymbol{u} \in \Delta^{|\mathcal{A}_s|} : \forall a \in \mathcal{A}_s, \ u_a \geq \gamma_s \nu_{s,a} \right\}$, and $\tau \geq 0$ controls the magnitude of the regularizer. Note that the NE of the non-regularized game can be computed by annealing the regularization coefficient $\tau$ by using a standard technique (Liu et al., 2023).

To decompose the regularizer $\psi_{\mathrm{bi}}^{\Pi_i}$ to each infoset for efficient update, we resort to the concept *dilated regularizer* (Hoda et al., 2010). Take Euclidean norm for example, unlike naively choosing $\frac{1}{2} \|\mu_i^{\pi_i}\|^2$ as the regularizer, dilated regularizer weights the regularizer $\frac{1}{2} \|\pi_i(\cdot \mid s)\|^2$ at each infoset $s \in \mathcal{S}_i$ by the reach probability of player $i$ to $s$, i.e. $\frac{1}{2} \sum_{s \in \mathcal{S}_i} \mu_i^{\pi_i}(\sigma(s)) \|\pi_i(\cdot \mid s)\|^2$. It is shown in Hoda et al. (2010) that the dilated regularizer is strongly convex with respect to $\mu_i^{\pi_i}$. However, dilated regularizer $\psi^{\Pi_i}$ only weights the reach probability of player $i$, neglecting that of the opponents, of which the asymmetry causes importance sampling when estimating the regularizer term in a rolling trajectory. Therefore, a natural solution is weighting the strongly convex regularizer $\psi_s^\Delta : \Delta^{|\mathcal{A}_s|} \to \mathbb{R}$ at each infoset $s \in \mathcal{S}$ (typically it is Euclidean norm or negative entropy) by the reach probability of all players. Formally, the *bidilated* regularizer $\psi_{\mathrm{bi}}^{\Pi_1}(\mu_1^{\pi_1}, \mu_2^{\pi_2})$ of player 1 is defined as (that of player 2 can be defined similarly),

$$\psi_{\mathrm{bi}}^{\Pi_1}(\mu_1^{\pi_1}, \mu_2^{\pi_2}) := \sum_{s \in \mathcal{S}_1} \mu_1^{\pi_1}(\sigma(s)) \left( \sum_{h \in s} \mu_c(h) \mu_2^{\pi_2}(\sigma_2(h)) \right) \psi_s^\Delta(\pi_1(\cdot \mid s)), \quad \text{(Bidilated regularizer)}$$

$$\psi^{\Pi_1}(\mu_1^{\pi_1}) = \sum_{s \in \mathcal{S}_1} \mu_1^{\pi_1}(\sigma(s)) \psi_s^\Delta(\pi_1(\cdot \mid s)) \quad \text{(Dilated regularizer)}$$

The details can be referred to Appendix D. For notational simplicity, let $\psi^{\Pi}(\boldsymbol{\mu}^{\boldsymbol{\pi}}) \colon \Pi \to \mathbb{R} = \psi^{\Pi_1}(\mu_1^{\pi_1}) + \psi^{\Pi_2}(\mu_2^{\pi_2})$ and $\psi_{\text{bi}}^{\Pi}(\boldsymbol{\mu}^{\boldsymbol{\pi}}) \colon \Pi \to \mathbb{R} = \psi_{\text{bi}}^{\Pi_1}(\mu_1^{\pi_1}, \mu_2^{\pi_2}) + \psi_{\text{bi}}^{\Pi_2}(\mu_1^{\pi_1}, \mu_2^{\pi_2})$. With all ingredients at hand, we can introduce the algorithm `Q-Function based Regret minimization` (`QFR`). At each timestep $t$, `QFR` will sample a trajectory according to the current strategy $\boldsymbol{\pi}^{(t-1)}$ (Line 3 of Algorithm 1). Then, the *trajectory Q-value* will be estimated from the trajectory (Line 10-12, $\psi$ estimates the trajectory Q-value contributed by additional bidilated regularizer of the game, $W$ estimates the that contributed by the reward of the original game). Lastly, all infosets along the trajectory will be updated with a variant of Regularized Optimistic Mirror Descent (Reg-OMD) proposed in Liu et al. (2023) (Line 14). In Reg-OMD, at timestep $t$, the strategy $\boldsymbol{\pi}^{(t)}$ serves as the prediction of $\overline{\boldsymbol{\pi}}^{(t+1)}$. Then in the next timestep $t+1$, $\overline{\boldsymbol{\pi}}^{(t)}$ will be updated with the trajectory Q-value estimated at $\boldsymbol{\pi}^{(t)}$. The pseudocode of `QFR` is proposed in Algorithm 1.

---

**Algorithm 1** Q-Function based Regret minimization (QFR)

---

1: Initialize $\pi_i^{(1)}(\cdot \mid s), \overline{\pi}_i^{(1)}(\cdot \mid s)$ as uniform distribution over $\Delta^{|\mathcal{A}_s|}$ for any $i \in [2]$ and $s \in \mathcal{S}_i$.
2: **for** $t = 2, 3, \cdots, T$ **do**
3:     Sample a trajectory $(h_0, a_0, h_1, \cdots, h_K) \sim \boldsymbol{\pi}^{(t-1)}$, where $h_0 = \emptyset$ and $h_K$ is terminal
4:     Let $s_0, s_1, \cdots, s_K$ be the infosets corresponding to $h_0, h_1, \cdots, h_K$
5:     **for** $k = K, K-1, \cdots, 0$ **do**
6:        **if** node $h_k$ belongs to the chance player **then**
7:           **continue** to the new iteration directly
8:        **else**
9:           $p \leftarrow p(s_k)$
10:           $\psi \leftarrow -\sum_{k'=k+1:p(s_{k'})=p}^{K} \psi_{s_{k'}}^{\Delta}(\pi_p^{(t-1)}) + \sum_{k'=k+1:p(s_{k'})=3-p}^{K} \psi_{s_{k'}}^{\Delta}(\pi_{3-p}^{(t-1)})$
11:           $W \leftarrow \mathcal{U}_p(h_K)$, end-of-trajectory utility of player $p$ (only terminal nodes have utility $\neq 0$)
12:           Compute the unbiased estimator of trajectory Q-value as

$$\widetilde{q}^{(t-1)}(s_k, a) = \begin{cases} (W + \tau \psi)\, \nabla_a \log \pi_p^{(t-1)}(a \mid s_k) & a = a_k \\ 0 & \text{Otherwise.} \end{cases}$$

13:           Compute the estimated value function as

$$\widetilde{V}^{\pi_p}(s_k) = \mathbb{E}_{a \sim \pi_p(\cdot \mid s_k)}\left[\widetilde{q}^{(t-1)}(s_k, a)\right] - \tau \psi_{s_k}^{\Delta}\left(\pi_p(\cdot \mid s_k)\right).$$

14:           Update $\overline{\pi}_p^{(t-1)}, \pi_p^{(t-1)}$ according to

$$\overline{\pi}_p^{(t)}(\cdot \mid s_k) \leftarrow \operatorname*{argmax}_{\pi_p(\cdot \mid s_k) \in \Delta_{\gamma_{s_k}, \boldsymbol{\nu}_{s_k}}^{|\mathcal{A}_{s_k}|}} \widetilde{V}^{\pi_p}(s_k) - \frac{1}{\eta_{s_k}} D_{\psi_{s_k}^{\Delta}}\left(\pi_p(\cdot \mid s_k), \overline{\pi}_p^{(t-1)}(\cdot \mid s_k)\right)$$

$$\pi_p^{(t)}(\cdot \mid s_k) \leftarrow \operatorname*{argmax}_{\pi_p(\cdot \mid s_k) \in \Delta_{\gamma_{s_k}, \boldsymbol{\nu}_{s_k}}^{|\mathcal{A}_{s_k}|}} \widetilde{V}^{\pi_p}(s_k) - \frac{1}{\eta_{s_k}} D_{\psi_{s_k}^{\Delta}}\left(\pi_p(\cdot \mid s_k), \overline{\pi}_p^{(t)}(\cdot \mid s_k)\right), \quad (3.2)$$

$$\text{where } \Delta_{\gamma_s, \boldsymbol{\nu}_s}^{|\mathcal{A}_s|} \coloneqq \left\{\boldsymbol{u} \in \Delta^{|\mathcal{A}_s|} \colon \forall a \in \mathcal{A}_s, \ u_a \geq \gamma_s \nu_{s,a}\right\}$$

15:           For all infosets $s$ not visited at timestep $t$, let $\pi_{p(s)}^{(t)}(\cdot \mid s) = \pi_{p(s)}^{(t-1)}$ (same for $\overline{\pi}_{p(s)}^{(t)}$)

---

We define the largest learning rate among all ancestor infosets of $s \in \mathcal{S}$ as $\eta_s^{\text{anc}} \coloneqq \max_{i \in [2], h \in s} \max_{(s', a') \sqsubseteq \sigma_i(h)} \eta_{s'}$ ($\eta_s$ is the learning rate of infoset $s$), and we have the following theorem that establishes the best-iterate convergence of Algorithm 1 to the NE $\boldsymbol{\mu}^{(\tau, \gamma),*} = (\mu_1^{(\tau, \gamma),*}, \mu_2^{(\tau, \gamma),*})$ of Equation (3.1) with high-probability.

**Theorem 3.1** (Informal). Consider Algorithm 1. When $\frac{\eta_s^{\text{anc}}}{\eta_s} \leq \tau C_s^{\eta, T}$, where $C_s^{\eta}$ is a game-dependent constant, and $\eta_s$ is smaller than a game dependent constant (formally defined in Appendix F) for any $s \in \mathcal{S}$, we have the following guarantee with probability $1 - 2\delta$.

$$\sum_{t=2}^{T} D_{\psi^{\Pi}}(\boldsymbol{\mu}^{(\tau, \gamma),*}, \boldsymbol{\mu}^{\overline{\boldsymbol{\pi}}^{(t)}}) \leq \widetilde{O}\left(\max_{s \in \mathcal{S}} \eta_s T\right) + \widetilde{O}\left(\frac{1}{\min_{s \in \mathcal{S}} \eta_s}\right) + \widetilde{O}\left(\sqrt{T \log \frac{1}{\delta}}\right). \quad (3.3)$$

The $\widetilde{O}$ notion hides the logarithm of $T$. The proof and the formal version are postponed to Appendix F. Theorem 3.1 gives a high-probability upper-bound on the cumulative Bregman divergence. By letting $\eta_s = \Theta(1/\sqrt{T})$ for all infoset $s \in \mathcal{S}$, the right-hand-side of (3.3) is bounded by $\widetilde{O}(\sqrt{T})$. Therefore, it implies that $\sum_{t=2}^{T} D_{\psi^{\Pi}}(\boldsymbol{\mu}^{(\tau,\gamma),*}, \boldsymbol{\mu}^{\overline{\boldsymbol{\pi}}^{(t)}})$ is upper-bounded by $\widetilde{O}(\sqrt{T})$ with probability $1-2\delta$. Then, there must exist some $t^* \in [T]$ so that $D_{\psi^{\Pi}}(\boldsymbol{\mu}^{(\tau,\gamma),*}, \boldsymbol{\mu}^{\overline{\boldsymbol{\pi}}^{(t^*)}}) \leq \widetilde{O}(1/\sqrt{T})$, because the minimum over $\left\{ D_{\psi^{\Pi}}(\boldsymbol{\mu}^{(\tau,\gamma),*}, \boldsymbol{\mu}^{\overline{\boldsymbol{\pi}}^{(t)}}) \colon t = 2, 3, \cdots, T \right\}$ must be bounded by the average, which is $\widetilde{O}(1/\sqrt{T})$. Therefore, by computing the exploitability (the expected utility confronting a best-responding opponent) routinely, we can find an approximate NE (Liu et al., 2023) of the regularized game. Moreover, according to Lemma Liu et al. (2023, Lemma D.1.), the exploitability of the regularized NE will be bounded by $O(\tau)$ in the original game so that our iterates will also get a low exploitability in the original game by fixing $\tau$ to be small or anneal it as Liu et al. (2023).

## 4 ANALYSIS

In this section, we provide the proof sketch of Theorem 3.1. Section 4.1 introduces some necessary notions and properties for the analysis. Section 4.2 shows the convergence of QFR under full-information feedback (traversing all infosets at each iteration), and Section 4.3 generalizes to the stochastic setting in Section 3.

### 4.1 PRELIMINARIES AND BASIC PROPERTIES

In order to keep the presentation modular between Q-values and trajectory Q-values, we will assume that, at each iteration $t$, the local strategy at each infoset will be updated by taking a step in the direction of some generalized *value vector* $q^{(t)}(s, \cdot) \in \mathbb{R}^{|\mathcal{A}_s|}$. For each $a \in \mathcal{A}_s$, (we will use $\mathsf{CF}_1^{(t)}(s, a)$ instead of $\mathsf{CF}_1^{\boldsymbol{\pi}^{(t)}}(s, a)$ and $\mu_i^{(t)}$ instead of $\mu_i^{\pi_i^{(t)}}$ as the shorthand notion), the relationship of counterfactual values and the feedback is,

$$\mathsf{CF}_1^{(t)}(s, a) = \begin{cases} \sum_{h \in s} \mu_c(h) \mu_2^{(t)}(\sigma_2(h)) \cdot q^{(t)}(s, a) & q^{(t)}(s, \cdot) \text{ is Q-value} \\ \frac{q^{(t)}(s, a)}{\mu_1^{(t)}(\sigma(s))} & q^{(t)}(s, \cdot) \text{ is trajectory Q-value} \\ q^{(t)}(s, a) & q^{(t)}(s, \cdot) \text{ is counterfactual value} \end{cases} \qquad (4.1)$$

It is noteworthy that when sampling a trajectory from the root to a terminal node, the utility will be a good estimator of trajectory Q-value. Therefore, to estimate $\mathsf{CF}_1^{(t)}(s, a)$, we need to divide the reaching probability $\mu_1^{(t)}(\sigma(s))$ of $s$, which can be extremely small and thus induces a large variance. In the following, we will write $\mathsf{CF}_1^{(t)}(s, a) = m_s^{(t)} q^{(t)}(s, a)$ and $m_s^{(t)}$ is different for different types of $q^{(t)}(s, \cdot)$ according to Equation (4.1).

Note that the value of $m_s^{(t)}$ in most cases depends on the strategies that are produced by the algorithm; hence, there is some circularity in the dependence between the properties satisfied by $m_s^{(t)}$ and those satisfied by our algorithm. To break this circularity, at the heart of our correctness proof we will verify and leverage two key properties of the sequence of $m_s^{(t)}$ that arises from using our algorithm: *boundedness* and *stability*, as detailed next.

**Property 1** (Boundedness). For any $t \in [T]$ and $s \in \mathcal{S}$, we have $m_s^{(t)} \in [M_1, M_2]$ where $0 < M_1 \leq M_2 < +\infty$.

**Property 2** (Stability). For any $t \in [T-1]$ and $s \in \mathcal{S}$, we define the largest learning rate among all ancestor infosets of $s$ as $\eta_s^{\mathrm{anc}} := \max_{i \in [2], h \in s} \max_{(s',a') \sqsubseteq \sigma_i(h)} \eta_{s'}$, where $\eta_{s'}$ is the learning rate of infoset $s'$, then

$$\left| m_s^{(t+1)} - m_s^{(t)} \right| \leq C_s^- \eta_s^{\mathrm{anc}} \qquad \text{(Additive Stability)}$$

$$\left| \frac{m_s^{(t+1)}}{m_s^{(t)}} - 1 \right| \leq C_s^/ \eta_s^{\mathrm{anc}}. \qquad \text{(Multiplicative Stability)}$$

Property 1 will be satisfied by enforcing $\pi^{(t)}(\cdot \mid s) \succeq \gamma_s \boldsymbol{\nu}_s$ for every $s \in \mathcal{S}$, where $\gamma_s \in (0, 1]$ and $\boldsymbol{\nu}_s \in \Delta^{|\mathcal{A}_s|}$ are specified in Appendix G.2. The proof that our algorithm produces iterates that satisfy Property 2 can be found in Appendix C.

## 4.2 Convergence with Full Information Feedback

QFR runs a variant of Regularized Optimistic Mirror Descent (Reg-OMD) (Liu et al., 2023) algorithm to update the strategy in each infoset. For notational simplicity, we define the reach probability of opponents to an infoset $s \in \mathcal{S}$ as $\mu_{-p(s)}^{(t)}(s) := \sum_{h \in s} \mu_c(h) \mu_{3-p(s)}^{(t)}(\sigma_{3-p(s)}(h))$. The update rule is

$$
\begin{aligned}
\pi_{p(s)}^{(t)}(\cdot \mid s) = \ & \operatorname*{argmin}_{\pi_{p(s)}(\cdot \mid s) \in \Delta_{\gamma_s, \boldsymbol{\nu}_s}^{|\mathcal{A}_s|}} \left\langle \pi_{p(s)}(\cdot \mid s), -q^{(t-1)}(s, \cdot) \right\rangle + \frac{\tau \mu_{-p(s)}^{(t-1)}(s)}{m_s^{(t-1)}} \psi_s^\Delta(\pi_{p(s)}(\cdot \mid s)) \\
& + \frac{1}{\eta_s} D_{\psi_s^\Delta}(\pi_{p(s)}(\cdot \mid s), \bar{\pi}_{p(s)}^{(t)}(\cdot \mid s)) \\
\bar{\pi}_{p(s)}^{(t+1)}(\cdot \mid s) = \ & \operatorname*{argmin}_{\pi_{p(s)}(\cdot \mid s) \in \Delta_{\gamma_s, \boldsymbol{\nu}_s}^{|\mathcal{A}_s|}} \left\langle \pi_{p(s)}(\cdot \mid s), -q^{(t)}(s, \cdot) \right\rangle + \frac{\tau \mu_{-p(s)}^{(t)}(s)}{m_s^{(t)}} \psi_s^\Delta(\pi_{p(s)}(\cdot \mid s)) \\
& + \frac{1}{\eta_s} D_{\psi_s^\Delta}(\pi_{p(s)}(\cdot \mid s), \bar{\pi}_{p(s)}^{(t)}(\cdot \mid s))
\end{aligned}
\tag{4.2}
$$

where $\Delta_{\gamma_s, \boldsymbol{\nu}_s}^{|\mathcal{A}_s|} := \{\boldsymbol{u} \in \Delta^{|\mathcal{A}_s|} : \forall a \in \mathcal{A}_s, \ u_a \geq \gamma_s \nu_{s, a}\}$ and $\psi_s^\Delta$ is the regularizer chosen for infoset $s$. Here $q^{(t)}(s, \cdot)$ can be the trajectory Q-value, Q-value, or counterfactual value associated with $\boldsymbol{\pi}^{(t)}$. When $q^{(t)}(s, \cdot)$ is the trajectory Q-value, (4.2) is the full-information version of (3.2).

By analyzing the update rule (4.2), we can get the following inequality. For any $s \in \mathcal{S}$ and strategy $\pi_{p(s)}(\cdot \mid s) \in \Delta_{\gamma_s, \boldsymbol{\nu}_s}^{|\mathcal{A}_s|}$, we have

$$
\begin{aligned}
& \sum_{t=1}^T \left( \tau \mu_{-p(s)}^{(t)}(s) \psi_s^\Delta(\pi_{p(s)}^{(t)}(\cdot \mid s)) - \tau \mu_{-p(s)}^{(t)}(s) \psi_s^\Delta(\pi_{p(s)}(\cdot \mid s)) + m_s^{(t)} \langle -q^{(t)}(s, \cdot), \pi_{p(s)}^{(t)}(\cdot \mid s) - \pi_{p(s)}(\cdot \mid s) \rangle \right) \\
& \leq \sum_{t=2}^T \underbrace{\left( \frac{m_s^{(t)} - m_s^{(t-1)}}{\eta_s} - \tau \mu_{-p(s)}^{(t-1)}(s) \right)}_{\textcircled{1}} D_{\psi_s^\Delta}(\pi_{p(s)}(\cdot \mid s), \bar{\pi}_{p(s)}^{(t)}(\cdot \mid s)) + O(\eta_s T) + O\left(\frac{1}{\eta_s}\right).
\end{aligned}
\tag{4.3}
$$

Then, since $\left| m_s^{(t)} - m_s^{(t-1)} \right| \leq O(\eta_s^{\mathrm{anc}})$, $\textcircled{1} \leq -\frac{\tau}{2} \mu_{-p(s)}^{(t-1)}(s) \leq -\frac{\tau \gamma}{2} \sum_{h \in s} \mu_c(h)$ when $\eta_s^{\mathrm{anc}}$ is smaller then $\eta_s$ by a small enough constant (please refer to Appendix E for details). Then, by applying the generalized regret decomposition lemma (Liu et al., 2023) (details can be found in Lemma E.3) to Equation (4.3), the difference of $\pi_{p(s)}^{(t)}$ and $\pi_{p(s)}$ in an infoset $s$ can be extended to the difference of the whole game. Specifically, by letting

$$
\mathrm{diff}_1(\boldsymbol{\mu}^{\boldsymbol{\pi}}, \boldsymbol{\mu}^{\boldsymbol{\pi}'}) := \left( \mu_1^{\pi_1} - \mu_1^{\pi_1'} \right)^\top \boldsymbol{A} \mu_2^{\pi_2} - \tau \left( \psi_{\mathrm{bi}}^{\Pi_1}(\mu_1^{\pi_1}, \mu_2^{\pi_2}) - \psi_{\mathrm{bi}}^{\Pi_1}(\mu_1^{\pi_1'}, \mu_2^{\pi_2}) \right) + \tau \left( \psi_{\mathrm{bi}}^{\Pi_2}(\mu_1^{\pi_1}, \mu_2^{\pi_2}) - \psi_{\mathrm{bi}}^{\Pi_2}(\mu_1^{\pi_1'}, \mu_2^{\pi_2}) \right)
$$

and $\mathrm{diff}_2$ similarly, then the summation of the left-hand-side of Equation (4.3) over all infoset $s \in \mathcal{S}$ is equal to is equal to

$$
\sum_{t=1}^T \mathrm{diff}_1(\boldsymbol{\mu}^{(\tau, \gamma), *}, \boldsymbol{\mu}^{(t)}) + \mathrm{diff}_2(\boldsymbol{\mu}^{(\tau, \gamma), *}, \boldsymbol{\mu}^{(t)}) \geq 0.
\tag{4.4}
$$

The non-negativity is because $\boldsymbol{\mu}^{(\tau, \gamma), *}$ is the NE of Equation (3.1). By combining (4.3) and (4.4), we have

$$
\begin{aligned}
0 \leq & \sum_{s \in \mathcal{S}} \mu^{(\tau, \gamma), *}(\sigma(s)) \left( -\frac{\tau \gamma}{2} \sum_{h \in s} \mu_c(h) \sum_{t=2}^T D_{\psi_s^\Delta}(\pi_{p(s)}^{(\tau, \gamma), *}(\cdot \mid s), \bar{\pi}_{p(s)}^{(t)}(\cdot \mid s)) + O(\eta_s T) + O\left(\frac{1}{\eta_s}\right) \right) \\
\overset{(a)}{=} & -\frac{\tau \gamma}{2} \min_{s \in \mathcal{S}} \sum_{h \in s} \mu_c(h) \sum_{t=2}^T D_{\psi^\Pi}(\boldsymbol{\mu}^{(\tau, \gamma), *}, \boldsymbol{\mu}^{\bar{\boldsymbol{\pi}}^{(t)}}) + O(\max_{s \in \mathcal{S}} \eta_s T) + O\left(\frac{1}{\min_{s \in \mathcal{S}} \eta_s}\right).
\end{aligned}
$$

$(a)$ is by Lemma 4.1 in the following. By rearranging the terms, we can get an upperbound on $\sum_{t=2}^{T} D_{\psi^{\Pi}}(\boldsymbol{\mu}^{(\tau,\gamma),*}, \boldsymbol{\mu}^{\overline{\boldsymbol{\pi}}^{(t)}})$.

**Lemma 4.1** (Lemma D.2. in Liu et al. (2022)). For any strategy $\boldsymbol{\mu}^{\boldsymbol{\pi}}, \boldsymbol{\mu}^{\widetilde{\boldsymbol{\pi}}} \in \Pi$ and regularizer $\psi_s^{\Delta} \colon \Delta^{|\mathcal{A}_s|} \to \mathbb{R}$ for each infoset $s \in \mathcal{S}$, we have

$$D_{\psi^{\Pi}}(\boldsymbol{\mu}^{\boldsymbol{\pi}}, \boldsymbol{\mu}^{\widetilde{\boldsymbol{\pi}}}) = \sum_{s \in \mathcal{S}} \mu^{\boldsymbol{\pi}}(\sigma(s)) D_{\psi_s^{\Delta}}(\pi_{p(s)}(\cdot \mid s), \widetilde{\pi}_{p(s)}(\cdot \mid s)). \quad (4.5)$$

For completeness, the proof of the lemma can be found in Appendix E.1. Here is the full theorem.

**Theorem 4.2** (Informal). Consider the update rule (4.2) and $q^{(t)}(s, \cdot)$ is chosen to be counterfactual value, trajectory Q-value, or Q-value. When $\frac{\eta_s^{\text{anc}}}{\eta_s} \leq \tau C_s^{\eta}$, where $C_s^{\eta}$ is a game-dependent constant, and $\eta_s$ is smaller than a game dependent constant (formally defined in Appendix E) for any $s \in \mathcal{S}$, we have the following guarantee.

$$\sum_{t=2}^{T} D_{\psi^{\Pi}}(\boldsymbol{\mu}^{(\tau,\gamma),*}, \boldsymbol{\mu}^{\overline{\boldsymbol{\pi}}^{(t)}}) \leq O\left(\max_{s \in \mathcal{S}} \eta_s T\right) + O\left(\frac{1}{\min_{s \in \mathcal{S}} \eta_s}\right). \quad (4.6)$$

The proof and the formal version are postponed to Appendix E. Therefore, by choosing $\eta_s = \Theta\left(\frac{1}{\sqrt{T}}\right)$ for any $s \in \mathcal{S}$ as in Theorem 4.2, QFR enjoys best-iterate convergence with full-information feedback.

### 4.3 Convergence with Stochastic Feedback

We complement the results of Section 4.2 by showing that the best-iterate convergence guaranteed by QFR is still guaranteed when only visiting a trajectory at each iteration. The proof utilizes standard concentration inequalities, incurring an additional sublinear cost caused by the noise incurred from sampling, as recalled in the following lemma.

**Lemma 4.3** (Generalization of Proposition 1 in Farina et al. (2020)). Let $M, \widetilde{M}$ be positive constants such that $\left|f^{(t)}(\boldsymbol{u}) - f^{(t)}(\boldsymbol{u}')\right| \leq M$ and $\left|\widetilde{f}^{(t)}(\boldsymbol{u}) - \widetilde{f}^{(t)}(\boldsymbol{u}')\right| \leq \widetilde{M}$ for any $\boldsymbol{u}, \boldsymbol{u}' \in \mathcal{C}$ for any $t \in [T]$, where $\mathcal{C}$ is a convex set. Then, if for any $\boldsymbol{u}$, $\mathbb{E}[\widetilde{f}^{(t)}(\boldsymbol{u}) \mid \widetilde{f}^{(1)}, \widetilde{f}^{(2)}, \cdots, \widetilde{f}^{(t-1)}] = f^{(t)}(\boldsymbol{u})$ and $\boldsymbol{u}^{(t)}$ is deterministically influenced by $\widetilde{f}^{(1)}, \widetilde{f}^{(2)}, \cdots, \widetilde{f}^{(t-1)}$, then for any $\delta \in (0, 1)$ and $\boldsymbol{u} \in \mathcal{C}$, we have

$$\Pr\left(\sum_{t=1}^{T}\left(f^{(t)}(\boldsymbol{u}) - f^{(t)}(\boldsymbol{u}^{(t)})\right) \leq \sum_{t=1}^{T}\left(\widetilde{f}^{(t)}(\boldsymbol{u}) - \widetilde{f}^{(t)}(\boldsymbol{u}^{(t)})\right) + (M + \widetilde{M})\sqrt{2T \log \frac{1}{\delta}}\right) \geq 1 - \delta.$$

Next, we will substitute the following values into Lemma 4.3,

$$f_s^{(t)}(\boldsymbol{u}) := \frac{1}{\mu_{p(s)}^{(t)}(\sigma(s))} \langle q^{(t)}(s, \cdot), \boldsymbol{u} \rangle - \tau \mu_{-p(s)}^{(t)}(s) \psi_s^{\Delta}(\boldsymbol{u})$$

$$\widetilde{f}_s^{(t)}(\boldsymbol{u}) := \begin{cases} \frac{1}{\mu_{p(s)}^{(t)}(\sigma(s))} \langle \widetilde{q}^{(t)}(s, \cdot), \boldsymbol{u} \rangle - \frac{\tau}{\mu_{p(s)}^{(t)}(\sigma(s))} \psi_s^{\Delta}(\boldsymbol{u}) & s \text{ is visited at timestep } t \\ 0 & \text{Otherwise} \end{cases}$$

, where $\widetilde{q}^{(t)}$ is defined in Algorithm 1. The proof of $\widetilde{f}_s^{(t)}$ being an unbiased estimator of $f_s^{(t)}$ is postponed to Appendix F. Then, Equation (4.3) can be bounded by $\sum_{t=1}^{T}\left(\widetilde{f}_s^{(t)}(\pi_{p(s)}(\cdot \mid s)) - \widetilde{f}_s^{(t)}(\pi_{p(s)}^{(t)}(\cdot \mid s))\right)$, which can be further bounded by analyzing the update-rule Equation (3.2). The analysis is similar to the one in Section 4.2 and can be found in Appendix F. Finally, we have Theorem 3.1.

## 5 Experiments

In the experiments, we apply QFR in 4-Sided Liar's Dice, Leduc Poker (Southey et al., 2005), Kuhn Poker (Kuhn, 1950), and $2 \times 2$ Abrupt Dark Hex. The learning rate is the *same* in all infosets,

unlike what the theorem requires, which shows that QFR is easier to implement than what the theory suggests. Note that for MMD (Sokota et al., 2023), there is no theory for convergence when using trajectory Q-value and Q-value as feedback, while QFR has.

In order to pick hyperparameters, we performed a grid-search for QFR and MMD on learning rate $\eta$, regularization $\tau$, perturbation $\gamma$, and the regularizer is either negative entropy or Euclidean distance. For BalancedOMD (BOMD) (Bai et al., 2022) and BalancedFTRL (BFTRL) (Fiegel et al., 2023), we applied grid search to the learning rate $\eta$ and fixed the exploration rate (IX parameter) to $\frac{\eta}{20}$ as suggested in Fiegel et al. (2023). For the outcome-sampling CFR / CFR+, we also applies grid-search on the exploration parameter. The details of experiments, the comparison between QFR and baselines with full-information, and the ablation study of QFR when the strategy is approximated by neural network, can be found in Appendix H.

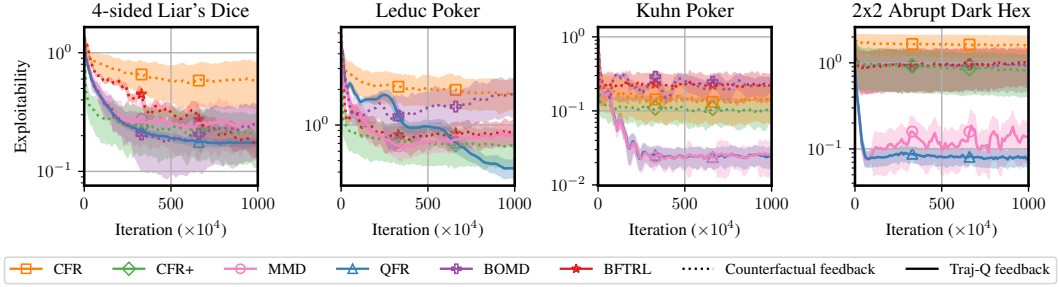

Figure 1: Exploitability of Algorithm 1 in 4 benchmark games. We can see that QFR outperforms outcome-sampling CFR / CFR+, MMD, and BOMD in all games. It outperforms BFTRL in all games except Liar's Dice. For each line, we repeat the experiments 100 times with different seeds.

The experimental result of Algorithm 1 is presented in Figure 1. In the experiments, QFR and MMD are both using an unbiased estimator of the trajectory Q-value, while CFR and CFR+ use an unbiased estimator of the counterfactual value.

Figure 1 shows that QFR outperforms outcome-sampling CFR, CFR+, and BOMD in all games. Moreover, QFR outperforms BFTRL in all games except Liar's Dice, with a relatively small gap (QFR 0.174 v.s. BFTRL 0.167 in exploitability). The reason may be Liar's Dice is too easy since it can be solved within 50 iterations with full-information feedback (see Figure 2 in Appendix H). Lastly, QFR outperforms MMD in all games.

The superiority of QFR over CFR, CFR+, BOMD, and BFTRL may be attributed to both the additional regularization (best-iterate convergence) and the avoidance of importance sampling. For MMD, QFR is superior due to the optimistic updates, since optimistic updates allow predictions of the gradients at the next iteration. The code of QFR and baselines for tabular games can be found in LiteEFG[3] (Liu et al., 2024).

# 6 CONCLUSIONS AND FUTURE WORK

In this paper, we focused on the question of whether the following properties can coexist while solving two-player zero-sum extensive-form games: **(I)** learning with random rollouts, **(II)** converging in iterates, and **(III)** avoiding importance sampling. These properties are standard desiderata in the reinforcement learning literature for single-agent settings, but have thus far eluded extensive-form games. The answer is affirmative, and we propose an algorithm, QFR, that achieves all of them. We hope this work can serve as a step in the direction of bridging the gap between reinforcement learning techniques and imperfect-information extensive-form games. In the future, it would be interesting to investigate the performance of algorithms that are both sound and work well in large domains.

---

[3]https://github.com/liumy2010/LiteEFG/tree/main/LiteEFG/baselines

## 7 ACKNOWLEDGEMENT

The authors would like to thank the support of Siebel Scholarship and NSF Award CCF-2443068.

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

# Supplementary Material

## A  ADDITIONAL RELATED WORK

This section compares our paper to prior literature on three aspects: convergence guarantees, notion of values used by the algorithm, and support of value estimation via rollouts. We provide a visual comparison of the most relevant algorithms in Table 1.

**Convergence Guarantee.** Most CFR-based algorithms (Zinkevich et al., 2007; Tammelin et al., 2015; Steinberger et al., 2020) only guarantee that the *average* strategy converges to an NE, though empirically some variants of CFR exhibit last-iterate convergence (Bowling et al., 2015; Tammelin et al., 2015). Motivated by the success of *Optimistic Mirror Descent* (OMD) achieving last-iterate convergence in normal-form games (NFGs) (Mertikopoulos et al., 2019; Wei et al., 2021; Cai et al., 2022), Farina et al. (2019b) first empirically showed that OMD also enjoys last-iterate convergence in EFGs. Then, Lee et al. (2021) theoretically proved that *Optimistic Multiplicative Weights Update* (OMWU), an instance of OMD, converges in EFGs with unique NE assumption.

**Use of Q-Values.** To achieve last-iterate convergence in EFGs, additional regularization and optimism are widely used. Perolat et al. (2021) used that approach in continuous time, using counterfactual values under full-information (*i.e.*, non-sampled) feedback. Lee et al. (2021); Liu et al. (2023) achieved last-iterate convergence in EFGs using discrete-time updates, but both of their convergence results are based on counterfactual values in the non-sampled case. Sokota et al. (2023)'s MMD algorithm empirically observed convergence by using sampled (trajectory) Q-values in conjunction with entropic regularization, without theoretical guarantees. In this paper, we combine regularization and optimism, and obtain a theoretically sound algorithm (QFR) for solving two-player zero-sum EFGs using sampled Q-values / trajectory Q-values.

**Rollout-based estimation.** Lanctot et al. (2009) proposed Outcome-Sampling Monte-Carlo CFR (OS-MCCFR), a variant of CFR which uses random rollouts to estimate counterfactual values. Later, Farina and Sandholm (2021); Farina et al. (2021b); Bai et al. (2022), and Fiegel et al. (2023) proposed algorithms that converge in EFGs with trajectories at each iteration. However, those algorithms rely on importance sampling, which causes numerical instability due to the large range of feedback. ESCHER McAleer et al. (2023) and LocalOMD (Fiegel et al., 2024) sample trajectories off-policy. This is usually undesirable as it favors exploring parts of the game tree according to uniform random probability, rather than focusing on those that are more likely given the policy. ARMAC (Gruslys et al., 2020) and ACH (Fu et al., 2021) support both Q-values and approximately-on-policy estimation, but like ESCHER and LocalOMD they do not guarantee convergence of the iterates. Moreover, neither of them is computationally efficient since they need to sample many trajectories (possibly infinite) at each iteration to ensure that the estimation of feedback is totally accurate. Also, ACH does not converge to the set of NEs, even in terms of average-iterate convergence.

## B  FORMAL DEFINITION OF TRAJECTORY Q-VALUE, Q-VALUE, AND COUNTERFACTUAL VALUE

In the following, we will formally define trajectory Q-value, Q-value, and counterfactual value respectively.

For any $s \in \mathcal{S}, a \in \mathcal{A}_s$, the *trajectory Q-value* $\overline{Q}_1^{\boldsymbol{\pi}}(s, a)$ is formally defined as,

$$\overline{Q}_1^{\boldsymbol{\pi}}(s, a) := \frac{1}{\pi_1(a \mid s)} \sum_{h': \ \exists h \in s, (h,a) \sqsubseteq h'} \mu_c(h') \mu_1^{\pi_1}(\sigma_1(h')) \mu_2^{\pi_2}(\sigma_2(h')) \mathcal{U}_1(h') \tag{B.1}$$

In (B.1), $\mu_c(h)$ denotes the probability of reaching $h$ contributed by the chance player. We use $\{\mathcal{U}_i \colon \mathcal{H} \to [-1, 1]\}_{i \in [2]}$ to denote the utility assigned to each player at each node of the game. We

assume this is nonzero only at terminal nodes (nodes with an empty action set) without loss of generality.

At the same time, we can define the *Q-value*, defined as the expected utility *conditioned on* reaching infoset $s$,

$$Q_1^{\boldsymbol{\pi}}(s,a) := \frac{\overline{Q}_1^{\boldsymbol{\pi}}(s,a)}{\sum_{h \in s} \mu_c(h) \mu_1^{\pi_1}(\sigma_1(h)) \mu_2^{\pi_2}(\sigma_2(h))}. \tag{B.2}$$

Typically, algorithms like CFR and its variants (Tammelin et al., 2015; Brown and Sandholm, 2019) use counterfactual values as feedback to learn the equilibrium. The counterfactual value at infoset $s$ of player $i$ is the Q-value at $s$ times the reach probability of the opponent and the chance player to $s$. Formally, for any $s \in \mathcal{S}, a \in \mathcal{A}_s$, we can define the *counterfactual value* $\mathsf{CF}_1^{\boldsymbol{\pi}}(s)$ as,

$$\mathsf{CF}_1^{\boldsymbol{\pi}}(s,a) := \sum_{h' \,:\, \exists h \in s, (h,a) \sqsubseteq h'} \mu_c(h') \frac{\mu_1^{\pi_1}(\sigma_1(h'))}{\mu_1^{\pi_1}(\sigma(s,a))} \mu_2^{\pi_2}(\sigma_2(h')) \mathcal{U}_1(h'). \tag{B.3}$$

$\mathsf{CF}_i^{\boldsymbol{\pi}}$ is not compatible with rolling out trajectories as reinforcement learning usually does, since a rolling trajectory includes the probability of both players reaching infoset $s$, which hinders extending algorithms to large games.

Also, for any $s \in \mathcal{S}_1$ and $a \in \mathcal{A}_s$ (similar for player 2), we have

$$\mathsf{CF}_1^{\boldsymbol{\pi}}(s,a) = \frac{1}{\mu_1^{\pi_1}(\sigma(s))} \overline{Q}_1^{\boldsymbol{\pi}}(s,a) \tag{B.4}$$

$$\mathsf{CF}_1^{\boldsymbol{\pi}}(s,a) = \frac{\sum_{h \in s} \mu_c(h) \mu_1^{\pi_1}(\sigma_1(h)) \mu_2^{\pi_2}(\sigma_2(h))}{\mu_1^{\pi_1}(\sigma(s))} \cdot Q_1^{\boldsymbol{\pi}}(s,a) \overset{(i)}{=} \sum_{h \in s} \mu_c(h) \mu_2^{\pi_2}(\sigma_2(h)) Q_1^{\boldsymbol{\pi}}(s,a), \tag{B.5}$$

where equality $(i)$ follows from $\sigma_1(h) = \sigma(s)$ for any $s \in \mathcal{S}_1$ and $h \in s$, which is a consequence to the perfect-recall assumption.

## C  Stability of Trajectory Q-value and Q-Value

In this section, we will show the stability of trajectory Q-value and Q-value, *i.e.* proving Property 2 when the regularizer in each infoset $s \in \mathcal{S}$ can be written as,

$$\psi_s^\Delta(\boldsymbol{u}) = \begin{cases} \frac{\alpha_s}{2} \sum_{a \in \mathcal{A}_s} u_a^2 & \text{(Euclidean Norm)} \\ \alpha_s \left( \log |\mathcal{A}_s| + \sum_{a \in \mathcal{A}_s} u_a \log u_a \right) & \text{(Negative Entropy)}, \end{cases} \tag{C.1}$$

where $\alpha_s > 0$ is a state-dependent constant. We add $\log |\mathcal{A}_s|$ to the negative entropy to ensure the regularizer is always positive. Previous work (Kroer et al., 2020) chose specific $\alpha_s$ to ensure the dilated regularizer associated with $\psi_s^\Delta$ is 1-strongly convex. For generality of the result, we keep the $\alpha_s$ in the regularizer.

Due to the symmetry between two players, we will only prove that for $s \in \mathcal{S}_1$. Moreover, to stabilize trajectory Q-value and Q-value, the learning rate need to satisfy the following conditions.

(A)  $\max_{h \in s} \sum_{(s',a') \sqsubseteq \sigma_i(h)} \eta_{s'} \le \eta_s$ for any $s \in \mathcal{S}, i \in [2]$

(B)  $6\eta_s^{\mathrm{anc}} \max_{s' \in \mathcal{S}} \left( \frac{2\|\boldsymbol{q}\|_\infty}{\alpha_{s'}} + \frac{\tau}{M_1} \log \frac{1}{\gamma} \right) \le 1$ for any $s \in \mathcal{S}$, where $\|\boldsymbol{q}\|_\infty := \max_{t \in [T], s \in \mathcal{S}} \|q^{(t)}(s,\cdot)\|_\infty$ and its upperbound is given in Lemma G.1

(C)  $\eta_s \left( 2\|\boldsymbol{q}\|_\infty + \frac{\tau \alpha_s}{M_1} \log \frac{1}{\gamma} \right) \le 1$ for any $s \in \mathcal{S}$

(A) ensures that $\sum_{(s',a') \sqsubseteq \sigma(s)} \eta_{s'} \le 2\eta_s^{\mathrm{anc}}$ for any $s \in \mathcal{S}$. (B), (C) ensure that the update at each iteration will not change the strategy too much.

## C.1 STABILITY OF TRAJECTORY Q-VALUE

**Lemma C.1** (Stability of $m_s^{(t)}$ under Euclidean regularizer). Consider when $\psi_s^\Delta(\boldsymbol{u}) = \frac{\alpha_s}{2}\sum_{a\in\mathcal{A}_s} u_a^2$ for any $s \in \mathcal{S}$ and (**A**) is satisfied. For any $s \in \mathcal{S}$ and $t = 1, 2, \cdots, T$, when $m_s^{(t)}$ is the trajectory Q-value feedback, we have

$$C_s^- = \frac{6}{\gamma^2}\max_{s'\in\mathcal{S}} C_{s'}^{\text{diff}} \qquad\qquad C_s^/ = \frac{6}{\gamma^2 M_1}\max_{s'\in\mathcal{S}} C_{s'}^{\text{diff}}. \tag{C.2}$$

*Proof.* For trajectory Q-value feedback, for any $s \in \mathcal{S}_1$,

$$\begin{aligned}
\left|m_s^{(t+1)} - m_s^{(t)}\right| &= \left|\frac{1}{\mu_1^{(t+1)}(\sigma(s))} - \frac{1}{\mu_1^{(t)}(\sigma(s))}\right| \\
&= \frac{\left|\mu_1^{(t+1)}(\sigma(s)) - \mu_1^{(t)}(\sigma(s))\right|}{\mu_1^{(t+1)}(\sigma(s))\mu_1^{(t)}(\sigma(s))} \\
&= \frac{\left|\prod_{(s',a')\sqsubseteq\sigma(s)} \pi_1^{(t+1)}(a'\mid s') - \prod_{(s',a')\sqsubseteq\sigma(s)} \pi_1^{(t)}(a'\mid s')\right|}{\mu_1^{(t+1)}(\sigma(s))\mu_1^{(t)}(\sigma(s))} \\
&\leq \frac{1}{\gamma^2}\sum_{(s',a')\sqsubseteq\sigma(s)} \left|\pi_1^{(t+1)}(a'\mid s') - \pi_1^{(t)}(a'\mid s')\right|.
\end{aligned}$$

In the last line, we use the fact $\mu_1^{(t+1)}(\sigma(s)), \mu_1^{(t)}(\sigma(s)) \geq \gamma$, and

$$\begin{aligned}
&\left|\prod_{(s',a')\sqsubseteq\sigma(s)} \pi_1^{(t+1)}(a'\mid s') - \prod_{(s',a')\sqsubseteq\sigma(s)} \pi_1^{(t)}(a'\mid s')\right| \\
&\leq \prod_{(s',a')\sqsubseteq\sigma_1(\sigma(s))} \pi_1^{(t+1)}(a'\mid s')\left|\pi_1^{(t+1)}(\sigma(s)) - \pi_1^{(t)}(\sigma(s))\right| \\
&\quad + \pi_1^{(t)}(\sigma(s))\left|\prod_{(s',a')\sqsubseteq\sigma(\sigma(s))} \pi_1^{(t+1)}(a'\mid s') - \prod_{(s',a')\sqsubseteq\sigma(\sigma(s))} \pi_1^{(t)}(a'\mid s')\right| \\
&\leq \left|\pi_1^{(t+1)}(\sigma(s)) - \pi_1^{(t)}(\sigma(s))\right| + \left|\prod_{(s',a')\sqsubseteq\sigma(\sigma(s))} \pi_1^{(t+1)}(a'\mid s') - \prod_{(s',a')\sqsubseteq\sigma(\sigma(s))} \pi_1^{(t)}(a'\mid s')\right|.
\end{aligned}$$

We abuse the notion of $\pi_1(\sigma(s))$ as $\pi_1(a'\mid s')$ and $\sigma(\sigma(s)) = \sigma(s')$, given $\sigma(s) = (s', a')$.

By recursively applying the process above, we will get

$$\left|\prod_{(s',a')\sqsubseteq\sigma(s)} \pi_1^{(t+1)}(a'\mid s') - \prod_{(s',a')\sqsubseteq\sigma(s)} \pi_1^{(t)}(a'\mid s')\right| \leq \sum_{(s',a')\sqsubseteq\sigma(s)} \left|\pi_1^{(t+1)}(a'\mid s') - \pi_1^{(t)}(a'\mid s')\right|.$$

**Lemma C.2.** Consider the update-rule (4.2). When we choose $\psi_s^\Delta$ to be negative entropy or Euclidean distance, we have

$$\left\|\pi_{p(s)}^{(t)}(\cdot\mid s) - \bar{\pi}_{p(s)}^{(t)}(\cdot\mid s)\right\|_1, \left\|\bar{\pi}_{p(s)}^{(t+1)}(\cdot\mid s) - \bar{\pi}_{p(s)}^{(t)}(\cdot\mid s)\right\|_1 \leq C_s^{\text{diff}}\eta_s, \tag{C.3}$$

where

$$C_s^{\text{diff}} := \begin{cases} \frac{2}{\alpha_s}\left(2\|\boldsymbol{q}\|_\infty + \frac{\tau\alpha_s}{M_1}\log\frac{1}{\gamma}\right) & \text{Negative Entropy} \\ \frac{|\mathcal{A}_s|}{\alpha_s}\|\boldsymbol{q}\|_\infty + \frac{2\sqrt{|\mathcal{A}_s|}\tau}{M_1} & \text{Euclidean Distance}. \end{cases} \tag{C.4}$$

The proof is postponed to Appendix G.3. By using Lemma C.2, we have

$$
\left| m_s^{(t+1)} - m_s^{(t)} \right|
$$

$$
\leq \frac{1}{\gamma^2} \sum_{(s',a') \sqsubseteq \sigma(s)} \left| \pi_1^{(t+1)}(a' \mid s') - \pi_1^{(t)}(a' \mid s') \right|
$$

$$
\leq \frac{1}{\gamma^2} \sum_{(s',a') \sqsubseteq \sigma(s)} \left( \left| \bar{\pi}_1^{(t+1)}(a' \mid s') - \bar{\pi}_1^{(t)}(a' \mid s') \right| + \left| \pi_1^{(t+1)}(a' \mid s') - \bar{\pi}_1^{(t+1)}(a' \mid s') \right| \right.
$$

$$
\left. + \left| \bar{\pi}_1^{(t)}(a' \mid s') - \pi_1^{(t)}(a' \mid s') \right| \right)
$$

$$
\leq \frac{3}{\gamma^2} \sum_{(s',a') \sqsubseteq \sigma(s)} C_{s'}^{\text{diff}} \eta_{s'}.
$$

At the same time,

$$
\left| \frac{m_s^{(t+1)}}{m_s^{(t)}} - 1 \right| = \frac{1}{m_s^{(t)}} \left| m_s^{(t+1)} - m_s^{(t)} \right| \leq \frac{3 \sum_{(s',a') \sqsubseteq \sigma(s)} C_{s'}^{\text{diff}} \eta_{s'}}{\gamma^2 M_1}.
$$

Therefore, $C_s^- = \frac{6}{\gamma^2} \max_{s' \in \mathcal{S}} C_{s'}^{\text{diff}}$ and $C_s^/ = \frac{6 \max_{s' \in \mathcal{S}} C_{s'}^{\text{diff}}}{\gamma^2 M_1}$ by (A). $\qquad \square$

**Lemma C.3** (Stability of $m_s^{(t)}$ under entropy regularizer). Consider when $\psi_s^\Delta(\boldsymbol{u}) = \alpha_s \left( \log |\mathcal{A}_s| + \sum_{a \in \mathcal{A}_s} u_a \log u_a \right)$ for any $s \in \mathcal{S}$, and (A), (B), (C) are satisfied. For any $s \in \mathcal{S}$ and $t = 1, 2, \cdots, T$, when $m_s^{(t)}$ is the trajectory Q-value feedback, we have

$$
C_s^- = \frac{12 \max_{s' \in \mathcal{S}} \left( \frac{2\|\boldsymbol{q}\|_\infty}{\alpha_{s'}} + \frac{\tau}{M_1} \log \frac{1}{\gamma} \right)}{\gamma} \qquad C_s^/ = 12 \max_{s' \in \mathcal{S}} \left( \frac{2\|\boldsymbol{q}\|_\infty}{\alpha_{s'}} + \frac{\tau}{M_1} \log \frac{1}{\gamma} \right). \qquad \text{(C.5)}
$$

*Proof.*

$$
\left| m_s^{(t+1)} - m_s^{(t)} \right| = \left| \frac{1}{\mu_1^{(t+1)}(\sigma(s))} - \frac{1}{\mu_1^{(t)}(\sigma(s))} \right| = \frac{1}{\mu_1^{(t+1)}(\sigma(s))} \left| \frac{\mu_1^{(t+1)}(\sigma(s))}{\mu_1^{(t)}(\sigma(s))} - 1 \right|.
$$

We will then use the following lemma which shows the multiplicative stability when using negative entropy regularizer.

**Lemma C.4.** When $\psi_s^\Delta(\boldsymbol{u}) = \alpha_s \left( \log |\mathcal{A}_s| + \sum_{a \in \mathcal{A}_s} u_a \log u_a \right)$ for any $s \in \mathcal{S}$, (A), (B), (C) are satisfied, then for any $s \in \mathcal{S}, h \in s, t = 1, 2, \cdots, T$, we have

$$
\left| \frac{\mu_1^{(t+1)}(\sigma_1(h))}{\mu_1^{(t)}(\sigma_1(h))} - 1 \right|, \left| \frac{\mu_2^{(t+1)}(\sigma_2(h))}{\mu_2^{(t)}(\sigma_2(h))} - 1 \right| \leq 12 \eta_s^{\text{anc}} \max_{s' \in \mathcal{S}} \left( \frac{2\|\boldsymbol{q}\|_\infty}{\alpha_{s'}} + \frac{\tau}{M_1} \log \frac{1}{\gamma} \right). \qquad \text{(C.6)}
$$

The proof can be found at the end of this section. Then, for any $s \in \mathcal{S}_1$, we have

$$
\left| m_s^{(t+1)} - m_s^{(t)} \right| = \frac{1}{\mu_1^{(t+1)}(\sigma(s))} \left| \frac{\mu_1^{(t+1)}(\sigma(s))}{\mu_1^{(t)}(\sigma(s))} - 1 \right| \leq \frac{12 \eta_s^{\text{anc}} \max_{s' \in \mathcal{S}} \left( \frac{2\|\boldsymbol{q}\|_\infty}{\alpha_{s'}} + \frac{\tau}{M_1} \log \frac{1}{\gamma} \right)}{\mu_1^{(t+1)}(\sigma(s))}
$$

$$
\leq \frac{12 \eta_s^{\text{anc}} \max_{s' \in \mathcal{S}} \left( \frac{2\|\boldsymbol{q}\|_\infty}{\alpha_{s'}} + \frac{\tau}{M_1} \log \frac{1}{\gamma} \right)}{\gamma}.
$$

Therefore, $C_s^- = \frac{12 \max_{s' \in \mathcal{S}} \left( \frac{2\|\boldsymbol{q}\|_\infty}{\alpha_{s'}} + \frac{\tau}{M_1} \log \frac{1}{\gamma} \right)}{\gamma}$.

Similarly, we have $C_s^/ = 12 \max_{s' \in \mathcal{S}} \left( \frac{2\|\boldsymbol{q}\|_\infty}{\alpha_{s'}} + \frac{\tau}{M_1} \log \frac{1}{\gamma} \right)$. $\qquad \square$

*Proof of Lemma C.4.* Firstly, we invoke Lemma C.5.

**Lemma C.5.** Consider update-rule (4.2). When $\psi_s^\Delta(\boldsymbol{u}) = \alpha_s \left( \log |\mathcal{A}| + \sum_{a \in \mathcal{A}_s} u_a \log u_a \right)$ for any $s \in \mathcal{S}$, and (**B**) is satisfied, for any $s \in \mathcal{S}$, $a \in \mathcal{A}_s$ and $t = 1, 2, \cdots, T$,

$$\exp\left( -\frac{\eta_s}{\alpha_s} \left( 2 \|\boldsymbol{q}\|_\infty + \frac{\tau\alpha_s}{M_1} \log\frac{1}{\gamma} \right) \right) \leq \frac{\pi^{(t)}(a \mid s)}{\bar{\pi}^{(t)}(a \mid s)}, \frac{\bar{\pi}^{(t+1)}(a \mid s)}{\bar{\pi}^{(t)}(a \mid s)} \leq \exp\left( \frac{\eta_s}{\alpha_s} \left( 2 \|\boldsymbol{q}\|_\infty + \frac{\tau\alpha_s}{M_1} \log\frac{1}{\gamma} \right) \right)$$
(C.7)

$$C_s^{\mathrm{diff}} = \frac{2}{\alpha_s} \left( 2 \|\boldsymbol{q}\|_\infty + \frac{\tau\alpha_s}{M_1} \log\frac{1}{\gamma} \right).$$
(C.8)

The proof is postponed to Appendix G.3.

By Lemma C.5, we have

$$\frac{\mu_1^{(t+1)}(\sigma_1(h))}{\mu_1^{(t)}(\sigma_1(h))}$$

$$= \frac{\prod_{(s',a') \sqsubseteq \sigma_1(h)} \pi_1^{(t+1)}(a' \mid s')}{\prod_{(s',a') \sqsubseteq \sigma_1(h)} \pi_1^{(t)}(a' \mid s')}$$

$$= \frac{\prod_{(s',a') \sqsubseteq \sigma_1(h)} \bar{\pi}_1^{(t+1)}(a' \mid s')}{\prod_{(s',a') \sqsubseteq \sigma_1(h)} \bar{\pi}_1^{(t)}(a' \mid s')} \cdot \frac{\prod_{(s',a') \sqsubseteq \sigma_1(h)} \pi_1^{(t+1)}(a' \mid s')}{\prod_{(s',a') \sqsubseteq \sigma_1(h)} \bar{\pi}_1^{(t+1)}(a' \mid s')} \cdot \frac{\prod_{(s',a') \sqsubseteq \sigma_1(h)} \bar{\pi}_1^{(t)}(a' \mid s')}{\prod_{(s',a') \sqsubseteq \sigma_1(h)} \pi_1^{(t)}(a' \mid s')}$$

$$\leq \exp\left( 3 \sum_{(s',a') \sqsubseteq \sigma_1(h)} \frac{\eta_{s'}}{\alpha_{s'}} \left( 2 \|\boldsymbol{q}\|_\infty + \frac{\tau\alpha_{s'}}{M_1} \log\frac{1}{\gamma} \right) \right)$$

$$\leq \exp\left( 3 \max_{s' \in \mathcal{S}} \left( \frac{2 \|\boldsymbol{q}\|_\infty}{\alpha_{s'}} + \frac{\tau}{M_1} \log\frac{1}{\gamma} \right) \sum_{(s',a') \sqsubseteq \sigma_1(h)} \eta_{s'} \right)$$

$$\overset{(\mathbf{A})}{\leq} \exp\left( 6 \max_{s' \in \mathcal{S}} \left( \frac{2 \|\boldsymbol{q}\|_\infty}{\alpha_{s'}} + \frac{\tau}{M_1} \log\frac{1}{\gamma} \right) \eta_s^{\mathrm{anc}} \right)$$

$$\leq 1 + 12\eta_s^{\mathrm{anc}} \max_{s' \in \mathcal{S}} \left( \frac{2 \|\boldsymbol{q}\|_\infty}{\alpha_{s'}} + \frac{\tau}{M_1} \log\frac{1}{\gamma} \right).$$

At the last line, we use the fact that $e^x \leq 1 + 2x$ for $x \in [0, 1]$. Similarly, by using $1 + x \leq e^x$, we can also get the lower-bound $1 - 6\eta_s^{\mathrm{anc}} \max_{s' \in \mathcal{S}} \left( \frac{2\|\boldsymbol{q}\|_\infty}{\alpha_{s'}} + \frac{\tau}{M_1} \log\frac{1}{\gamma} \right)$. □

## C.2 STABILITY OF Q-VALUE

**Lemma C.6** (Stability of $m_s^{(t)}$ under Euclidean regularizer)**.** Consider when $\psi_s^\Delta(\boldsymbol{u}) = \frac{\alpha_s}{2} \sum_{a \in \mathcal{A}_s} u_a^2$ for any $s \in \mathcal{S}$, and (**A**) is satisfied. For any $s \in \mathcal{S}$ and $t = 1, 2, \cdots, T$, when $m_s^{(t)}$ is the Q-value feedback, we have

$$C_s^- = 6|s| \max_{s' \in \mathcal{S}} C_{s'}^{\mathrm{diff}} \qquad\qquad C_s' = \frac{6|s|}{M_1} \max_{s' \in \mathcal{S}} C_{s'}^{\mathrm{diff}}$$
(C.9)

where $|s|$ is the number of nodes in infoset $s$.

*Proof.* With Q-value feedback, for any $s \in \mathcal{S}_1$,

$$\left| m_s^{(t+1)} - m_s^{(t)} \right| = \left| \sum_{h \in s} \mu_c(h)(\mu_2^{(t+1)}(\sigma_2(h)) - \mu_2^{(t)}(\sigma_2(h))) \right|$$

$$\leq \sum_{h \in s} \mu_c(h) \left| \mu_2^{(t+1)}(\sigma_2(h)) - \mu_2^{(t)}(\sigma_2(h)) \right|$$

$$\leq |s| \max_{h \in s} \left| \mu_2^{(t+1)}(\sigma_2(h)) - \mu_2^{(t)}(\sigma_2(h)) \right|.$$

In the last line, $|s|$ denotes the number of nodes in $s$. By similar argument as in Lemma C.1, for any $h \in s$, we have

$$\left| \mu_2^{(t+1)}(\sigma_2(h)) - \mu_2^{(t)}(\sigma_2(h)) \right|$$

$$= \left| \prod_{(s',a') \sqsubseteq \sigma_2(h)} \pi_2^{(t+1)}(a' \mid s') - \prod_{(s',a') \sqsubseteq \sigma_2(h)} \pi_2^{(t)}(a' \mid s') \right|$$

$$\leq \sum_{(s',a') \sqsubseteq \sigma_2(h)} \left| \pi_2^{(t+1)}(a' \mid s') - \pi_2^{(t)}(a' \mid s') \right|$$

$$\leq \sum_{(s',a') \sqsubseteq \sigma_2(h)} \left( \left| \bar{\pi}_2^{(t+1)}(a' \mid s') - \bar{\pi}_2^{(t)}(a' \mid s') \right| + \left| \pi_2^{(t+1)}(a' \mid s') - \bar{\pi}_2^{(t+1)}(a' \mid s') \right| \right.$$

$$\left. + \left| \bar{\pi}_2^{(t)}(a' \mid s') - \pi_2^{(t)}(a' \mid s') \right| \right)$$

$$\leq 3 \sum_{(s',a') \sqsubseteq \sigma_2(h)} C_{s'}^{\mathrm{diff}} \eta_{s'}.$$

Therefore, $\left| m_s^{(t+1)} - m_s^{(t)} \right| \leq 3|s| \max_{h \in s} \sum_{(s',a') \sqsubseteq \sigma_2(h)} C_{s'}^{\mathrm{diff}} \eta_{s'}$. Similarly,

$$\left| \frac{m_s^{(t+1)}}{m_s^{(t)}} - 1 \right| = \frac{1}{m_s^{(t)}} \left| m_s^{(t+1)} - m_s^{(t)} \right| \leq \frac{3|s|}{M_1} \max_{h \in s} \sum_{(s',a') \sqsubseteq \sigma_2(h)} C_{s'}^{\mathrm{diff}} \eta_{s'}.$$

Finally, $C_s^- = 6|s| \max_{s' \in \mathcal{S}} C_{s'}^{\mathrm{diff}}$ and $C_s^/ = \frac{6|s| \max_{s' \in \mathcal{S}} C_{s'}^{\mathrm{diff}}}{M_1}$ according to (**A**). $\qquad \square$

**Lemma C.7** (Stability of $m_s^{(t)}$ under Entropy regularizer)**.** Consider when $\psi_s^{\Delta}(\boldsymbol{u}) = \alpha_s \left( \log |\mathcal{A}_s| + \sum_{a \in \mathcal{A}_s} u_a \log u_a \right)$ for any $s \in \mathcal{S}$, and (**A**), (**B**), (**C**) are satisfied. For any $s \in \mathcal{S}$ and $t = 1, 2, \cdots, T$, when $m_s^{(t)}$ is the Q-value feedback, we have

$$C_s^- = 12 M_2 \max_{s' \in \mathcal{S}} \left( \frac{2 \|\boldsymbol{q}\|_\infty}{\alpha_{s'}} + \frac{\tau}{M_1} \log \frac{1}{\gamma} \right) \quad C_s^/ = 12 \max_{s' \in \mathcal{S}} \left( \frac{2 \|\boldsymbol{q}\|_\infty}{\alpha_{s'}} + \frac{\tau}{M_1} \log \frac{1}{\gamma} \right). \quad \text{(C.10)}$$

*Proof.* With Q-value, for any $s \in \mathcal{S}_1$, we have

$$\left| \frac{m_s^{(t+1)}}{m_s^{(t)}} - 1 \right| = \left| \frac{\sum_{h \in s} \mu_c(h) \mu_2^{(t+1)}(\sigma_2(h))}{\sum_{h \in s} \mu_c(h) \mu_2^{(t)}(\sigma_2(h))} - 1 \right|.$$

By Lemma C.4, we have

$$\frac{\mu_2^{(t+1)}(\sigma_2(h))}{\mu_2^{(t)}(\sigma_2(h))} \leq 1 + 12 \eta_s^{\mathrm{anc}} \max_{s' \in \mathcal{S}} \left( \frac{2 \|\boldsymbol{q}\|_\infty}{\alpha_{s'}} + \frac{\tau}{M_1} \log \frac{1}{\gamma} \right)$$

Therefore,

$$\frac{\sum_{h \in s} \mu_c(h) \mu_2^{(t+1)}(\sigma_2(h))}{\sum_{h \in s} \mu_c(h) \mu_2^{(t)}(\sigma_2(h))}$$

$$\leq \frac{\sum_{h \in s} \mu_c(h) \left( 1 + 12 \eta_s^{\mathrm{anc}} \max_{s' \in \mathcal{S}} \left( \frac{2 \|\boldsymbol{q}\|_\infty}{\alpha_{s'}} + \frac{\tau}{M_1} \log \frac{1}{\gamma} \right) \right) \mu_2^{(t)}(\sigma_2(h))}{\sum_{h \in s} \mu_c(h) \mu_2^{(t)}(\sigma_2(h))}$$

$$\leq 1 + 12 \eta_s^{\mathrm{anc}} \max_{s' \in \mathcal{S}} \left( \frac{2 \|\boldsymbol{q}\|_\infty}{\alpha_{s'}} + \frac{\tau}{M_1} \log \frac{1}{\gamma} \right).$$

Similarly, we have

$$\frac{\sum_{h \in s} \mu_c(h) \mu_2^{(t+1)}(\sigma_2(h))}{\sum_{h \in s} \mu_c(h) \mu_2^{(t)}(\sigma_2(h))} \geq 1 - 12 \eta_s^{\mathrm{anc}} \max_{s' \in \mathcal{S}} \left( \frac{2 \|\boldsymbol{q}\|_\infty}{\alpha_{s'}} + \frac{\tau}{M_1} \log \frac{1}{\gamma} \right).$$

Therefore, $\left| \frac{m_s^{(t+1)}}{m_s^{(t)}} - 1 \right| \leq 12\eta_s^{\mathrm{anc}} \max_{s' \in \mathcal{S}} \left( \frac{2\|\boldsymbol{q}\|_\infty}{\alpha_{s'}} + \frac{\tau}{M_1} \log \frac{1}{\gamma} \right)$. At the same time,

$$\left| m_s^{(t+1)} - m_s^{(t)} \right| = m_s^{(t)} \left| \frac{m_s^{(t+1)}}{m_s^{(t)}} - 1 \right| \leq 12\eta_s^{\mathrm{anc}} M_2 \max_{s' \in \mathcal{S}} \left( \frac{2\|\boldsymbol{q}\|_\infty}{\alpha_{s'}} + \frac{\tau}{M_1} \log \frac{1}{\gamma} \right). \qquad \square$$

## D  BIDILATED REGULARIZER

Dilated regularizer (Hoda et al., 2010) is the foundation of previous work (Lee et al., 2021; Liu et al., 2023; Sokota et al., 2023) to apply mirror-descent and its variants on sequence-form strategies. Recently, additional regularization has become a powerful tool for learning in EFGs (Liu et al., 2023; Sokota et al., 2023). Specifically, we can change the objective of the game to $\max_{\mu_1^{\pi_1} \in \Pi_1} \min_{\mu_2^{\pi_2} \in \Pi_2} (\mu_1^{\pi_1})^\top \boldsymbol{A} \mu_2^{\pi_2} - \tau\psi^{\Pi_1}(\mu_1^{\pi_1}) + \tau\psi^{\Pi_2}(\mu_2^{\pi_2})$, where $\tau\psi^{\Pi_1}(\mu_1^{\pi_1}), \tau\psi^{\Pi_2}(\mu_2^{\pi_2})$ is the additional regularizer and $\tau$ controls its magnitude. By adding the additional regularizer, the objective becomes strongly convex-concave instead of convex-concave, and thus linear convergence rate can be achieved.

However, the dilated regularizer of player $i \in [2]$ is $\psi^{\Pi_i}(\mu_i^{\pi_i}) = \sum_{s \in \mathcal{S}_i} \mu_i^{\pi_i}(\sigma(s))\psi_s^\Delta(\pi_i(\cdot \mid s))$, which only counts the reach probability $\mu_i^{\pi_i}(\sigma(s))$ of player $i$. Therefore, when sampling a trajectory, to estimate the additional regularization, importance sampling is needed to offset the reach probability of player $3 - i$ and the chance player, which causes a large dispersion of feedback. Therefore, to avoid importance sampling on the regularizer, we propose the bidilated regularizer in this section, to which all players contribute symmetrically. The bidilated regularizer of player 1 is defined as,

$$\psi_{\mathrm{bi}}^{\Pi_1}(\mu_1^{\pi_1}, \mu_2^{\pi_2}) := \sum_{s \in \mathcal{S}_1} \mu_1^{\pi_1}(\sigma(s)) \left( \sum_{h \in s} \mu_c(h)\mu_2^{\pi_2}(\sigma_2(h)) \right) \psi_s^\Delta(\pi_1(\cdot \mid s)). \tag{D.1}$$

The additional term is the probability of reaching infoset $s$ contributed by player 2 and the chance player. The bidilated regularizer for player 2 can also be defined similarly. In the following, we will show that several preferable properties of dilated regularizer still hold for its bidilated version.

Firstly, the bidilated regularizer $\psi_{\mathrm{bi}}^{\Pi_1}(\mu_1^{\pi_1}, \mu_2^{\pi_2})$ is still convex with respect to $\mu_1^{\pi_1}$ and $\mu_2^{\pi_2}$ individually. This can be inferred from the fact that the dilated regularizer is convex with respect to $\mu_1^{\pi_1}$ (Hoda et al., 2010). By enforcing $\pi_2(\cdot \mid s) \geq \gamma_s \boldsymbol{\nu}_s$ for every $s \in \mathcal{S}_2$ with $\gamma_s > 0, \boldsymbol{\nu}_s \in \Delta^{|\mathcal{A}_s|}$, where $\boldsymbol{\nu}_s$ has full support, we have $\mu_2^{\pi_2}(s, a) \geq \gamma > 0$ for any $s \in \mathcal{S}_2, a \in \mathcal{A}_s$, where $\gamma$ is a constant. Then, we have the following lemma.

**Lemma D.1.** For any $\tau, \gamma > 0$, the Nash equilibrium $\boldsymbol{\mu}^{(\tau,\gamma),*} = (\mu_1^{(\tau,\gamma),*}, \mu_2^{(\tau,\gamma),*})$ of Equation (3.1) is unique.

*Proof.* Let define $F_1^\tau(\mu_1^{\pi_1}, \mu_2^{\pi_2}) := -\boldsymbol{A}\mu_2^{\pi_2} + \tau\nabla_{\mu_1^{\pi_1}}\psi_{\mathrm{bi}}^{\Pi_1}(\mu_1^{\pi_1}, \mu_2^{\pi_2}) - \tau\nabla_{\mu_1^{\pi_1}}\psi_{\mathrm{bi}}^{\Pi_2}(\mu_1^{\pi_1}, \mu_2^{\pi_2})$ and $F_2^\tau(\mu_1^{\pi_1}, \mu_2^{\pi_2}) := \boldsymbol{A}^\top \mu_1^{\pi_1} + \tau\nabla_{\mu_2^{\pi_2}}\psi_{\mathrm{bi}}^{\Pi_2}(\mu_1^{\pi_1}, \mu_2^{\pi_2}) - \tau\nabla_{\mu_2^{\pi_2}}\psi_{\mathrm{bi}}^{\Pi_2}(\mu_1^{\pi_1}, \mu_2^{\pi_2})$.

For any $\mu_1^{\pi_1'} \in \Pi_1$, we have

$$\begin{aligned}
&\left\langle F_1^\tau(\mu_1^{\pi_1}, \mu_2^{\pi_2}), \mu_1^{\pi_1} - \mu_1^{\pi_1'} \right\rangle \\
&= \left\langle -\boldsymbol{A}\mu_2^{\pi_2} + \tau\nabla_{\mu_1^{\pi_1}}\psi_{\mathrm{bi}}^{\Pi_1}(\mu_1^{\pi_1}, \mu_2^{\pi_2}) - \tau\nabla_{\mu_1^{\pi_1}}\psi_{\mathrm{bi}}^{\Pi_2}(\mu_1^{\pi_1}, \mu_2^{\pi_2}), \mu_1^{\pi_1} - \mu_1^{\pi_1'} \right\rangle \\
&= -\left\langle \boldsymbol{A}\mu_2^{\pi_2}, \mu_1^{\pi_1} - \mu_1^{\pi_1'} \right\rangle + \tau D_{\psi_{\mathrm{bi}}^{\Pi_1}(\cdot, \mu_2^{\pi_2})}\left( \mu_1^{\pi_1'}, \mu_1^{\pi_1} \right) \\
&\quad - \tau\psi_{\mathrm{bi}}^{\Pi_1}(\mu_1^{\pi_1'}, \mu_2^{\pi_2}) + \tau\psi_{\mathrm{bi}}^{\Pi_1}(\mu_1^{\pi_1}, \mu_2^{\pi_2}) - \tau\psi_{\mathrm{bi}}^{\Pi_2}(\mu_1^{\pi_1}, \mu_2^{\pi_2}) + \tau\psi_{\mathrm{bi}}^{\Pi_2}(\mu_1^{\pi_1'}, \mu_2^{\pi_2}).
\end{aligned}$$

The last line uses the fact that $\psi_{\mathrm{bi}}^{\Pi_2}(\mu_1^{\pi_1}, \mu_2^{\pi_2})$ is linear with respect to $\mu_1^{\pi_1}$.

The counterpart of $\mu_2^{\pi_2}$ is

$$
\left\langle F_2^\tau(\mu_1^{\pi_1}, \mu_2^{\pi_2}), \mu_2^{\pi_2} - \mu_2^{\pi_2'} \right\rangle
$$
$$
= \left\langle \boldsymbol{A}^\top \mu_1^{\pi_1} + \tau \nabla_{\mu_2^{\pi_2}} \psi_{\mathrm{bi}}^{\Pi_2}(\mu_1^{\pi_1}, \mu_2^{\pi_2}) - \tau \nabla_{\mu_2^{\pi_2}} \psi_{\mathrm{bi}}^{\Pi_2}(\mu_1^{\pi_1}, \mu_2^{\pi_2}), \mu_2^{\pi_2} - \mu_2^{\pi_2'} \right\rangle
$$
$$
= \left\langle \boldsymbol{A}^\top \mu_1^{\pi_1}, \mu_2^{\pi_2} - \mu_2^{\pi_2'} \right\rangle + \tau D_{\psi_{\mathrm{bi}}^{\Pi_2}(\mu_1^{\pi_1}, \cdot)}\left(\mu_2^{\pi_2'}, \mu_2^{\pi_2}\right)
$$
$$
- \tau \psi_{\mathrm{bi}}^{\Pi_2}(\mu_1^{\pi_1}, \mu_2^{\pi_2'}) + \tau \psi_{\mathrm{bi}}^{\Pi_2}(\mu_1^{\pi_1}, \mu_2^{\pi_2}) - \tau \psi_{\mathrm{bi}}^{\Pi_1}(\mu_1^{\pi_1}, \mu_2^{\pi_2}) + \tau \psi_{\mathrm{bi}}^{\Pi_1}(\mu_1^{\pi_1}, \mu_2^{\pi_2'}).
$$

Let $\boldsymbol{\mu}^{\boldsymbol{\pi}} = (\mu_1^{\pi_1}, \mu_2^{\pi_2})$ and $F^\tau(\boldsymbol{\mu}^{\boldsymbol{\pi}}) = (F_1^\tau(\mu_1^{\pi_1}, \mu_2^{\pi_2}), F_2^\tau(\mu_1^{\pi_1}, \mu_2^{\pi_2}))$. Then by taking the summation of equations above, we have

$$
\left\langle F^\tau(\boldsymbol{\mu}^{\boldsymbol{\pi}}), \boldsymbol{\mu}^{\boldsymbol{\pi}} - \boldsymbol{\mu}^{\boldsymbol{\pi}'} \right\rangle
$$
$$
= -(\mu_1^{\pi_1})^\top \boldsymbol{A} \mu_2^{\pi_2'} + (\mu_1^{\pi_1'})^\top \boldsymbol{A} \mu_2^{\pi_2} + \tau D_{\psi_{\mathrm{bi}}^{\Pi_1}(\cdot, \mu_2^{\pi_2})}\left(\mu_1^{\pi_1'}, \mu_1^{\pi_1}\right) + \tau D_{\psi_{\mathrm{bi}}^{\Pi_2}(\mu_1^{\pi_1}, \cdot)}\left(\mu_2^{\pi_2'}, \mu_2^{\pi_2}\right)
$$
$$
+ \tau \psi_{\mathrm{bi}}^{\Pi_1}(\mu_1^{\pi_1}, \mu_2^{\pi_2'}) - \tau \psi_{\mathrm{bi}}^{\Pi_1}(\mu_1^{\pi_1'}, \mu_2^{\pi_2}) + \tau \psi_{\mathrm{bi}}^{\Pi_2}(\mu_1^{\pi_1'}, \mu_2^{\pi_2}) - \tau \psi_{\mathrm{bi}}^{\Pi_2}(\mu_1^{\pi_1}, \mu_2^{\pi_2'}).
$$

Then,

$$
\left\langle F^\tau(\boldsymbol{\mu}^{\boldsymbol{\pi}}) - F^\tau(\boldsymbol{\mu}^{\boldsymbol{\pi}'}), \boldsymbol{\mu}^{\boldsymbol{\pi}} - \boldsymbol{\mu}^{\boldsymbol{\pi}'} \right\rangle
$$
$$
= \tau \Big( D_{\psi_{\mathrm{bi}}^{\Pi_1}(\cdot, \mu_2^{\pi_2})}\left(\mu_1^{\pi_1'}, \mu_1^{\pi_1}\right) + D_{\psi_{\mathrm{bi}}^{\Pi_2}(\mu_1^{\pi_1}, \cdot)}\left(\mu_2^{\pi_2'}, \mu_2^{\pi_2}\right)
$$
$$
+ D_{\psi_{\mathrm{bi}}^{\Pi_1}(\cdot, \mu_2^{\pi_2'})}\left(\mu_1^{\pi_1}, \mu_1^{\pi_1'}\right) + D_{\psi_{\mathrm{bi}}^{\Pi_2}(\mu_1^{\pi_1'}, \cdot)}\left(\mu_2^{\pi_2}, \mu_2^{\pi_2'}\right) \Big).
$$

Since $\boldsymbol{\mu}^{\boldsymbol{\pi}}, \boldsymbol{\mu}^{\boldsymbol{\pi}'} \succeq \gamma$, $D_{\psi_{\mathrm{bi}}^{\Pi_1}(\cdot, \mu_2^{\pi_2'})}\left(\mu_1^{\pi_1}, \mu_1^{\pi_1'}\right) \geq \gamma \min_{h \in \mathcal{H}} \mu_c(h) D_{\psi^{\Pi_1}}\left(\mu_1^{\pi_1}, \mu_1^{\pi_1'}\right)$ by Lemma 4.1. Moreover, there exists $M > 0$ so that $D_{\psi^{\Pi_1}}\left(\mu_1^{\pi_1}, \mu_1^{\pi_1'}\right) \geq M \left\|\mu_1^{\pi_1} - \mu_1^{\pi_1'}\right\|^2$ according to Hoda et al. (2010); Lee et al. (2021). Therefore, the NE is unique when $\tau, \gamma > 0$ by Rosen (1965). $\square$

## E  PROOF OF THEOREM 4.2

**Theorem E.1** (Formal Version of Theorem 4.2). Consider the update rule (4.2) and $q^{(t)}(s, \cdot)$ is chosen to be counterfactual value, trajectory Q-value, or Q-value. When $\frac{\eta_s^{\mathrm{anc}}}{\eta_s} \leq \tau C_s^\eta$, where $C_s^\eta := \frac{\gamma}{2C_s^-} \sum_{h \in s} \mu_c(h)$ for any $s \in \mathcal{S}$ and (**A**), (**B**), (**C**) are satisfied, we have the following guarantee.

$$
\sum_{t=2}^T D_{\psi^\Pi}(\boldsymbol{\mu}^{(\tau,\gamma),*}, \boldsymbol{\mu}^{\overline{\boldsymbol{\pi}}^{(t)}})
$$
$$
\leq \frac{2}{\gamma \min_{s \in \mathcal{S}} \sum_{h \in s} \mu_c(h)} \sum_{s \in \mathcal{S}} \left( C_s^/ + C_s^{-,Q} \right) \eta_s^{\mathrm{anc}} \mu^{(\tau,\gamma),*}(\sigma(s)) \sum_{t=1}^T \left| \psi_s^\Delta(\pi_{p(s)}^{(t)}(\cdot \,|\, s)) - \psi_s^\Delta(\overline{\pi}_{p(s)}^{(t+1)}(\cdot \,|\, s)) \right|
$$
$$
+ \frac{4}{\tau \gamma \min_{s \in \mathcal{S}} \sum_{h \in s} \mu_c(h)} \sum_{s \in \mathcal{S}} C_s^{\mathrm{diff}} \mu^{(\tau,\gamma),*}(\sigma(s)) \|\boldsymbol{q}\|_\infty \eta_s M_2 T \tag{E.1}
$$
$$
+ \frac{2}{\tau \gamma \min_{s \in \mathcal{S}} \sum_{h \in s} \mu_c(h)} \sum_{s \in \mathcal{S}} \frac{m_s^{(1)}}{\eta_s} \mu^{(\tau,\gamma),*}(\sigma(s)) D_{\psi_s^\Delta}(\pi_{p(s)}^{(\tau,\gamma),*}(\cdot \,|\, s), \overline{\pi}_{p(s)}^{(1)}(\cdot \,|\, s)),
$$

where $C_s^{-,Q}$ denotes $C_s^-$ associated with Q-value, regardless of which feedback type $q^{(t)}(s, \cdot)$ is.

**Proof Sketch.**  The structure of this section will be as follows. (i). By analyzing the update-rule (4.2), we can get the difference of utilities between our strategy $\boldsymbol{\pi}^{(t)}$ and an arbitrary strategy $\boldsymbol{\pi}$ at a single timestep $t$ in each infoset. (ii). By telescoping and using the smoothness (the strategy as well as the feedback will not change much at each iteration) of the update-rule, we can further get an upperbound on the cumulated difference. (iii). By decomposition lemma (Liu et al., 2023), the

difference in each infoset can be extended to the difference of utility in the whole game. Then, by rearranging the terms we can get an upperbound on the cumulated distance to the NE.

Firstly, we will use a standard analysis of the update rule (4.2). For notational simplicity, we define $\mu_{-p(s)}^{(t)}(s) \coloneqq \sum_{h \in s} \mu_c(h) \mu_{3-p(s)}^{(t)}(\sigma_{3-p(s)}(h))$.

**Lemma E.2** (Generalized from Lemma C.2. in Liu et al. (2023))**.** Consider the update rule in (4.2). When $\psi_s^\Delta$ is strongly convex, then for any $\pi_{p(s)}(\cdot \mid s) \in \Delta^{|\mathcal{A}_s|}$ and $t \geq 1$, we have

$$
\eta_s \frac{\tau \mu_{-p(s)}^{(t)}(s)}{m_s^{(t)}} \psi_s^\Delta(\pi_{p(s)}^{(t)}(\cdot \mid s)) - \eta_s \frac{\tau \mu_{-p(s)}^{(t)}(s)}{m_s^{(t)}} \psi_s^\Delta(\pi_{p(s)}(\cdot \mid s))
$$

$$
+ \eta_s \tau \left( \frac{\mu_{-p(s)}^{(t-1)}(s)}{m_s^{(t-1)}} - \frac{\mu_{-p(s)}^{(t)}(s)}{m_s^{(t)}} \right) \left( \psi_s^\Delta(\pi_{p(s)}^{(t)}(\cdot \mid s)) - \psi_s^\Delta(\bar{\pi}_{p(s)}^{(t+1)}(\cdot \mid s)) \right)
$$

$$
+ \eta_s \left\langle -q^{(t)}(s, \cdot), \pi_{p(s)}^{(t)}(\cdot \mid s) - \pi_{p(s)}(\cdot \mid s) \right\rangle
$$

$$
\leq D_{\psi_s^\Delta}(\pi_{p(s)}(\cdot \mid s), \bar{\pi}_{p(s)}^{(t)}(\cdot \mid s)) - (1 + \eta_s \frac{\tau \mu_{-p(s)}^{(t)}(s)}{m_s^{(t)}}) D_{\psi_s^\Delta}(\pi_{p(s)}(\cdot \mid s), \bar{\pi}_{p(s)}^{(t+1)}(\cdot \mid s))
$$

$$
- (1 + \eta_s \frac{\tau \mu_{-p(s)}^{(t-1)}(s)}{m_s^{(t-1)}}) D_{\psi_s^\Delta}(\bar{\pi}_{p(s)}^{(t+1)}(\cdot \mid s), \pi_{p(s)}^{(t)}(\cdot \mid s))
$$

$$
- D_{\psi_s^\Delta}(\pi_{p(s)}^{(t)}(\cdot \mid s), \bar{\pi}_{p(s)}^{(t)}(\cdot \mid s)) + \eta_s \left\langle q^{(t-1)}(s, \cdot) - q^{(t)}(s, \cdot), \pi_{p(s)}^{(t)}(\cdot \mid s) - \bar{\pi}_{p(s)}^{(t+1)}(\cdot \mid s) \right\rangle.
$$

The proof is postponed to Appendix E.2.

Multiplying $m_s^{(t)}$ on both sides of Lemma E.2, we have

$$
\eta_s \tau \mu_{-p(s)}^{(t)}(s) \psi_s^\Delta(\pi_{p(s)}^{(t)}(\cdot \mid s)) - \eta_s \tau \mu_{-p(s)}^{(t)}(s) \psi_s^\Delta(\pi_{p(s)}(\cdot \mid s))
$$

$$
+ \eta_s \tau \left( \frac{m_s^{(t)}}{m_s^{(t-1)}} \mu_{-p(s)}^{(t-1)}(s) - \mu_{-p(s)}^{(t)}(s) \right) \left( \psi_s^\Delta(\pi_{p(s)}^{(t)}(\cdot \mid s)) - \psi_s^\Delta(\bar{\pi}_{p(s)}^{(t+1)}(\cdot \mid s)) \right)
$$

$$
+ \eta_s m_s^{(t)} \langle -q^{(t)}(s, \cdot), \pi_{p(s)}^{(t)}(\cdot \mid s) - \pi_{p(s)}(\cdot \mid s) \rangle
$$

$$
\leq m_s^{(t)} D_{\psi_s^\Delta}(\pi_{p(s)}(\cdot \mid s), \bar{\pi}_{p(s)}^{(t)}(\cdot \mid s)) - (m_s^{(t)} + \eta_s \tau \mu_{-p(s)}^{(t)}(s)) D_{\psi_s^\Delta}(\pi_{p(s)}(\cdot \mid s), \bar{\pi}_{p(s)}^{(t+1)}(\cdot \mid s))
$$

$$
- (m_s^{(t)} + \eta_s \tau \frac{m_s^{(t)}}{m_s^{(t-1)}} \mu_{-p(s)}^{(t-1)}(s)) D_{\psi_s^\Delta}(\bar{\pi}_{p(s)}^{(t+1)}(\cdot \mid s), \pi_{p(s)}^{(t)}(\cdot \mid s))
$$

$$
- m_s^{(t)} D_{\psi_s^\Delta}(\pi_{p(s)}^{(t)}(\cdot \mid s), \bar{\pi}_{p(s)}^{(t)}(\cdot \mid s)) + \eta_s m_s^{(t)} \left\langle q^{(t-1)}(s, \cdot) - q^{(t)}(s, \cdot), \pi_{p(s)}^{(t)}(\cdot \mid s) - \bar{\pi}_{p(s)}^{(t+1)}(\cdot \mid s) \right\rangle.
$$

By noticing the fact that $\mu_{-p(s)}^{(t)}(s)$ is equal to $m_s^{(t)}$ associated with Q-value, we can use Property 1 and Property 2 and get,

$$
\left| \frac{m_s^{(t)}}{m_s^{(t-1)}} \mu_{-p(s)}^{(t-1)}(s) - \mu_{-p(s)}^{(t)}(s) \right| \leq \left| \frac{m_s^{(t)}}{m_s^{(t-1)}} - 1 \right| \mu_{-p(s)}^{(t-1)}(s) + \left| \mu_{-p(s)}^{(t-1)}(s) - \mu_{-p(s)}^{(t)}(s) \right|
$$

$$
\leq C_s' \eta_s^{\mathrm{anc}} + C_s^{-,Q} \eta_s^{\mathrm{anc}}.
$$

We also use the fact that $\mu_{-p(s)}^{(t)}(s) \leq 1$ in the last inequality. We use $C_s^{-,Q}$ to denote the $C_s^-$ associated with Q-value for simplicity.

Furthermore, by using Lemma C.2 and Hölder's Inequality, we have

$$
\left| \left\langle q^{(t-1)}(s, \cdot) - q^{(t)}(s, \cdot), \pi_{p(s)}^{(t)}(\cdot \mid s) - \bar{\pi}_{p(s)}^{(t+1)}(\cdot \mid s) \right\rangle \right|
$$

$$
\leq \left\| q^{(t)}(s, \cdot) - q^{(t-1)}(s, \cdot) \right\|_\infty \cdot \left\| \pi_{p(s)}^{(t)}(\cdot \mid s) - \bar{\pi}_{p(s)}^{(t+1)}(\cdot \mid s) \right\|_1 \leq 2 C_s^{\mathrm{diff}} \|q\|_\infty \eta_s.
$$

where $\|q\|_\infty = \max_{t\in[T],s\in\mathcal{S}} \|q^{(t)}(s,\cdot)\|_\infty$.

By telescoping and non-negativity of Bregman divergence, we have

$$
\sum_{t=1}^{T} \Big( \eta_s \tau \mu_{-p(s)}^{(t)}(s) \psi_s^\Delta(\pi_{p(s)}^{(t)}(\cdot\,|\,s)) - \eta_s \tau \mu_{-p(s)}^{(t)}(s) \psi_s^\Delta(\pi_{p(s)}(\cdot\,|\,s))
$$

$$
+ \eta_s m_s^{(t)} \langle -q^{(t)}(s,\cdot), \pi_{p(s)}^{(t)}(\cdot\,|\,s) - \pi_{p(s)}(\cdot\,|\,s) \rangle \Big)
$$

$$
\leq \sum_{t=2}^{T} \underbrace{\Big( m_s^{(t)} - m_s^{(t-1)} - \eta_s \tau \mu_{-p(s)}^{(t-1)}(s) \Big)}_{①} D_{\psi_s^\Delta}(\pi_{p(s)}(\cdot\,|\,s), \bar\pi_{p(s)}^{(t)}(\cdot\,|\,s))
$$

$$
+ \Big( C_s^/ + C_s^{-,Q} \Big) \eta_s^{\mathrm{anc}} \eta_s \tau \sum_{t=1}^{T} \Big| \psi_s^\Delta(\pi_{p(s)}^{(t)}(\cdot\,|\,s)) - \psi_s^\Delta(\bar\pi_{p(s)}^{(t+1)}(\cdot\,|\,s)) \Big| + 2 C_s^{\mathrm{diff}} \|q\|_\infty \eta_s^2 M_2 T
$$

$$
+ m_s^{(1)} D_{\psi_s^\Delta}(\pi_{p(s)}(\cdot\,|\,s), \bar\pi_{p(s)}^{(1)}(\cdot\,|\,s)).
$$

① can be upper-bounded by $C_s^- \eta_s^{\mathrm{anc}} - \eta_s \tau\gamma \sum_{h\in s} \mu_c(h) \leq -\frac{\eta_s\tau\gamma}{2}\sum_{h\in s}\mu_c(h)$ by Property 2 and letting $\frac{\eta_s^{\mathrm{anc}}}{\eta_s} \leq \frac{\tau\gamma}{2C_s^-}\sum_{h\in s}\mu_c(h)$. By non-negativity of Bregman divergence, we have

$$
\sum_{t=1}^{T} \Big( \eta_s \tau \mu_{-p(s)}^{(t)}(s) \psi_s^\Delta(\pi_{p(s)}^{(t)}(\cdot\,|\,s)) - \eta_s \tau \mu_{-p(s)}^{(t)}(s) \psi_s^\Delta(\pi_{p(s)}(\cdot\,|\,s))
$$

$$
+ \eta_s m_s^{(t)} \langle -q^{(t)}(s,\cdot), \pi_{p(s)}^{(t)}(\cdot\,|\,s) - \pi_{p(s)}(\cdot\,|\,s) \rangle \Big)
$$

$$
\leq - \frac{\eta_s \tau\gamma \sum_{h\in s}\mu_c(h)}{2} \sum_{t=2}^{T} D_{\psi_s^\Delta}(\pi_{p(s)}(\cdot\,|\,s), \bar\pi_{p(s)}^{(t)}(\cdot\,|\,s))
$$

$$
+ \Big( C_s^/ + C_s^{-,Q} \Big) \eta_s \tau \eta_s^{\mathrm{anc}} \sum_{t=1}^{T} \Big| \psi_s^\Delta(\pi_{p(s)}^{(t)}(\cdot\,|\,s)) - \psi_s^\Delta(\bar\pi_{p(s)}^{(t+1)}(\cdot\,|\,s)) \Big|
$$

$$
+ 2 C_s^{\mathrm{diff}} \|q\|_\infty \eta_s^2 M_2 T + m_s^{(1)} D_{\psi_s^\Delta}(\pi_{p(s)}(\cdot\,|\,s), \bar\pi_{p(s)}^{(1)}(\cdot\,|\,s)).
$$

Then, we will use the following regret decomposition lemma to extend the difference within an infoset above to the difference of the game.

**Lemma E.3** (Lemma 5.1 in Liu et al. (2023)). Let $\Pi := \Pi_1 \times \Pi_2$, the polytope of all valid sequence-form joint strategies. For any $\mu_1^{\pi_1} \in \Pi_1, \mu_2^{\pi_2} \in \Pi_2$, we let $\boldsymbol{\mu^\pi} = (\mu_1^{\pi_1}, \mu_2^{\pi_2}) \in \Pi$ to denote the joint strategy, $\psi^\Pi(\boldsymbol{\mu^\pi}): \Pi \to \mathbb{R} = \psi^{\Pi_1}(\mu_1^{\pi_1}) + \psi^{\Pi_2}(\mu_2^{\pi_2})$, and $F(\boldsymbol{\mu^\pi}) := (-A\mu_2^{\pi_2}, A^\top \mu_1^{\pi_1})$. For any $\boldsymbol{\mu}^{(1)}, \boldsymbol{\mu}^{(2)}, \cdots, \boldsymbol{\mu}^{(T)}, \boldsymbol{\mu^\pi} \in \Pi$ and $\tau \geq 0$, we have

$$
G^{(T),\Pi}(\boldsymbol{\mu^\pi}) := \sum_{t=1}^{T} (F(\boldsymbol{\mu}^{(t)})^\top (\boldsymbol{\mu}^{(t)} - \boldsymbol{\mu^\pi}) + \tau\psi^\Pi(\boldsymbol{\mu}^{(t)}) - \tau\psi^\Pi(\boldsymbol{\mu^\pi}))
$$

$$
= \sum_{s\in\mathcal{S}} \mu^\pi(\sigma(s)) G^{(T)}(s; \pi_{p(s)}(\cdot\,|\,s)) \tag{E.2}
$$

$$
R^{(T),\Pi} := \max_{\boldsymbol{\mu}^{\widehat\pi}\in\Pi} G^{(T),\Pi}(\boldsymbol{\mu}^{\widehat\pi}) \leq \max_{\boldsymbol{\mu}^{\widehat\pi}\in\Pi} \sum_{s\in\mathcal{S}} \mu^{\widehat\pi}(\sigma(s)) R^{(T)}(s) \tag{E.3}
$$

where

$$B_p^{(t)}(s,a) := \sum_{(s,a) \sqsubseteq s'} \frac{\mu_p^{(t)}(\sigma(s'))}{\mu_p^{(t)}(s,a)} \psi_{s'}^\Delta(\pi_p^{(t)}(\cdot \mid s')) \tag{E.4}$$

$$G^{(T)}(s; \pi_{p(s)}(\cdot \mid s)) := \sum_{t=1}^T \Big( \Big\langle -\mathsf{CF}_{p(s)}^{(t)}(s,\cdot) + \tau B_{p(s)}^{(t)}(s,\cdot), \pi_{p(s)}^{(t)}(\cdot \mid s) - \pi_{p(s)}(\cdot \mid s) \Big\rangle$$
$$+ \tau \psi_s^\Delta(\pi_{p(s)}^{(t)}(\cdot \mid s)) - \tau \psi_s^\Delta(\pi_{p(s)}(\cdot \mid s)) \Big) \tag{E.5}$$

$$R^{(T)}(s) := \max_{\widehat{\pi}_{p(s)}(\cdot \mid s) \in \Delta^{|\mathcal{A}_s|}} G^{(T)}(s; \widehat{\pi}_{p(s)}(\cdot \mid s)). \tag{E.6}$$

By using Lemma E.3[4], we have

$$\sum_{s \in \mathcal{S}} \mu^{(\tau,\gamma),*}(\sigma(s)) \sum_{t=1}^T \Big( \tau \mu_{-p(s)}^{(t)}(s) \psi_s^\Delta(\pi_{p(s)}^{(t)}(\cdot \mid s)) - \tau \mu_{-p(s)}^{(t)}(s) \psi_s^\Delta(\pi_{p(s)}^{(\tau,\gamma),*}(\cdot \mid s))$$
$$+ m_s^{(t)} \langle -q^{(t)}(s,\cdot), \pi_{p(s)}^{(t)}(\cdot \mid s) - \pi_{p(s)}^{(\tau,\gamma),*}(\cdot \mid s) \rangle \Big)$$
$$= \sum_{t=1}^T \Big( \Big( \mu_1^{(\tau,\gamma),*} - \mu_1^{(t)} \Big)^\top \boldsymbol{A} \mu_2^{(t)} + \Big( \mu_1^{(t)} \Big)^\top \boldsymbol{A} \Big( \mu_2^{(t)} - \mu_2^{(\tau,\gamma),*} \Big)$$
$$+ \tau \Big( \psi_{\mathrm{bi}}^{\Pi_1}(\mu_1^{(t)}, \mu_2^{(t)}) - \psi_{\mathrm{bi}}^{\Pi_1}(\mu_1^{(\tau,\gamma),*}, \mu_2^{(t)}) - \psi_{\mathrm{bi}}^{\Pi_2}(\mu_1^{(t)}, \mu_2^{(t)}) + \psi_{\mathrm{bi}}^{\Pi_2}(\mu_1^{(\tau,\gamma),*}, \mu_2^{(t)}) \Big)$$
$$+ \tau \Big( \psi_{\mathrm{bi}}^{\Pi_2}(\mu_1^{(t)}, \mu_2^{(t)}) - \psi_{\mathrm{bi}}^{\Pi_2}(\mu_1^{(t)}, \mu_2^{(\tau,\gamma),*}) - \psi_{\mathrm{bi}}^{\Pi_1}(\mu_1^{(t)}, \mu_2^{(t)}) + \psi_{\mathrm{bi}}^{\Pi_1}(\mu_1^{(t)}, \mu_2^{(\tau,\gamma),*}) \Big) \Big)$$
$$= \sum_{t=1}^T \Big( \Big( \mu_1^{(\tau,\gamma),*} - \mu_1^{(t)} \Big)^\top \boldsymbol{A} \mu_2^{(\tau,\gamma),*} + \Big( \mu_1^{(\tau,\gamma),*} \Big)^\top \boldsymbol{A} \Big( \mu_2^{(t)} - \mu_2^{(\tau,\gamma),*} \Big)$$
$$+ \tau \Big( \psi_{\mathrm{bi}}^{\Pi_1}(\mu_1^{(t)}, \mu_2^{(\tau,\gamma),*}) - \psi_{\mathrm{bi}}^{\Pi_1}(\mu_1^{(\tau,\gamma),*}, \mu_2^{(\tau,\gamma),*}) + \psi_{\mathrm{bi}}^{\Pi_2}(\mu_1^{(\tau,\gamma),*}, \mu_1^{(\tau,\gamma),*}) - \psi_{\mathrm{bi}}^{\Pi_2}(\mu_1^{(t)}, \mu_2^{(\tau,\gamma),*}) \Big)$$
$$+ \tau \Big( \psi_{\mathrm{bi}}^{\Pi_2}(\mu_1^{(\tau,\gamma),*}, \mu_2^{(t)}) - \psi_{\mathrm{bi}}^{\Pi_2}(\mu_1^{(\tau,\gamma),*}, \mu_2^{(\tau,\gamma),*}) + \psi_{\mathrm{bi}}^{\Pi_1}(\mu_1^{(\tau,\gamma),*}, \mu_1^{(\tau,\gamma),*}) - \psi_{\mathrm{bi}}^{\Pi_1}(\mu_1^{(\tau,\gamma),*}, \mu_2^{(t)}) \Big) \Big)$$
$$\geq 0.$$

The last inequality is because $\boldsymbol{\mu}^{(\tau,\gamma),*}$ is the NE of $\max_{\mu_1^{\pi_1} \in \Pi_1 : \mu_1^{\pi_1} \succeq \gamma} \min_{\mu_2^{\pi_2} \in \Pi_2 : \mu_2^{\pi_2} \succeq \gamma} (\mu_1^{\pi_1})^\top \boldsymbol{A} \mu_2^{\pi_2} - \tau \psi_{\mathrm{bi}}^{\Pi_1}(\mu_1^{\pi_1}, \mu_2^{\pi_2}) + \tau \psi_{\mathrm{bi}}^{\Pi_2}(\mu_1^{\pi_1}, \mu_2^{\pi_2})$.

Therefore,

$$0 \leq \sum_{s \in \mathcal{S}} \mu^{(\tau,\gamma),*}(\sigma(s)) \sum_{t=1}^T \Big( \tau \mu_{-p(s)}^{(t)}(s) \psi_s^\Delta(\pi_{p(s)}^{(t)}(\cdot \mid s)) - \tau \mu_{-p(s)}^{(t)}(s) \psi_s^\Delta(\pi_{p(s)}^{(\tau,\gamma),*}(\cdot \mid s))$$
$$+ m_s^{(t)} \langle -q^{(t)}(s,\cdot), \pi_{p(s)}^{(t)}(\cdot \mid s) - \pi_{p(s)}^{(\tau,\gamma),*}(\cdot \mid s) \rangle \Big)$$
$$\leq - \frac{\tau \gamma \min_{s \in \mathcal{S}} \sum_{h \in s} \mu_c(h)}{2} \sum_{t=2}^T \sum_{s \in \mathcal{S}} \mu^{(\tau,\gamma),*}(\sigma(s)) D_{\psi_s^\Delta}(\pi_{p(s)}^{(\tau,\gamma),*}(\cdot \mid s), \bar{\pi}_{p(s)}^{(t)}(\cdot \mid s))$$
$$+ \sum_{s \in \mathcal{S}} \Big( C_s' + C_s^{-,Q} \Big) \tau \eta_s^{\mathrm{anc}} \mu^{(\tau,\gamma),*}(\sigma(s)) \sum_{t=1}^T \Big| \psi_s^\Delta(\pi_{p(s)}^{(t)}(\cdot \mid s)) - \psi_s^\Delta(\bar{\pi}_{p(s)}^{(t+1)}(\cdot \mid s)) \Big|$$
$$+ 2 \sum_{s \in \mathcal{S}} C_s^{\mathrm{diff}} \mu^{(\tau,\gamma),*}(\sigma(s)) \|\boldsymbol{q}\|_\infty \eta_s M_2 T$$
$$+ \sum_{s \in \mathcal{S}} \frac{m_s^{(1)}}{\eta_s} \mu^{(\tau,\gamma),*}(\sigma(s)) D_{\psi_s^\Delta}(\pi_{p(s)}^{(\tau,\gamma),*}(\cdot \mid s), \bar{\pi}_{p(s)}^{(1)}(\cdot \mid s)).$$

---

[4]It can be easily generalized to bidilated version by absorbing the reach probability of player $3 - p$ and the chance player into $\psi_s^\Delta$ for $\psi_{\mathrm{bi}}^{\Pi_p}$ and player $p$. For $\psi_{\mathrm{bi}}^{\Pi_{3-p}}$, since it is linear with respect to $\mu_p^{\pi_p}$, it can be combined into the counterfactual value.

$(i)$ is because $\boldsymbol{\mu}^{(\tau,\gamma),*}$ is the NE of the regularized and perturbed EFG. Then, by rearranging the terms, we have

$$\sum_{t=2}^{T} D_{\psi^{\Pi}}(\boldsymbol{\mu}^{(\tau,\gamma),*}, \boldsymbol{\mu}^{\overline{\boldsymbol{\pi}}^{(t)}})$$

$$=\sum_{t=2}^{T}\sum_{s\in\mathcal{S}}\mu^{(\tau,\gamma),*}(\sigma(s))D_{\psi_s^{\Delta}}(\pi_{p(s)}^{(\tau,\gamma),*}(\cdot\,|\,s),\overline{\pi}_{p(s)}^{(t)}(\cdot\,|\,s))$$

$$\leq\frac{2}{\gamma\min_{s\in\mathcal{S}}\sum_{h\in s}\mu_c(h)}\sum_{s\in\mathcal{S}}\left(C_s' + C_s^{-,Q}\right)\eta_s^{\mathrm{anc}}\mu^{(\tau,\gamma),*}(\sigma(s))\sum_{t=1}^{T}\left|\psi_s^{\Delta}(\pi_{p(s)}^{(t)}(\cdot\,|\,s)) - \psi_s^{\Delta}(\overline{\pi}_{p(s)}^{(t+1)}(\cdot\,|\,s))\right|$$

$$+\frac{4}{\tau\gamma\min_{s\in\mathcal{S}}\sum_{h\in s}\mu_c(h)}\sum_{s\in\mathcal{S}}C_s^{\mathrm{diff}}\mu^{(\tau,\gamma),*}(\sigma(s))\,\|\boldsymbol{q}\|_{\infty}\,\eta_s M_2 T$$

$$+\frac{2}{\tau\gamma\min_{s\in\mathcal{S}}\sum_{h\in s}\mu_c(h)}\sum_{s\in\mathcal{S}}\frac{m_s^{(1)}}{\eta_s}\mu^{(\tau,\gamma),*}(\sigma(s))D_{\psi_s^{\Delta}}(\pi_{p(s)}^{(\tau,\gamma),*}(\cdot\,|\,s),\overline{\pi}_{p(s)}^{(1)}(\cdot\,|\,s)).$$

The first equality is by Lemma 4.1. Now, we achieved best-iterate convergence to the regularized NE $\boldsymbol{\mu}^{(\tau,\gamma),*}$ in terms of Bregman divergence. $\qquad\square$

### E.1 PROOF OF LEMMA 4.1

By definition of Bregman divergence, we have

$$D_{\psi^{\Pi}}(\boldsymbol{\mu}^{\boldsymbol{\pi}}, \boldsymbol{\mu}^{\widetilde{\boldsymbol{\pi}}}) = \psi^{\Pi}(\boldsymbol{\mu}^{\boldsymbol{\pi}}) - \psi^{\Pi}(\boldsymbol{\mu}^{\widetilde{\boldsymbol{\pi}}}) - \left\langle \nabla\psi^{\Pi}(\boldsymbol{\mu}^{\widetilde{\boldsymbol{\pi}}}), \boldsymbol{\mu}^{\boldsymbol{\pi}} - \boldsymbol{\mu}^{\widetilde{\boldsymbol{\pi}}} \right\rangle.$$

For notational simplicity, we use $\mu^{\boldsymbol{\pi}}(s,a)$ as $\mu_{p(s)}^{\boldsymbol{\pi}}(s,a)$. Since $\psi^{\Pi}(\boldsymbol{\mu}^{\widetilde{\boldsymbol{\pi}}}) = \sum_{s\in\mathcal{S}}\mu^{\widetilde{\boldsymbol{\pi}}}(\sigma(s))\psi_s^{\Delta}\left(\frac{\mu^{\widetilde{\boldsymbol{\pi}}}(s,\cdot)}{\mu^{\widetilde{\boldsymbol{\pi}}}(\sigma(s))}\right)$, for any $s\in\mathcal{S}, a\in\mathcal{A}_s$, the gradient $\nabla_{\mu^{\widetilde{\boldsymbol{\pi}}}(s,a)}\psi^{\Pi}(\boldsymbol{\mu}^{\widetilde{\boldsymbol{\pi}}})$ is equal to

$$\nabla_{\mu^{\widetilde{\boldsymbol{\pi}}}(s,a)}\psi^{\Pi}(\boldsymbol{\mu}^{\widetilde{\boldsymbol{\pi}}}) = \sum_{s'\in\mathcal{S}:\,\sigma(s')=(s,a)}\left(\psi_{s'}^{\Delta}\left(\frac{\mu^{\widetilde{\boldsymbol{\pi}}}(s',\cdot)}{\mu^{\widetilde{\boldsymbol{\pi}}}(\sigma(s'))}\right) - \left\langle\nabla\psi_{s'}^{\Delta}\left(\frac{\mu^{\widetilde{\boldsymbol{\pi}}}(s',\cdot)}{\mu^{\widetilde{\boldsymbol{\pi}}}(\sigma(s'))}\right), \frac{\mu^{\widetilde{\boldsymbol{\pi}}}(s',\cdot)}{\mu^{\widetilde{\boldsymbol{\pi}}}(\sigma(s'))}\right\rangle\right)$$
$$+ \nabla_a\psi_s^{\Delta}(\frac{\mu^{\widetilde{\boldsymbol{\pi}}}(s,\cdot)}{\mu^{\widetilde{\boldsymbol{\pi}}}(\sigma(s))}).$$

Therefore,

$$\left\langle\nabla\psi^{\Pi}(\boldsymbol{\mu}^{\widetilde{\boldsymbol{\pi}}}), \boldsymbol{\mu}^{\widetilde{\boldsymbol{\pi}}}\right\rangle$$

$$=\sum_{s\in\mathcal{S},a\in\mathcal{A}_s}\mu^{\widetilde{\boldsymbol{\pi}}}(s,a)\sum_{s'\in\mathcal{S}:\,\sigma(s')=(s,a)}\left(\psi_{s'}^{\Delta}\left(\frac{\mu^{\widetilde{\boldsymbol{\pi}}}(s',\cdot)}{\mu^{\widetilde{\boldsymbol{\pi}}}(\sigma(s'))}\right) - \left\langle\nabla\psi_{s'}^{\Delta}\left(\frac{\mu^{\widetilde{\boldsymbol{\pi}}}(s',\cdot)}{\mu^{\widetilde{\boldsymbol{\pi}}}(\sigma(s'))}\right), \frac{\mu^{\widetilde{\boldsymbol{\pi}}}(s',\cdot)}{\mu^{\widetilde{\boldsymbol{\pi}}}(\sigma(s'))}\right\rangle\right)$$

$$+\sum_{s\in\mathcal{S},a\in\mathcal{A}_s}\mu^{\widetilde{\boldsymbol{\pi}}}(s,a)\nabla_a\psi_s^{\Delta}(\frac{\mu^{\widetilde{\boldsymbol{\pi}}}(s,\cdot)}{\mu^{\widetilde{\boldsymbol{\pi}}}(\sigma(s))})$$

$$=\sum_{s\in\mathcal{S},a\in\mathcal{A}_s}\sum_{s'\in\mathcal{S}:\,\sigma(s')=(s,a)}\mu^{\widetilde{\boldsymbol{\pi}}}(\sigma(s'))\psi_{s'}^{\Delta}\left(\frac{\mu^{\widetilde{\boldsymbol{\pi}}}(s',\cdot)}{\mu^{\widetilde{\boldsymbol{\pi}}}(\sigma(s'))}\right) \qquad\text{(E.7)}$$

$$-\sum_{s\in\mathcal{S},a\in\mathcal{A}_s}\sum_{s'\in\mathcal{S}:\,\sigma(s')=(s,a)}\left\langle\nabla\psi_{s'}^{\Delta}\left(\frac{\mu^{\widetilde{\boldsymbol{\pi}}}(s',\cdot)}{\mu^{\widetilde{\boldsymbol{\pi}}}(\sigma(s'))}\right), \mu^{\widetilde{\boldsymbol{\pi}}}(s',\cdot)\right\rangle \qquad\text{(E.8)}$$

$$+\sum_{s\in\mathcal{S},a\in\mathcal{A}_s}\mu^{\widetilde{\boldsymbol{\pi}}}(s,a)\nabla_a\psi_s^{\Delta}(\frac{\mu^{\widetilde{\boldsymbol{\pi}}}(s,\cdot)}{\mu^{\widetilde{\boldsymbol{\pi}}}(\sigma(s))}). \qquad\text{(E.9)}$$

Note that (E.7) is equal to $\sum_{s\in\mathcal{S}}\mu^{\widetilde{\boldsymbol{\pi}}}(\sigma(s))\psi_s^{\Delta}\left(\frac{\mu^{\widetilde{\boldsymbol{\pi}}}(s,\cdot)}{\mu^{\widetilde{\boldsymbol{\pi}}}(\sigma(s))}\right) = \psi^{\Pi}(\boldsymbol{\mu}^{\widetilde{\boldsymbol{\pi}}})$ due to the uniqueness of $\sigma(s')$.

Similarly, due to the uniqueness of $\sigma(s')$, (E.8) is equal to $-\sum_{s \in \mathcal{S}} \left\langle \nabla \psi_s^{\Delta} \left( \frac{\mu^{\widetilde{\pi}}(s,\cdot)}{\mu^{\widetilde{\pi}}(\sigma(s))} \right), \mu^{\widetilde{\pi}}(s,\cdot) \right\rangle$, which is equal to the negative of (E.9) and thus cancel out. Therefore,

$$\left\langle \nabla \psi^{\Pi}(\boldsymbol{\mu}^{\widetilde{\boldsymbol{\pi}}}), \boldsymbol{\mu}^{\widetilde{\boldsymbol{\pi}}} \right\rangle = \psi^{\Pi}(\boldsymbol{\mu}^{\widetilde{\boldsymbol{\pi}}}).$$

Moreover,

$$\left\langle \nabla \psi^{\Pi}(\boldsymbol{\mu}^{\widetilde{\boldsymbol{\pi}}}), \boldsymbol{\mu}^{\boldsymbol{\pi}} \right\rangle$$

$$= \sum_{s \in \mathcal{S}, a \in \mathcal{A}_s} \sum_{s' \in \mathcal{S}:\, \sigma(s')=(s,a)} \mu^{\boldsymbol{\pi}}(\sigma(s')) \psi_{s'}^{\Delta} \left( \frac{\mu^{\widetilde{\pi}}(s',\cdot)}{\mu^{\widetilde{\pi}}(\sigma(s'))} \right)$$

$$- \sum_{s \in \mathcal{S}, a \in \mathcal{A}_s} \mu^{\boldsymbol{\pi}}(s,a) \sum_{s' \in \mathcal{S}:\, \sigma(s')=(s,a)} \left\langle \nabla \psi_{s'}^{\Delta} \left( \frac{\mu^{\widetilde{\pi}}(s',\cdot)}{\mu^{\widetilde{\pi}}(\sigma(s'))} \right), \frac{\mu^{\widetilde{\pi}}(s',\cdot)}{\mu^{\widetilde{\pi}}(\sigma(s'))} \right\rangle$$

$$+ \sum_{s \in \mathcal{S}, a \in \mathcal{A}_s} \mu^{\boldsymbol{\pi}}(s,a) \nabla_a \psi_s^{\Delta}(\frac{\mu^{\widetilde{\pi}}(s,\cdot)}{\mu^{\widetilde{\pi}}(s,a)})$$

$$= \sum_{s \in \mathcal{S}} \mu^{\boldsymbol{\pi}}(\sigma(s)) \left( \psi_s^{\Delta} \left( \frac{\mu^{\widetilde{\pi}}(s,\cdot)}{\mu^{\widetilde{\pi}}(\sigma(s))} \right) + \left\langle \nabla \psi_s^{\Delta} \left( \frac{\mu^{\widetilde{\pi}}(s,\cdot)}{\mu^{\widetilde{\pi}}(\sigma(s))} \right), \frac{\mu^{\boldsymbol{\pi}}(s,\cdot)}{\mu^{\boldsymbol{\pi}}(\sigma(s))} - \frac{\mu^{\widetilde{\pi}}(s,\cdot)}{\mu^{\widetilde{\pi}}(\sigma(s))} \right\rangle \right).$$

Therefore,

$$\left\langle F^{\tau}(\boldsymbol{\mu}^{\boldsymbol{\pi}}), \boldsymbol{\mu}^{\boldsymbol{\pi}} - \boldsymbol{\mu}^{\boldsymbol{\pi}'} \right\rangle$$

$$= - (\mu_1^{\pi_1})^{\top} \boldsymbol{A} \mu_2^{\pi_2'} + (\mu_1^{\pi_1'})^{\top} \boldsymbol{A} \mu_2^{\pi_2} + \tau D_{\psi_{\mathrm{bi}}^{\Pi_1}(\cdot,\mu_2^{\pi_2})} \left( \mu_1^{\pi_1'}, \mu_1^{\pi_1} \right) + \tau D_{\psi_{\mathrm{bi}}^{\Pi_2}(\mu_1^{\pi_1},\cdot)} \left( \mu_2^{\pi_2'}, \mu_2^{\pi_2} \right)$$

$$+ \tau \psi_{\mathrm{bi}}^{\Pi_1}(\mu_1^{\pi_1}, \mu_2^{\pi_2'}) - \tau \psi_{\mathrm{bi}}^{\Pi_1}(\mu_1^{\pi_1'}, \mu_2^{\pi_2}) + \tau \psi_{\mathrm{bi}}^{\Pi_2}(\mu_1^{\pi_1'}, \mu_2^{\pi_2}) - \tau \psi_{\mathrm{bi}}^{\Pi_2}(\mu_1^{\pi_1}, \mu_2^{\pi_2'}).$$

### E.2 PROOF OF LEMMA E.2

Firstly, we introduce the following lemma.

**Lemma E.4.** Let $\mathcal{C}$ be a convex set and $\boldsymbol{x}^{(1)} = \mathrm{argmin}_{\boldsymbol{x} \in \mathcal{C}} \left\{ \langle \boldsymbol{g}, \boldsymbol{x} \rangle + \tau_0 \psi^{\mathcal{C}}(\boldsymbol{x}) + \frac{1}{\eta} D_{\psi^{\mathcal{C}}}(\boldsymbol{x}, \boldsymbol{x}^{(0)}) \right\}$, where $\psi^{\mathcal{C}}$ is a strongly-convex function in $\mathcal{C}$ and $\tau_0 \geq 0$ is a constant. Then, for any $\boldsymbol{x}^{(2)} \in \mathcal{C}$, we have

$$\eta \tau_0 \psi^{\mathcal{C}}(\boldsymbol{x}^{(1)}) - \eta \tau_0 \psi^{\mathcal{C}}(\boldsymbol{x}^{(2)}) + \eta \left\langle \boldsymbol{g}, \boldsymbol{x}^{(1)} - \boldsymbol{x}^{(2)} \right\rangle \tag{E.10}$$

$$\leq D_{\psi^{\mathcal{C}}}(\boldsymbol{x}^{(2)}, \boldsymbol{x}^{(0)}) - (1 + \eta \tau_0) D_{\psi^{\mathcal{C}}}(\boldsymbol{x}^{(2)}, \boldsymbol{x}^{(1)}) - D_{\psi^{\mathcal{C}}}(\boldsymbol{x}^{(1)}, \boldsymbol{x}^{(0)}).$$

The proof is postponed to the end of this section.

Plug $\boldsymbol{x}^{(0)} = \bar{\pi}_{p(s)}^{(t)}(\cdot \mid s), \boldsymbol{x}^{(1)} = \bar{\pi}_{p(s)}^{(t+1)}(\cdot \mid s), \boldsymbol{x}^{(2)} = \pi_{p(s)}(\cdot \mid s), \boldsymbol{g} = -q^{(t)}(s,\cdot), \psi^{\mathcal{C}} = \psi_s^{\Delta}, \eta = \eta_s, \tau_0 = \frac{\tau \mu_{-p(s)}^{(t)}(s)}{m_s^{(t)}}$ into Lemma E.4, with $\mathcal{C} = \Delta_{\gamma_s,\boldsymbol{\nu}_s}^{|\mathcal{A}_s|}$,

$$\eta_s \frac{\tau \mu_{-p(s)}^{(t)}(s)}{m_s^{(t)}} \psi_s^{\Delta}(\bar{\pi}_{p(s)}^{(t+1)}(\cdot \mid s)) - \eta_s \frac{\tau \mu_{-p(s)}^{(t)}(s)}{m_s^{(t)}} \psi_s^{\Delta}(\pi_{p(s)}(\cdot \mid s))$$

$$+ \eta_s \langle \bar{\pi}_{p(s)}^{(t+1)}(\cdot \mid s) - \pi_{p(s)}(\cdot \mid s), -q^{(t)}(s,\cdot) \rangle$$

$$\leq D_{\psi_s^{\Delta}}(\pi_{p(s)}(\cdot \mid s), \bar{\pi}_{p(s)}^{(t)}(\cdot \mid s)) - \left( 1 + \eta_s \frac{\tau \mu_{-p(s)}^{(t)}(s)}{m_s^{(t)}} \right) D_{\psi_s^{\Delta}}(\pi_{p(s)}(\cdot \mid s), \bar{\pi}_{p(s)}^{(t+1)}(\cdot \mid s))$$

$$- D_{\psi_s^{\Delta}}(\bar{\pi}_{p(s)}^{(t+1)}(\cdot \mid s), \bar{\pi}_{p(s)}^{(t)}(\cdot \mid s)).$$

Plug $\boldsymbol{x}^{(0)} = \bar{\pi}^{(t)}_{p(s)}(\cdot\,|\,s), \boldsymbol{x}^{(1)} = \pi^{(t)}_{p(s)}(\cdot\,|\,s), \boldsymbol{x}^{(2)} = \bar{\pi}^{(t+1)}_{p(s)}(\cdot\,|\,s), \boldsymbol{g} = -q^{(t-1)}(s,\cdot), \psi^{\mathcal{C}} = \psi^{\Delta}_s, \eta = \eta_s, \tau_0 = \frac{\tau\mu^{(t-1)}_{-p(s)}(s)}{m^{(t-1)}_s}$ into Lemma E.4, with $\mathcal{C} = \Delta^{|\mathcal{A}_s|}_{\gamma_s,\boldsymbol{\nu}_s}$,

$$\eta_s \frac{\tau\mu^{(t-1)}_{-p(s)}(s)}{m^{(t-1)}_s}\psi^{\Delta}_s(\pi^{(t)}_{p(s)}(\cdot\,|\,s)) - \eta_s \frac{\tau\mu^{(t-1)}_{-p(s)}(s)}{m^{(t-1)}_s}\psi^{\Delta}_s(\bar{\pi}^{(t+1)}_{p(s)}(\cdot\,|\,s))$$
$$+ \eta_s\langle\pi^{(t)}_{p(s)}(\cdot\,|\,s) - \bar{\pi}^{(t+1)}_{p(s)}(\cdot\,|\,s), -q^{(t-1)}(s,\cdot)\rangle$$
$$\leq D_{\psi^{\Delta}_s}(\bar{\pi}^{(t+1)}_{p(s)}(\cdot\,|\,s), \bar{\pi}^{(t)}_{p(s)}(\cdot\,|\,s)) - (1 + \eta_s\frac{\tau\mu^{(t-1)}_{-p(s)}(s)}{m^{(t-1)}_s})D_{\psi^{\Delta}_s}(\bar{\pi}^{(t+1)}_{p(s)}(\cdot\,|\,s), \pi^{(t)}_{p(s)}(\cdot\,|\,s))$$
$$- D_{\psi^{\Delta}_s}(\pi^{(t)}_{p(s)}(\cdot\,|\,s), \bar{\pi}^{(t)}_{p(s)}(\cdot\,|\,s)).$$

Summing them up and adding $\eta_s\left\langle q^{(t-1)}(s,\cdot) - q^{(t)}(s,\cdot), \pi^{(t)}_{p(s)}(\cdot\,|\,s) - \bar{\pi}^{(t+1)}_{p(s)}(\cdot\,|\,s)\right\rangle$ to both sides, we have

$$\eta_s\frac{\tau\mu^{(t)}_{-p(s)}(s)}{m^{(t)}_s}\psi^{\Delta}_s(\pi^{(t)}_{p(s)}(\cdot\,|\,s)) - \eta_s\frac{\tau\mu^{(t)}_{-p(s)}(s)}{m^{(t)}_s}\psi^{\Delta}_s(\pi_{p(s)}(\cdot\,|\,s)) \qquad\text{(E.11)}$$
$$+ \eta_s\tau\big(\frac{\mu^{(t-1)}_{-p(s)}(s)}{m^{(t-1)}_s} - \frac{\mu^{(t)}_{-p(s)}(s)}{m^{(t)}_s}\big)(\psi^{\Delta}_s(\pi^{(t)}_{p(s)}(\cdot\,|\,s)) - \psi^{\Delta}_s(\bar{\pi}^{(t+1)}_{p(s)}(\cdot\,|\,s)))$$
$$+ \eta_s\langle-q^{(t)}(s,\cdot), \pi^{(t)}_{p(s)}(\cdot\,|\,s) - \pi_{p(s)}(\cdot\,|\,s)\rangle$$
$$\leq D_{\psi^{\Delta}_s}(\pi_{p(s)}(\cdot\,|\,s), \bar{\pi}^{(t)}_{p(s)}(\cdot\,|\,s)) - (1 + \eta_s\frac{\tau\mu^{(t)}_{-p(s)}(s)}{m^{(t)}_s})D_{\psi^{\Delta}_s}(\pi_{p(s)}(\cdot\,|\,s), \bar{\pi}^{(t+1)}_{p(s)}(\cdot\,|\,s))$$
$$- (1 + \eta_s\frac{\tau\mu^{(t-1)}_{-p(s)}(s)}{m^{(t-1)}_s})D_{\psi^{\Delta}_s}(\bar{\pi}^{(t+1)}_{p(s)}(\cdot\,|\,s), \pi^{(t)}_{p(s)}(\cdot\,|\,s))$$
$$- D_{\psi^{\Delta}_s}(\pi^{(t)}_{p(s)}(\cdot\,|\,s), \bar{\pi}^{(t)}_{p(s)}(\cdot\,|\,s)) + \eta_s\left\langle q^{(t-1)}(s,\cdot) - q^{(t)}(s,\cdot), \pi^{(t)}_{p(s)}(\cdot\,|\,s) - \bar{\pi}^{(t+1)}_{p(s)}(\cdot\,|\,s)\right\rangle.$$
$$\square$$

*Proof of Lemma E.4.*

$$D_{\psi^{\mathcal{C}}}(\boldsymbol{x}^{(2)}, \boldsymbol{x}^{(0)}) - (1 + \eta\tau_0)D_{\psi^{\mathcal{C}}}(\boldsymbol{x}^{(2)}, \boldsymbol{x}^{(1)}) - D_{\psi^{\mathcal{C}}}(\boldsymbol{x}^{(1)}, \boldsymbol{x}^{(0)})$$
$$= \left(\psi^{\mathcal{C}}(\boldsymbol{x}^{(2)}) - \psi^{\mathcal{C}}(\boldsymbol{x}^{(0)}) - \left\langle\nabla\psi^{\mathcal{C}}(\boldsymbol{x}^{(0)}), \boldsymbol{x}^{(2)} - \boldsymbol{x}^{(0)}\right\rangle\right)$$
$$- (1 + \eta\tau_0)\left(\psi^{\mathcal{C}}(\boldsymbol{x}^{(2)}) - \psi^{\mathcal{C}}(\boldsymbol{x}^{(1)}) - \left\langle\nabla\psi^{\mathcal{C}}(\boldsymbol{x}^{(1)}), \boldsymbol{x}^{(2)} - \boldsymbol{x}^{(1)}\right\rangle\right)$$
$$- \left(\psi^{\mathcal{C}}(\boldsymbol{x}^{(1)}) - \psi^{\mathcal{C}}(\boldsymbol{x}^{(0)}) - \left\langle\nabla\psi^{\mathcal{C}}(\boldsymbol{x}^{(0)}), \boldsymbol{x}^{(1)} - \boldsymbol{x}^{(0)}\right\rangle\right)$$
$$= \eta\tau_0\psi^{\mathcal{C}}(\boldsymbol{x}^{(1)}) - \eta\tau_0\psi^{\mathcal{C}}(\boldsymbol{x}^{(2)}) + \left\langle(1 + \eta\tau_0)\nabla\psi^{\mathcal{C}}(\boldsymbol{x}^{(1)}) - \nabla\psi^{\mathcal{C}}(\boldsymbol{x}^{(0)}), \boldsymbol{x}^{(2)} - \boldsymbol{x}^{(1)}\right\rangle.$$

Since

$$\boldsymbol{x}^{(1)} = \operatorname*{argmin}_{\boldsymbol{x}\in\mathcal{C}}\left\{\langle\boldsymbol{g}, \boldsymbol{x}\rangle + \tau_0\psi^{\mathcal{C}}(\boldsymbol{x}) + \frac{1}{\eta}\left(\psi^{\mathcal{C}}(\boldsymbol{x}) - \psi^{\mathcal{C}}(\boldsymbol{x}^{(0)}) - \left\langle\nabla\psi^{\mathcal{C}}(\boldsymbol{x}^{(0)}), \boldsymbol{x} - \boldsymbol{x}^{(0)}\right\rangle\right)\right\},$$

by first-order optimality, we have,

$$\left\langle\eta\boldsymbol{g} + (1 + \eta\tau_0)\nabla\psi^{\mathcal{C}}(\boldsymbol{x}^{(1)}) - \nabla\psi^{\mathcal{C}}(\boldsymbol{x}^{(0)}), \boldsymbol{x}^{(2)} - \boldsymbol{x}^{(1)}\right\rangle \geq 0.$$

Therefore,

$$D_{\psi^{\mathcal{C}}}(\boldsymbol{x}^{(2)}, \boldsymbol{x}^{(0)}) - (1 + \eta\tau_0)D_{\psi^{\mathcal{C}}}(\boldsymbol{x}^{(2)}, \boldsymbol{x}^{(1)}) - D_{\psi^{\mathcal{C}}}(\boldsymbol{x}^{(1)}, \boldsymbol{x}^{(0)})$$
$$\geq \eta\tau_0\psi^{\mathcal{C}}(\boldsymbol{x}^{(1)}) - \eta\tau_0\psi^{\mathcal{C}}(\boldsymbol{x}^{(2)}) + \eta\left\langle\boldsymbol{g}, \boldsymbol{x}^{(1)} - \boldsymbol{x}^{(2)}\right\rangle.$$
$$\square$$

# F  PROOF OF THEOREM 3.1

**Theorem F.1** (Formal Version of Theorem 3.1). Consider Algorithm 1. When $\frac{\eta_s^{\mathrm{anc}}}{\eta_s} \leq \tau C_s^{\eta,T}$ for any $s \in \mathcal{S}$, where $C_s^{\eta,T} := \frac{\gamma^2 \sum_{h \in s} \mu_c(h)}{2 C_s^- \left( \log T + \log |\mathcal{S}| + \log \frac{1}{\delta} \right)}$, and (**A**), (**B**), (**C**) are satisfied, we have the following guarantee with probability $1 - 2\delta$.

$$
\sum_{t=2}^{T} D_{\psi^{\Pi}}(\boldsymbol{\mu}^{(\tau,\gamma),*}, \boldsymbol{\mu}^{\bar{\boldsymbol{\pi}}^{(t)}})
$$

$$
\leq \frac{4 C_{\mathrm{visit}}}{\tau} \sum_{s \in \mathcal{S}} C_s^{\mathrm{diff}} \mu^{(\tau,\gamma),*}(\sigma(s)) \|\boldsymbol{q}\|_\infty \eta_s M_2 T
$$

$$
+ \frac{2 C_{\mathrm{visit}}}{\tau} \sum_{s \in \mathcal{S}} \frac{1}{\eta_s \mu_{p(s)}^{(t_s^1)}(\sigma(s))} \mu^{(\tau,\gamma),*}(\sigma(s)) \max_{\boldsymbol{x},\boldsymbol{y} \in \Delta_{\gamma_s,\boldsymbol{\nu}_s}^{|\mathcal{A}_s|}} D_{\psi_s^{\Delta}}(\boldsymbol{x},\boldsymbol{y})
$$

$$
+ \frac{4 C_{\mathrm{visit}}^2}{\tau} \sum_{s \in \mathcal{S}} \frac{1}{\eta_s} \mu^{(\tau,\gamma),*}(\sigma(s)) \max_{\boldsymbol{x},\boldsymbol{y} \in \Delta_{\gamma_s,\boldsymbol{\nu}_s}^{|\mathcal{A}_s|}} D_{\psi_s^{\Delta}}(\boldsymbol{x},\boldsymbol{y}) \qquad \text{(F.1)}
$$

$$
+ \frac{4 C_{\mathrm{visit}}}{\tau} \left( 1 + \frac{1}{\gamma} \right) (\|\boldsymbol{q}\|_\infty + \tau \psi^{\mathrm{max}}) \sqrt{2 T \log \frac{|\mathcal{S}|}{\delta}} + \sum_{s \in \mathcal{S}} \frac{4 C_{\mathrm{visit}}^2}{\eta_s \tau} \max_{\boldsymbol{x},\boldsymbol{y} \in \Delta_{\gamma_s,\boldsymbol{\nu}_s}^{|\mathcal{A}_s|}} D_{\psi_s^{\Delta}}(\boldsymbol{x},\boldsymbol{y}),
$$

where $C_{\mathrm{visit}} := \frac{\log T + \log |\mathcal{S}| + \log \frac{1}{\delta}}{\gamma^2 \min_{s \in \mathcal{S}} \sum_{h \in s} \mu_c(h)}$ is the maximum gap between any two adjacent visits to an infoset.

**Proof Sketch.** (i). Firstly, we show that the estimates in Algorithm 1 are unbiased so that the conditions of Lemma 4.3 are met. (ii). By Lemma 4.3 and union bound, we can extend the result of full information feedback to the stochastic feedback. (iii). To ensure the coefficient for the cumulated distance to NE after telescoping is still positive, we need to bound the largest gap between the timesteps of two consecutive visits of any infoset.

For any infoset $s \in \mathcal{S}$, we define $T_s := \{t_s^1, t_s^2, \cdots\}$, where each $t_s^k \in [T]$ is the timestep that $s$ is along the sampled trajectory. Then, we will show Algorithm 1 uses unbiased estimators so that we can derive an upper-bound by Lemma 4.3. Note that for any $\boldsymbol{u} \in \Delta_{\gamma_s,\boldsymbol{\nu}_s}^{|\mathcal{A}_s|}$, the expectation of the additional regularizer term is,

$$
\Pr\left(t \in T_s\right) \left( \frac{\tau}{\mu_{p(s)}^{(t)}(\sigma(s))} \psi_s^{\Delta}(\boldsymbol{u}) \right) = \mu_{p(s)}^{(t)}(\sigma(s)) \mu_{-p(s)}^{(t)}(s) \left( \frac{\tau}{\mu_{p(s)}^{(t)}(\sigma(s))} \psi_s^{\Delta}(\boldsymbol{u}) \right)
$$

$$
= \tau \mu_{-p(s)}^{(t)}(s) \psi_s^{\Delta}(\boldsymbol{u}).
$$

Let $s(h)$ denote the infoset that the node $h$ is in. For the original utility, suppose $p(s) = 1$ without loss of generality, the expectation of $\widetilde{q}^{(t)}(s,a)$ for any $a \in \mathcal{A}_s$ is,

$$
\frac{1}{\pi_1^{(t)}(a \mid s)} \sum_{h' \in \mathcal{H} : \, \exists h \in s, (h,a) \sqsubseteq h'} \mu_1^{(t)}(\sigma_1(h')) \mu_2^{(t)}(\sigma_2(h')) \mu_c(h') \mathcal{U}_1(h')
$$

$$
- \frac{1}{\pi_1^{(t)}(a \mid s)} \sum_{h' \in \mathcal{H}_1 : \, \exists h \in s, (h,a) \sqsubseteq h'} \mu_1^{(t)}(\sigma_1(h')) \mu_2^{(t)}(\sigma_2(h')) \mu_c(h') \psi_{s(h)}^{\Delta}(\pi_1^{(t)})
$$

$$
+ \frac{1}{\pi_1^{(t)}(a \mid s)} \sum_{h' \in \mathcal{H}_2 : \, \exists h \in s, (h,a) \sqsubseteq h'} \mu_1^{(t)}(\sigma_1(h')) \mu_2^{(t)}(\sigma_2(h')) \mu_c(h') \psi_{s(h)}^{\Delta}(\pi_2^{(t)}),
$$

which is equal to $q^{(t)}(s,a)$ by definition.

By Lemma 4.3 and union bound, with probability at least $1 - \delta$, the following is satisfied for all infosets $s \in \mathcal{S}$,

$$
\Pr\left(t \in T_s\right) \left( \frac{\tau}{\mu_{p(s)}^{(t)}(\sigma(s))} \psi_s^{\Delta}(\boldsymbol{u}) \right) = \mu_{p(s)}^{(t)}(\sigma(s)) \mu_{-p(s)}^{(t)}(s) \left( \frac{\tau}{\mu_{p(s)}^{(t)}(\sigma(s))} \psi_s^{\Delta}(\boldsymbol{u}) \right) = \tau \mu_{-p(s)}^{(t)}(s) \psi_s^{\Delta}(\boldsymbol{u}).
$$

where $\psi^{\max}$ is the upperbound of $\psi_s^\Delta$, which is $\max_{s\in\mathcal{S}}\frac{\alpha_s}{2}$ when it is Euclidean norm and $\max_{s\in\mathcal{S}}\alpha_s\log|\mathcal{A}_s|$ when it is entropy.

Let's define $t_s^0 = 1$ for notational simplicity. Similar to the proof of Theorem 4.2, for each $k \leq |T_s| - 1$, we have

$$
\eta_s\tau\psi_s^\Delta(\pi_{p(s)}^{(t_s^k)}(\cdot\,|\,s)) - \eta_s\tau\psi_s^\Delta(\pi_{p(s)}(\cdot\,|\,s)) + \eta_s\left\langle -\widetilde{q}^{(t_s^k)}(s,\cdot), \pi_{p(s)}^{(t_s^k)}(\cdot\,|\,s) - \pi_{p(s)}(\cdot\,|\,s) \right\rangle
$$

$$
\leq D_{\psi_s^\Delta}(\pi_{p(s)}(\cdot\,|\,s), \overline{\pi}_{p(s)}^{(t_s^k)}(\cdot\,|\,s)) - (1 + \eta_s\tau)D_{\psi_s^\Delta}(\pi_{p(s)}(\cdot\,|\,s), \overline{\pi}_{p(s)}^{(t_s^{k+1})}(\cdot\,|\,s))
$$

$$
- (1 + \eta_s\tau)D_{\psi_s^\Delta}(\overline{\pi}_{p(s)}^{(t_s^{k+1})}(\cdot\,|\,s), \pi_{p(s)}^{(t_s^k)}(\cdot\,|\,s))
$$

$$
+ \eta_s\left\langle \widetilde{q}^{(t_s^{k-1})}(s,\cdot) - \widetilde{q}^{(t_s^k)}(s,\cdot), \pi_{p(s)}^{(t_s^k)}(\cdot\,|\,s) - \overline{\pi}_{p(s)}^{(t_s^{k+1})}(\cdot\,|\,s) \right\rangle.
$$

By multiplying $m_s^{(t_s^k)} = \frac{1}{\mu_{p(s)}^{(t_s^k)}(\sigma(s))}$ on both sides and telescoping, we have

$$
\sum_{k=1}^{|T_s|-1}\left( \frac{\tau}{\mu_{p(s)}^{(t_s^k)}(\sigma(s))}\left(\psi_s^\Delta(\pi_{p(s)}^{(t_s^k)}(\cdot\,|\,s)) - \psi_s^\Delta(\pi_{p(s)}(\cdot\,|\,s))\right)\right.
$$

$$
\left. + \frac{1}{\mu_{p(s)}^{(t_s^k)}(\sigma(s))}\langle -\widetilde{q}^{(t_s^k)}(s,\cdot), \pi_{p(s)}^{(t_s^k)}(\cdot\,|\,s) - \pi_{p(s)}(\cdot\,|\,s)\rangle\right)
$$

$$
\leq \sum_{k=2}^{|T_s|}\left( \frac{1}{\mu_{p(s)}^{(t_s^k)}(\sigma(s))} - \frac{1}{\mu_{p(s)}^{(t_s^{k-1})}(\sigma(s))} - \frac{\eta_s\tau}{\mu_{p(s)}^{(t_s^{k-1})}(\sigma(s))}\right) D_{\psi_s^\Delta}(\pi_{p(s)}(\cdot\,|\,s), \overline{\pi}_{p(s)}^{(t_s^k)}(\cdot\,|\,s))
$$

$$
+ 2C_s^{\mathrm{diff}}\|\boldsymbol{q}\|_\infty\eta_s M_2|T_s| + \frac{1}{\mu_{p(s)}^{(t_s^1)}(\sigma(s))}D_{\psi_s^\Delta}(\pi_{p(s)}(\cdot\,|\,s), \overline{\pi}_{p(s)}^{(t_s^1)}(\cdot\,|\,s)).
$$

Since the probability of visiting infoset $s$ at timestep $t$ is at least $\gamma^2\sum_{h\in s}\mu_c(h)$,

$$
\Pr\left(|t_s^k - t_s^{k-1}| > K_s\right) \leq (1 - \gamma^2\sum_{h\in s}\mu_c(h))^{K_s} \leq \exp(-\gamma^2\sum_{h\in s}\mu_c(h)K_s).
$$

Therefore, with probability $1-\delta$, all infosets $s \in \mathcal{S}$ satisfies that for any $2 \leq k \leq |T_s|$, $|t_s^k - t_s^{k-1}| \leq \frac{\log T + \log|\mathcal{S}| + \log\frac{1}{\delta}}{\gamma^2\sum_{h\in s}\mu_c(h)} =: K_s$. Then,

$$
\frac{1}{\mu_{p(s)}^{(t_s^k)}(\sigma(s))} - \frac{1}{\mu_{p(s)}^{(t_s^{k-1})}(\sigma(s))} - \frac{\eta_s\tau}{\mu_{p(s)}^{(t_s^{k-1})}(\sigma(s))} \leq C_s^-\eta_s^{\mathrm{anc}}\frac{\log T + \log|\mathcal{S}| + \log\frac{1}{\delta}}{\gamma^2\sum_{h\in s}\mu_c(h)} - \eta_s\tau
$$

$$
= C_s^-\eta_s^{\mathrm{anc}}\frac{\log T + \log|\mathcal{S}| + \log\frac{1}{\delta}}{\gamma^2\sum_{h\in s}\mu_c(h)} - \eta_s\tau.
$$

Therefore, when $\frac{\eta_s^{\mathrm{anc}}}{\eta_s} \leq \frac{\tau\gamma^2 \sum_{h\in s}\mu_c(h)}{2C_s^- \left(\log T + \log|\mathcal{S}| + \log\frac{1}{\delta}\right)}$, the inequality above is upper-bounded by $-\frac{\eta_s\tau}{2}$.

Moreover, we can write it as (let $t_s^{|T_s|+1} = T + 1$ for notational simplicity),

$$\sum_{k=2}^{|T_s|} \left( \frac{1}{\mu_{p(s)}^{(t_s^k)}(\sigma(s))} - \frac{1}{\mu_{p(s)}^{(t_s^{k-1})}(\sigma(s))} - \frac{\eta_s\tau}{\mu_{p(s)}^{(t_s^{k-1})}(\sigma(s))} \right) D_{\psi_s^\Delta}(\pi_{p(s)}(\cdot\,|\,s), \bar{\pi}_{p(s)}^{(t_s^k)}(\cdot\,|\,s))$$

$$\leq -\sum_{k=2}^{|T_s|}\sum_{t=t_s^{(k)}}^{t_s^{(k+1)}-1} \frac{\eta_s\tau}{2K_s} D_{\psi_s^\Delta}(\pi_{p(s)}(\cdot\,|\,s), \bar{\pi}_{p(s)}^{(t_s^k)}(\cdot\,|\,s))$$

$$= -\frac{\eta_s\tau\gamma^2 \sum_{h\in s}\mu_c(h)}{2\left(\log T + \log|\mathcal{S}| + \log\frac{1}{\delta}\right)} \sum_{k=2}^{|T_s|}\sum_{t=t_s^{(k)}}^{t_s^{(k+1)}-1} D_{\psi_s^\Delta}(\pi_{p(s)}(\cdot\,|\,s), \bar{\pi}_{p(s)}^{(t_s^k)}(\cdot\,|\,s))$$

$$\overset{(i)}{=} -\frac{\eta_s\tau\gamma^2 \sum_{h\in s}\mu_c(h)}{2\left(\log T + \log|\mathcal{S}| + \log\frac{1}{\delta}\right)} \sum_{t=t_s^{(2)}}^{T} D_{\psi_s^\Delta}(\pi_{p(s)}(\cdot\,|\,s), \bar{\pi}_{p(s)}^{(t)}(\cdot\,|\,s))$$

$$\leq -\frac{\eta_s\tau\gamma^2 \sum_{h\in s}\mu_c(h)}{2\left(\log T + \log|\mathcal{S}| + \log\frac{1}{\delta}\right)} \sum_{t=2}^{T} D_{\psi_s^\Delta}(\pi_{p(s)}(\cdot\,|\,s), \bar{\pi}_{p(s)}^{(t)}(\cdot\,|\,s)) + 2K_s \max_{\boldsymbol{x},\boldsymbol{y}\in\Delta_{\gamma_s,\nu_s}^{|\mathcal{A}_s|}} D_{\psi_s^\Delta}(\boldsymbol{x},\boldsymbol{y}).$$

$(i)$ uses the fact that for any $t \in [t_s^k, t_s^{k+1} - 1]$, $\bar{\pi}_{p(s)}^{(t)}(\cdot\,|\,s) = \bar{\pi}_{p(s)}^{(t_s^k)}(\cdot\,|\,s)$.

Then, following rest of the proof for Theorem 4.2, we finish the proof. $\square$

### F.1 Proof of Lemma 4.3

Let $d^{(t)}(\boldsymbol{u}) := \left(f^{(t)}(\boldsymbol{u}) - f^{(t)}(\boldsymbol{u}^{(t)})\right) - \left(\widetilde{f}^{(t)}(\boldsymbol{u}) - \widetilde{f}^{(t)}(\boldsymbol{u}^{(t)})\right)$. By the property of $f^{(t)}$, since $\boldsymbol{u}^{(t)}$ is deterministically influenced by $\widetilde{f}^{(1)}, \widetilde{f}^{(2)}, \cdots, \widetilde{f}^{(t-1)}$, $\mathbb{E}\left[\widetilde{f}^{(t)}(\boldsymbol{u}^{(t)})\,|\,\widetilde{f}^{(1)}, \widetilde{f}^{(2)}, \cdots, \widetilde{f}^{(t-1)}\right] = f^{(t)}(\boldsymbol{u}^{(t)})$. Moreover, $\mathbb{E}\left[\widetilde{f}^{(t)}(\boldsymbol{u})\,|\,\widetilde{f}^{(1)}, \widetilde{f}^{(2)}, \cdots, \widetilde{f}^{(t-1)}\right] = f^{(t)}(\boldsymbol{u})$ for any fixed $\boldsymbol{u} \in \mathcal{C}$. Therefore, $d^{(t)}(\boldsymbol{u})$ is a martingale difference sequence. Then, we can apply the Azuma-Hoeffding inequality in the following.

**Lemma F.2** (Azuma-Hoeffding inequality). *For any martingale difference sequence* $x^{(1)}, x^{(2)}, \cdots, x^{(T)}$ *with* $x^{(t)} \in [a^{(t)}, b^{(t)}]$, *we have*

$$\Pr\left(\sum_{t=1}^{T} x^{(t)} \geq w\right) \leq \exp\left(-\frac{2w^2}{\sum_{t=1}^{T}\left(b^{(t)} - a^{(t)}\right)^2}\right).$$

Note that $\left|d^{(t)}(\boldsymbol{u})\right| \leq M + \widetilde{M}$. By applying Lemma F.2, we have

$$\Pr\left(\sum_{t=1}^{T} d^{(t)}(\boldsymbol{u}) \geq w\right) \leq \exp\left(-\frac{2w^2}{\sum_{t=1}^{T} 4(M + \widetilde{M})^2}\right).$$

Therefore, when taking $w = (M + \widetilde{M})\sqrt{2T\log\frac{1}{\delta}}$, with probability at least $1 - \delta$,

$$\sum_{t=1}^{T} \left(f^{(t)}(\boldsymbol{u}) - f^{(t)}(\boldsymbol{u}^{(t)})\right) \leq \sum_{t=1}^{T} \left(\widetilde{f}^{(t)}(\boldsymbol{u}) - \widetilde{f}^{(t)}(\boldsymbol{u}^{(t)})\right) + (M + \widetilde{M})\sqrt{2T\log\frac{1}{\delta}}. \qquad \square$$

## G Auxiliary Lemmas

In this section, we present the auxiliary lemmas for the theorems proved in the previous part.

### G.1 UPPERBOUND OF FEEDBACK

**Lemma G.1** (Upperbound of Feedback $q^{(t)}(s, \cdot)$). Consider the update-rule (4.2). For any timestep $t \in [T]$, we have the following upper-bound on $q^{(t)}(s, \cdot)$ and its unbiased estimator $\widetilde{q}^{(t)}(s, \cdot)$, no matter whether it is counterfactual value, trajectory Q-value, or Q-value.

$$\|\boldsymbol{q}\|_\infty := \begin{cases} \frac{\frac{\tau}{M_1}\|\boldsymbol{\alpha}\|_\infty \mathcal{D}\psi^{\max}+1}{\min_{s \in \mathcal{S}, a \in \mathcal{A}_s} \gamma_s \nu_{s,a}} & \text{Outcome Sampling of Trajectory Q-value} \\ \frac{\tau}{M_1}\|\boldsymbol{\alpha}\|_\infty \mathcal{D}\psi^{\max}+1 & \text{Otherwise} \end{cases} \tag{G.1}$$

where $\mathcal{D} := \max_{h \in \mathcal{H}} \mathcal{D}(h)$ is the maximum depth of infoset and $\psi^{\max}$ is the maximum of the regularizer, which is $\frac{1}{2\min_{s \in \mathcal{S}}|\mathcal{A}_s|}$ for Euclidean distance and $\max_{s \in \mathcal{S}} \log|\mathcal{A}_s|$ for entropy.

*Proof.* When calculating the feedback $\widetilde{q}^{(t)}(s, a)$ for outcome sampling of trajectory Q-value, we need to divide the probability of choosing action $a$, which is $\pi^{(t)}(a \mid s) \geq \min_{s \in \mathcal{S}, a \in \mathcal{A}_s} \gamma_s \nu_{s,a}$. Then, its upperbound is that of the full-information feedback setting divided by the constant $\min_{s \in \mathcal{S}, a \in \mathcal{A}_s} \gamma_s \nu_{s,a}$. Therefore, in the following, we will focus on the upperbound of $q^{(t)}(s, \cdot)$ in the full-information feedback setting.

In the following proof, we only consider $s \in \mathcal{S}_1$ since player $1, 2$ are symmetric. Furthermore, we only need to prove the upper-bound above when $q^{(t)}(s, \cdot)$ is Q-value, since by definition, the Q-value $Q_i^{\boldsymbol{\pi}}(s, a) = \frac{\overline{Q}_i^{\boldsymbol{\pi}}(s,a)}{\sum_{h \in s} \mu_c(h)\mu_1^{\pi_1}(\sigma_1(h))\mu_2^{\pi_2}(\sigma_2(h))} \geq \overline{Q}_i^{\boldsymbol{\pi}}(s, a)$ (similarly, it is also larger than the counterfactual value).

Let $s(h)$ be the infoset that node $h$ is in. Firstly, when $\tau = 0$, which means only considering the contribution of $\mathcal{U}_1$ to $q^{(t)}(s, \cdot)$, for every $s \in \mathcal{S}_1$, we have

$$|Q_i^{\boldsymbol{\pi}}(s, a)|$$

$$= \frac{1}{\sum_{h \in s} \mu_c(h)\mu_1^{\pi_1}(\sigma_1(h))\mu_2^{\pi_2}(\sigma_2(h))} \left| \sum_{h' : \exists h \in s, (h,a) \sqsubseteq h'} \mu_c(h')\mathcal{U}_1(h')\mu_1^{\pi_1}(\sigma_1(h'))\mu_2^{\pi_2}(\sigma_2(h')) \right|$$

$$\overset{(i)}{=} \frac{1}{\sum_{h \in s} \mu_c(h)\mu_1^{\pi_1}(\sigma_1(h))\mu_2^{\pi_2}(\sigma_2(h))} \left| \sum_{h' : \exists h \in s, (h,a) \sqsubseteq h', \mathcal{A}_{s(h')}=\emptyset} \mu_c(h')\mathcal{U}_1(h')\mu_1^{\pi_1}(\sigma_1(h'))\mu_2^{\pi_2}(\sigma_2(h')) \right|$$

$$\leq \frac{1}{\sum_{h \in s} \mu_c(h)\mu_1^{\pi_1}(\sigma_1(h))\mu_2^{\pi_2}(\sigma_2(h))} \sum_{h' : \exists h \in s, (h,a) \sqsubseteq h', \mathcal{A}_{s(h')}=\emptyset} \mu_c(h')\mu_1^{\pi_1}(\sigma_1(h'))\mu_2^{\pi_2}(\sigma_2(h'))$$

$$\overset{(ii)}{=} \frac{1}{\sum_{h \in s} \mu_c(h)\mu_1^{\pi_1}(\sigma_1(h))\mu_2^{\pi_2}(\sigma_2(h))} \sum_{h \in s} \mu_c(h)\mu_1^{\pi_1}(\sigma_1(h))\mu_2^{\pi_2}(\sigma_2(h)) = 1.$$

$(i)$ is because $\mathcal{U}_1(h) \neq 0$ only if $h$ is a terminal node. $(ii)$ is by the tree structure of EFG. Now consider $\tau > 0$ and $\mathcal{U}_1(h) \equiv 0$ for any $h \in \mathcal{H}$. Moreover, we will only show the upperbound when using dilated regularizer, since bidilated regularizer is upperbounded by the dilated one.

Let $S^{(t)}(s) = \left\langle q^{(t)}(s, \cdot), \pi_{p(s)}^{(t)}(\cdot \mid s) \right\rangle - \frac{\tau}{m_s^{(t)}}\psi_s^\Delta(\pi_{p(s)}^{(t)}(\cdot \mid s))$ when $\mathcal{A}_s \neq \emptyset$ ($s$ is not terminal node) and $S^{(t)}(s) = 0$ when $\mathcal{A}_s = \emptyset$ ($s$ is the terminal node).

We will prove $\left|S^{(t)}(s)\right| \leq \frac{\tau}{M_1}\|\boldsymbol{\alpha}\|_\infty (\mathcal{D} - \mathcal{D}(s)) \psi^{\max}$ by induction. For infoset $s \in \mathcal{S}_1$ with $\mathcal{D}(s) = D$, we have $S^{(t)}(s) = 0 = \tau \|\boldsymbol{\alpha}\|_\infty (\mathcal{D} - \mathcal{D}(s)) \psi^{\max}$. Therefore, the initial step of induction is completed.

Consider when all $s' \in \mathcal{S}_1$ with $\mathcal{D}(s') > d$ for some constant $d$, $S^{(t)}(s') \leq \frac{\tau}{M_1}\|\boldsymbol{\alpha}\|_\infty (\mathcal{D} - \mathcal{D}(s')) \psi^{\max}$. Let $\Pr(h \to h')$ be the probability of reaching $h'$ from node $h$, when considering all nodes encountered along the path (player $1, 2$ action node and the chance node) for

notational simplicity. Then, for infoset $s \in \mathcal{S}_1$ with $\mathcal{D}(s) = d$, we have

$$
\begin{aligned}
&|Q_i^{\boldsymbol{\pi}}(s,a)| \\
&= \left| \sum_{s' \in \mathcal{S}_1 : \sigma(s') = (s,a)} S^{(t)}(s') \sum_{h \in s} \frac{\mu_c(h)\mu_1^{\pi_1}(\sigma_1(h))\mu_2^{\pi_2}(\sigma_2(h))}{\sum_{h' \in s} \mu_c(h')\mu_1^{\pi_1}(\sigma_1(h'))\mu_2^{\pi_2}(\sigma_2(h'))} \sum_{h' \in s'} \Pr(h \to h') \right| \\
&\leq \sum_{s' \in \mathcal{S}_1 : \sigma(s') = (s,a)} \frac{\tau}{M_1} \|\boldsymbol{\alpha}\|_\infty (\mathcal{D} - \mathcal{D}(s) - 1) \psi^{\max} \\
&\quad \cdot \sum_{h \in s} \frac{\mu_c(h)\mu_1^{\pi_1}(\sigma_1(h))\mu_2^{\pi_2}(\sigma_2(h))}{\sum_{h' \in s} \mu_c(h')\mu_1^{\pi_1}(\sigma_1(h'))\mu_2^{\pi_2}(\sigma_2(h'))} \sum_{h' \in s'} \Pr(h \to h') \\
&= \frac{\tau}{M_1} \|\boldsymbol{\alpha}\|_\infty (\mathcal{D} - \mathcal{D}(s) - 1) \psi^{\max} \\
&\quad \cdot \sum_{h \in s} \frac{\mu_c(h)\mu_1^{\pi_1}(\sigma_1(h))\mu_2^{\pi_2}(\sigma_2(h))}{\sum_{h' \in s} \mu_c(h')\mu_1^{\pi_1}(\sigma_1(h'))\mu_2^{\pi_2}(\sigma_2(h'))} \sum_{h' \in \mathcal{H}_1 : \sigma_1(h') = (s,a)} \Pr(h \to h') \\
&= \frac{\tau}{M_1} \|\boldsymbol{\alpha}\|_\infty (\mathcal{D} - \mathcal{D}(s) - 1) \psi^{\max}.
\end{aligned}
$$

Therefore,

$$
\begin{aligned}
\left| S^{(t)}(s) \right| &= \left| \left\langle q^{(t)}(s, \cdot), \pi_{p(s)}^{(t)}(\cdot \mid s) \right\rangle - \frac{\tau}{m_s^{(t)}} \psi_s^\Delta(\pi_{p(s)}^{(t)}(\cdot \mid s)) \right| \\
&\leq \frac{\tau}{M_1} \|\boldsymbol{\alpha}\|_\infty (\mathcal{D} - \mathcal{D}(s) - 1) \psi^{\max} + \frac{\tau}{M_1} \|\boldsymbol{\alpha}\|_\infty \psi^{\max} \\
&= \frac{\tau}{M_1} \|\boldsymbol{\alpha}\|_\infty (\mathcal{D} - \mathcal{D}(s)) \psi^{\max}.
\end{aligned}
$$

This concludes the induction step. $\qquad \square$

## G.2 BOUNDING $M_1, M_2$

By choosing $\gamma_s$ as a fixed constant $\gamma_0 > 0$ for any player $i \in [2]$ and $s \in \mathcal{S}_i$. $\nu_{s,a}$ is chosen to be proportional to the number of terminal infosets ($s' \in \mathcal{S}$ with $\mathcal{A}_{s'} = \emptyset$) in the subtree rooted at $(s, a)$, we can get $\gamma \geq \frac{\gamma_0^{\mathcal{D}}}{|\mathcal{S}|}$. Then, we have the following lowerbound and upperbound on $m_s^{(t)}$.

**Lemma G.2.** When $\mu_1^{(t)}, \mu_2^{(t)} \succeq \gamma$, $m_s^{(t)}$ are lowerbounded by $M_1$ and upperbounded by $M_2$ with the following $M_1, M_2$ for different feedback.

$$
M_1 := \begin{cases} \gamma \min_{s \in \mathcal{S}} \sum_{h \in s} \mu_c(h) & \text{Q-value} \\ 1 & \text{Trajectory Q-value} \\ 1 & \text{Counterfactual Value} \end{cases} \qquad M_2 := \begin{cases} 1 & \text{Q-value} \\ \frac{1}{\gamma} & \text{Trajectory Q-value} \\ 1 & \text{Counterfactual Value} \end{cases} \tag{G.2}
$$

*Proof.* We only prove the lowerbound and upperbound for infoset $s \in \mathcal{S}_1$ since two players are symmetric.

For counterfactual value, since $m_s^{(t)} \equiv 1$, $M_1, M_2 = 1$.

For trajectory Q-value, since $m_s^{(t)} = \frac{1}{\mu_1^{(t)}(\sigma(s))}$, we have $M_2 = \frac{1}{\gamma} \geq m_s^{(t)} \geq 1 = M_1$.

For Q-value, $m_s^{(t)} = \sum_{h \in s} \mu_c(h)\mu_2^{(t)}(\sigma_2(h))$. Since the reach probability $\mu_1^{\pi_1}(\sigma(s)) \sum_{h \in s} \mu_c(h)\mu_2^{(t)}(\sigma_2(h)) \leq 1$ for any $\pi_1$, we can let $\pi_1$ play deterministically to reach $s$. In this way, $m_s^{(t)}$ is equal to the reach probability so it is also upperbounded by one. At the same time, $m_s^{(t)} \geq \gamma \sum_{h \in s} \mu_c(h) \geq \gamma \min_{s \in \mathcal{S}} \sum_{h \in s} \mu_c(h)$. $\qquad \square$

## G.3 Proof of Lemma C.2

**Lemma G.3.** Consider update-rule (4.2). When $\psi_s^\Delta(\boldsymbol{u}) = \frac{\alpha_s}{2} \sum_{a \in \mathcal{A}_s} u_a^2$ is the Euclidean distance where $\alpha_s > 0$ is a constant, we have

$$C_s^{\text{diff}} = \frac{|\mathcal{A}_s|}{\alpha_s} \|\boldsymbol{q}\|_\infty + \frac{\tau}{M_1} \sqrt{|\mathcal{A}_s|}. \tag{G.3}$$

Let $\tau_s^{(t)} := \frac{\tau \mu_{-p(s)}^{(t)}(s)}{m_s^{(t)}}$. When $\psi_s^\Delta(\boldsymbol{u}) = \frac{\alpha_s}{2} \sum_{a \in \mathcal{A}_s} u_a^2$, we have

$$\left\| \pi_{p(s)}^{(t)}(\cdot \,|\, s) - \bar{\pi}_{p(s)}^{(t)}(\cdot \,|\, s) \right\|$$

$$= \left\| \text{Proj}_{\Delta_{\gamma_s, \nu_s}^{|\mathcal{A}_s|}} \left( \frac{\bar{\pi}_{p(s)}^{(t)}(\cdot \,|\, s)}{1 + \eta_s \tau_s^{(t-1)}} + \frac{\eta_s}{\alpha_s (1 + \eta_s \tau_s^{(t-1)})} q^{(t-1)}(s, \cdot) \right) - \bar{\pi}_{p(s)}^{(t)}(\cdot \,|\, s) \right\|$$

$$\leq \left\| \frac{\bar{\pi}_{p(s)}^{(t)}(\cdot \,|\, s)}{1 + \eta_s \tau_s^{(t-1)}} + \frac{\eta_s}{\alpha_s (1 + \eta_s \tau_s^{(t-1)})} q^{(t-1)}(s, \cdot) - \bar{\pi}_{p(s)}^{(t)}(\cdot \,|\, s) \right\|$$

$$\leq \eta_s \left( \frac{1}{\alpha_s (1 + \eta_s \tau_s^{(t-1)})} \left\| q^{(t-1)}(s, \cdot) \right\| + \frac{\tau_s^{(t-1)}}{1 + \eta_s \tau_s^{(t-1)}} \left\| \bar{\pi}_{p(s)}^{(t)}(\cdot \,|\, s) \right\| \right)$$

$$\leq \eta_s \left( \frac{1}{\alpha_s} \left\| q^{(t-1)}(s, \cdot) \right\| + \tau_s^{(t-1)} \left\| \bar{\pi}_{p(s)}^{(t)}(\cdot \,|\, s) \right\| \right)$$

$$\leq \eta_s \left( \frac{\sqrt{|\mathcal{A}_s|}}{\alpha_s} \|\boldsymbol{q}\|_\infty + \frac{\tau}{M_1} \right).$$

In the last line, we use the fact that $\mu_{-p(s)}^{(t)}(s) \leq 1$.

As a result, $\left\| \pi_{p(s)}^{(t)}(\cdot \,|\, s) - \bar{\pi}_{p(s)}^{(t)}(\cdot \,|\, s) \right\|_1 \leq \eta_s \left( \frac{|\mathcal{A}_s|}{\alpha_s} \|\boldsymbol{q}\|_\infty + \frac{\tau}{M_1} \sqrt{|\mathcal{A}_s|} \right).$

Similarly, $\left\| \bar{\pi}_{p(s)}^{(t+1)}(\cdot \,|\, s) - \bar{\pi}_{p(s)}^{(t)}(\cdot \,|\, s) \right\|_1 \leq \eta_s \left( \frac{|\mathcal{A}_s|}{\alpha_s} \|\boldsymbol{q}\|_\infty + \frac{\tau}{M_1} \sqrt{|\mathcal{A}_s|} \right).$ □

*Proof of Lemma C.5.* Let $\tau_s^{(t)} := \frac{\tau \mu_{-p(s)}^{(t)}(s)}{m_s^{(t)}}$. When $\psi_s^\Delta(\boldsymbol{u}) = \alpha_s \left( \log |\mathcal{A}_s| + \sum_{a \in \mathcal{A}_s} u_a \log u_a \right)$, the update-rule (4.2) is equivalent to

$$\pi^{(t)}(a \,|\, s)$$

$$= \max \left\{ \frac{\bar{\pi}^{(t)}(a \,|\, s)^{\frac{1}{1 + \eta_s \tau_s^{(t-1)}}} \exp \left( \frac{\eta_s}{\alpha_s \left( 1 + \eta_s \tau_s^{(t-1)} \right)} q^{(t)}(s, a) \right)}{Z}, \gamma_s \nu_{s,a} \right\}$$

$$= \max \left\{ \frac{\bar{\pi}^{(t)}(a \,|\, s) \exp \left( \frac{\eta_s}{\alpha_s \left( 1 + \eta_s \tau_s^{(t-1)} \right)} q^{(t)}(s, a) - \frac{\eta_s \tau}{m_s^{(t-1)} \left( 1 + \eta_s \tau_s^{(t-1)} \right)} \log \bar{\pi}^{(t)}(a \,|\, s) \right)}{Z}, \gamma_s \nu_{s,a} \right\}$$

for any $a \in \mathcal{A}_s$, where $Z > 0$ is a normalizing constant to ensure $\pi^{(t)}(a \,|\, s)$ is still a probability distribution over $\Delta^{|\mathcal{A}_s|}$. The equivalency is proved in Lemma G.4. For notational simplicity, we define $l^{(t)}(s, a) := -\frac{1}{\alpha_s \left( 1 + \eta_s \tau_s^{(t-1)} \right)} q^{(t)}(s, a) + \frac{\tau}{m_s^{(t-1)} \left( 1 + \eta_s \tau_s^{(t-1)} \right)} \log \bar{\pi}^{(t)}(a \,|\, s)$ so that

$$\pi^{(t)}(a \,|\, s) = \max \left\{ \frac{\bar{\pi}^{(t)}(a \,|\, s) \exp \left( -\eta_s l^{(t)}(s, a) \right)}{Z}, \gamma_s \nu_{s,a} \right\}.$$

Firstly, for $\gamma_s = 1$, we have $\pi_{p(s)}^{(t)}(\cdot \mid s) = \bar{\pi}_{p(s)}^{(t)}(\cdot \mid s) = \nu_s$. Therefore, $C_s^{\text{diff}} = 0$ and $\frac{\pi^{(t)}(a \mid s)}{\bar{\pi}^{(t)}(a \mid s)} = 1$ for any $a \in \mathcal{A}_s$. In the following, we assume $\gamma_s < 1$.

We can see that $\pi^{(t)}(a \mid s)$ is monotonically decreasing with respect to $Z$. When $Z < \exp\left(-\eta_s \max_{a' \in \mathcal{A}_s} l^{(t)}(s, a')\right)$, for any $a \in \mathcal{A}_s$, we have $\pi^{(t)}(a \mid s) \geq \frac{\bar{\pi}^{(t)}(a \mid s) \exp\left(-\eta_s l^{(t)}(s,a)\right)}{Z} > \bar{\pi}^{(t)}(a \mid s)$. Then, $\sum_{a \in \mathcal{A}_s} \pi^{(t)}(a \mid s) > 1$.

Therefore, $Z \geq \exp\left(-\eta_s \max_{a' \in \mathcal{A}_s} l^{(t)}(s, a')\right)$.

Similarly, when $Z > \exp\left(-\eta_s \min_{a' \in \mathcal{A}_s} l^{(t)}(s, a')\right)$, for any $a \in \mathcal{A}_s$, we have $\frac{\bar{\pi}^{(t)}(a \mid s) \exp\left(-\eta_s l^{(t)}(s,a)\right)}{Z} < \bar{\pi}^{(t)}(a \mid s)$. It implies that $\sum_{a \in \mathcal{A}_s} \pi^{(t)}(a \mid s) < \sum_{a \in \mathcal{A}_s} \bar{\pi}^{(t)}(a \mid s)$, unless $\bar{\pi}^{(t)}(a \mid s) = \gamma_s \nu_{s,a}$ for all $a \in \mathcal{A}_s$, which is impossible since we assume $\gamma_s < 1$. Therefore, $Z \leq \exp\left(-\eta_s \min_{a' \in \mathcal{A}_s} l^{(t)}(s, a')\right)$.

Then, if $\frac{\bar{\pi}^{(t)}(a \mid s) \exp\left(-\eta_s l^{(t)}(s,a)\right)}{Z} \geq \gamma_s \nu_{s,a}$, we have

$$
\begin{aligned}
1 &\leq \frac{\pi^{(t)}(a \mid s)}{\bar{\pi}^{(t)}(a \mid s)} \\
&= \frac{\exp\left(-\eta_s l^{(t)}(s, a)\right)}{Z} \\
&\leq \exp\left(\eta_s \max_{a' \in \mathcal{A}_s} l^{(t)}(s, a') - \eta_s l^{(t)}(s, a)\right) \\
&\leq \exp\left(\frac{\eta_s}{\alpha_s\left(1 + \eta_s \tau_s^{(t-1)}\right)} \left(\max_{a' \in \mathcal{A}_s} q^{(t)}(s, a') - q^{(t)}(s, a)\right) + \frac{\eta_s \tau_s^{(t-1)}}{1 + \eta_s \tau_s^{(t-1)}} \log \max_{a' \in \mathcal{A}_s} \frac{\bar{\pi}^{(t)}(a' \mid s)}{\bar{\pi}^{(t)}(a \mid s)}\right) \\
&\overset{(i)}{\leq} \exp\left(\frac{\eta_s}{\alpha_s}\left(2 \|\boldsymbol{q}\|_\infty + \frac{\tau \alpha_s}{M_1} \log \frac{1}{\gamma}\right)\right) \\
&\overset{(ii)}{\leq} 1 + 2\frac{\eta_s}{\alpha_s}\left(2 \|\boldsymbol{q}\|_\infty + \frac{\tau \alpha_s}{M_1} \log \frac{1}{\gamma}\right)
\end{aligned}
$$

In $(i)$ we use the fact $\|\boldsymbol{q}\|_\infty \geq \left|q^{(t)}(s, a)\right|$ for any $a \in \mathcal{A}_s$. In $(ii)$ we use $e^x \leq 1 + 2x$ for $x \in [0, 1]$.

If $\frac{\bar{\pi}^{(t)}(a \mid s) \exp\left(-\eta_s l^{(t)}(s,a)\right)}{Z} < \bar{\pi}^{(t)}(a \mid s)$, we have

$$
\begin{aligned}
\frac{\exp\left(-\eta_s l^{(t)}(s, a)\right)}{Z} &\geq \exp\left(\eta_s \min_{a' \in \mathcal{A}_s} l^{(t)}(s, a') - \eta_s l^{(t)}(s, a)\right) \\
&\geq \exp\left(-\frac{\eta_s}{\alpha_s}\left(2 \|\boldsymbol{q}\|_\infty + \frac{\tau \alpha_s}{M_1} \log \frac{1}{\gamma}\right)\right).
\end{aligned}
$$

Therefore,

$$
\begin{aligned}
1 \geq \frac{\pi^{(t)}(a \mid s)}{\bar{\pi}^{(t)}(a \mid s)} \geq \frac{\bar{\pi}^{(t)}(a \mid s) \exp\left(-\eta_s l^{(t)}(s, a)\right)}{\bar{\pi}^{(t)}(a \mid s) Z} &\geq \exp\left(-\frac{\eta_s}{\alpha_s}\left(2 \|\boldsymbol{q}\|_\infty + \frac{\tau \alpha_s}{M_1} \log \frac{1}{\gamma}\right)\right) \\
&\geq 1 - \frac{\eta_s}{\alpha_s}\left(2 \|\boldsymbol{q}\|_\infty + \frac{\tau \alpha_s}{M_1} \log \frac{1}{\gamma}\right).
\end{aligned}
$$

Then, for any $a \in \mathcal{A}_s$, we have

$$
\left|\frac{\pi^{(t)}(a \mid s)}{\bar{\pi}^{(t)}(a \mid s)} - 1\right| \leq 2\frac{\eta_s}{\alpha_s}\left(2 \|\boldsymbol{q}\|_\infty + \frac{\tau \alpha_s}{M_1} \log \frac{1}{\gamma}\right).
$$

Therefore,

$$
\left\|\pi_{p(s)}^{(t)}(\cdot \mid s) - \bar{\pi}_{p(s)}^{(t)}(\cdot \mid s)\right\|_1 = \sum_{a \in \mathcal{A}_s} \bar{\pi}^{(t)}(a \mid s)\left|\frac{\pi^{(t)}(a \mid s)}{\bar{\pi}^{(t)}(a \mid s)} - 1\right| \leq 2\frac{\eta_s}{\alpha_s}\left(2 \|\boldsymbol{q}\|_\infty + \frac{\tau \alpha_s}{M_1} \log \frac{1}{\gamma}\right).
$$

Similarly, the upperbound above also holds for $\left|\frac{\bar{\pi}^{(t+1)}(a \mid s)}{\bar{\pi}^{(t)}(a \mid s)} - 1\right|$. $\square$

### G.4 UPDATE RULE OF MWU

For ease of representation, we ignore the learning rate $\eta$ without loss of generality (the update-rule is the same without $\eta$ when multiplying both the gradient $\boldsymbol{g}$ and $\tau$ by $\eta$).

**Lemma G.4.** Consider the update-rule $\boldsymbol{x}^{(2)} = \mathrm{argmin}_{\boldsymbol{x} \in \Delta_{|\mathcal{A}|}^{\gamma,\boldsymbol{\nu}}} \langle \boldsymbol{x}, \boldsymbol{g} \rangle + \tau \psi(\boldsymbol{x}) + D_\psi(\boldsymbol{x}, \boldsymbol{x}^{(1)})$ where $\boldsymbol{x}^{(1)} \in \Delta_{|\mathcal{A}|}^{\gamma,\boldsymbol{\nu}}$, $\gamma \geq 0$ is a constant, and $\boldsymbol{\nu} \in \Delta^{|\mathcal{A}|}$, $\mathcal{A}$ is the action set and $\psi(\boldsymbol{x}) = \sum_{a \in \mathcal{A}} x \log x + \log|\mathcal{A}|$. Then, the update-rule is equivalent to

$$x_a^{(2)} = \max \left\{ \frac{\left(x_a^{(1)}\right)^{\frac{1}{1+\tau}} \exp(-\frac{g_a}{1+\tau})}{Z}, \gamma\nu_a \right\} \tag{G.4}$$

for any $a \in \mathcal{A}$. $Z > 0$ is the normalizing constant to ensure $\sum_{a \in \mathcal{A}} x_a^{(2)} = 1$.

*Proof.* By definition of Bregman divergence, $D_\psi(\boldsymbol{x}, \boldsymbol{x}^{(1)}) = \sum_{a \in \mathcal{A}} x_a \log \frac{x_a}{x_a^{(1)}}$. Therefore, the Lagrangian function of the update-rule is

$$\mathcal{F}(\boldsymbol{x}, \alpha, \boldsymbol{\beta}) := \langle \boldsymbol{x}, \boldsymbol{g} \rangle + \tau \sum_{a \in \mathcal{A}} x_a \log x_a + \sum_{a \in \mathcal{A}} x_a \log \frac{x_a}{x_a^{(1)}} + \alpha(\sum_{a \in \mathcal{A}} x_a - 1) + \sum_{a \in \mathcal{A}} \beta_a(x_a - \gamma\nu_a).$$

By taking $\nabla_{\boldsymbol{x}} \mathcal{F}(\boldsymbol{x}, \alpha, \boldsymbol{\lambda}) = 0$, for any $a \in \mathcal{A}$, we have

$$g_a + \tau \log x_a^{(2)} + \log \frac{x_a^{(2)}}{x_a^{(1)}} + 1 + \tau + \alpha + \beta_a = 0,$$

which implies that

$$x_a^{(2)} = \left(x_a^{(1)}\right)^{\frac{1}{1+\tau}} \exp\left(-\frac{1}{1+\tau}\left(g_a + 1 + \tau + \alpha + \beta_a\right)\right).$$

By duality, $\beta_a \leq 0$. By complementary slackness, we have $\beta_a(x_a^{(2)} - \gamma\nu_a) = 0$. Therefore, when $\beta_a < 0$, we have $x_a^{(2)} = \gamma\nu_a$, which implies that $\left(x_a^{(1)}\right)^{\frac{1}{1+\tau}} \exp\left(-\frac{1}{1+\tau}\left(g_a + 1 + \tau + \alpha\right)\right) < \gamma\nu_a$. When $\beta_a = 0$, we have $x_a^{(2)} \geq \gamma\nu_a$ so that $\left(x_a^{(1)}\right)^{\frac{1}{1+\tau}} \exp\left(-\frac{1}{1+\tau}\left(g_a + 1 + \tau + \alpha\right)\right) \geq \gamma\nu_a$. Therefore, the effect of $\beta_a$ is equivalent to take a max on $\left(x_a^{(1)}\right)^{\frac{1}{1+\tau}} \exp\left(-\frac{1}{1+\tau}\left(g_a + 1 + \tau + \alpha\right)\right)$ and we have the update-rule

$$x_a^{(2)} = \max \left\{ \left(x_a^{(1)}\right)^{\frac{1}{1+\tau}} \exp\left(-\frac{1}{1+\tau}\left(g_a + 1 + \tau + \alpha\right)\right), \gamma\nu_a \right\}$$

$$= \max \left\{ \frac{\left(x_a^{(1)}\right)^{\frac{1}{1+\tau}} \exp(-\frac{g_a}{1+\tau})}{Z}, \gamma\nu_a \right\}$$

where $Z = \exp\left(\frac{1+\tau+\alpha}{1+\tau}\right)$. $\square$

**Remark G.5.** In practice, we can implement the update-rule in Lemma G.4 as follows. We assume $\gamma < 1$ since otherwise we can simply let $\boldsymbol{x}^{(2)} = \boldsymbol{\nu}$.

- Compute $\widehat{x}_a = \left(x_a^{(1)}\right)^{\frac{1}{1+\tau}} \exp(-\frac{g_a}{1+\tau})$ and sort it in increasing order, which is $\widehat{x}_1 \leq \widehat{x}_2 \leq \cdots \leq \widehat{x}_{|\mathcal{A}|}$. Simultaneously, adjusting $\boldsymbol{\nu}$ according to the sorting of $\widehat{\boldsymbol{x}}$ to get $\widehat{\boldsymbol{\nu}}$, which is the lowerbound $\widehat{\boldsymbol{x}}$ should satisfy.

- Enumerate $i = 0, 1, 2, \cdots, |\mathcal{A}|$. Let $Z = \frac{\sum_{j>i} \widehat{x}_j}{1 - \gamma \sum_{j=1}^{i} \widehat{\nu}_j}$.

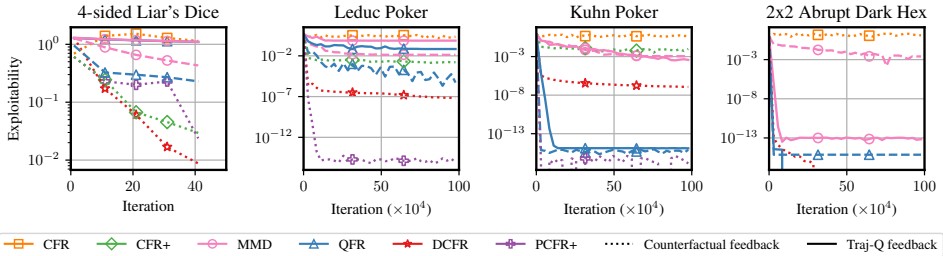

Figure 2: The result of full-information feedback in four benchmark games. We compare with CFR (Zinkevich et al., 2007), CFR+ (Tammelin et al., 2015), MMD (Sokota et al., 2023), DCFR (Brown and Sandholm, 2019), and PCFR+ (Farina et al., 2021a). We can see that QFR outperforms MMD in all games. However, due to multiplicative noise caused by using Q-values, QFR cannot outperform PCFR+, an advanced variant of CFR.

- Check $\widehat{x}_i \leq \gamma \widehat{\nu}_i Z$ if $i > 0$ and $\widehat{x}_{i+1} \geq \gamma \widehat{\nu}_{i+1} Z$ if $i < |\mathcal{A}|$. If the current $Z$ satisfies, return it. Otherwise, continue the enumeration.

According to the monotonicity of $\max\left\{\dfrac{\left(x_a^{(1)}\right)^{\frac{1}{1+\tau}} \exp(-\frac{g_a}{1+\tau})}{Z}, \gamma \nu_a\right\}$ with respect to $Z$, the algorithm above will definitely find the correct $Z$ and the time complexity is $O(|\mathcal{A}| \log |\mathcal{A}|)$ (the bottleneck is the sort).

## H    EXPERIMENT DETAILS

Figure 1 and Figure 2 are conducted on 240 cores of Intel Xeon Platinum 8260 and Figure 3 is conducted on Intel(R) Xeon Gold 6248 with NVidia Volta V100. The code is based on LiteEFG (Liu et al., 2024) with game environments implemented by OpenSpiel (Lanctot et al., 2019). The range of grid search and the best hyper-parameters can be found in the supplementary materials.

We present the experimental results for full information feedback in Figure 2, that is, we use trajectory Q-value, Q-value, or counterfactual value as $q^{(t)}(s, \cdot)$ in the update rule (4.2). In Figure 2, we plot the last-iterate performance for all the algorithms to ensure the comparison is fair.

In Figure 3, we present the ablation study on importance sampling when the strategy is approximated by the neural network. Our implementation of QFR is based on PPO (Schulman et al., 2017) in CleanRL (Huang et al., 2022). We can see that with importance sampling, the network gradient blows up so that the network does not converge, even though we have applied gradient clipping.

## I    QFR WITH LAZY UPDATE

In this section, we will show a variant of QFR, which we coin it as the lazy update version. In this version, the original dilated regularizer is applied instead of the bidilated one. The advantage of this variant is that the convergence only requires the proportion of $\eta_s$ and $\eta_s^{\text{anc}}$ to be a constant with Q-value and counterfactual value, so that the convergence rate depends polynomially on the game size.

The disadvantage is that importance sampling is needed for the additional regularizer when using stochastic feedback (the dispersion of feedback is not very large since the magnitude of the additional regularizer is controlled by $\tau$). Moreover, when sampling a trajectory, for those infosets that are not on the trajectory, we still need to update the strategy $\pi(\cdot \mid s)$ in them. In other words, for those infosets with $\widetilde{q}^{(t)} = 0$, we still need to update the strategy due to the additional regularizer (please refer to (I.4) for details). In practice, we can ignore the update of those infosets that are not along the trajectory, and postpone the update till the next time visiting them. That's why we call it a lazy update.

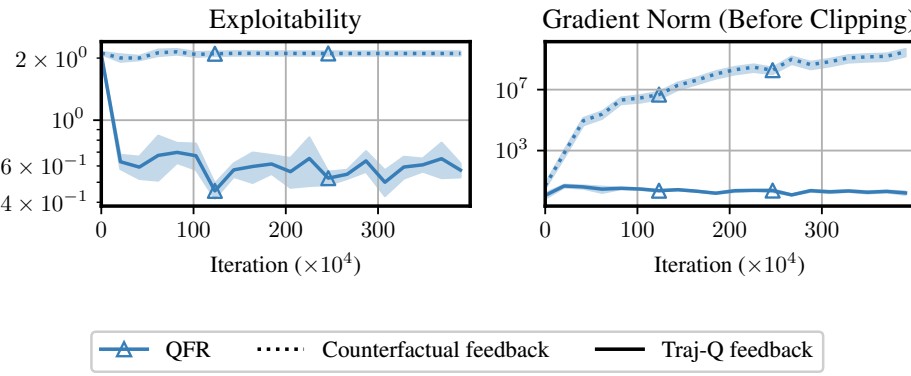

Figure 3: The result of QFR with sampling feedback. We can see that with importance sampling, the gradient norm keeps growing so that the network does not converge even with gradient clipping. The right figure shows the gradient **before clipping** and the gradient will be clipped so that its norm is bounded by 0.5.

Consider the following update-rule, which is (4.2) with dilated regularizer instead of the bidilated one.

$$
\pi_{p(s)}^{(t)}(\cdot \,|\, s) = \operatorname*{argmin}_{\pi_{p(s)}(\cdot \,|\, s) \in \Delta_{|\mathcal{A}_s|}^{\gamma_s, \nu_s}} \left\langle \pi_{p(s)}(\cdot \,|\, s), -q^{(t-1)}(s, \cdot) \right\rangle + \frac{\tau}{m_s^{(t-1)}} \psi_s^{\Delta}(\pi_{p(s)}(\cdot \,|\, s))
$$

$$
+ \frac{1}{\eta_s} D_{\psi_s^{\Delta}}(\pi_{p(s)}(\cdot \,|\, s), \bar{\pi}_{p(s)}^{(t)}(\cdot \,|\, s)) \tag{I.1}
$$

$$
\bar{\pi}_{p(s)}^{(t+1)}(\cdot \,|\, s) = \operatorname*{argmin}_{\pi_{p(s)}(\cdot \,|\, s) \in \Delta_{|\mathcal{A}_s|}^{\gamma_s, \nu_s}} \left\langle \pi_{p(s)}(\cdot \,|\, s), -q^{(t)}(s, \cdot) \right\rangle + \frac{\tau}{m_s^{(t)}} \psi_s^{\Delta}(\pi_{p(s)}(\cdot \,|\, s))
$$

$$
+ \frac{1}{\eta_s} D_{\psi_s^{\Delta}}(\pi_{p(s)}(\cdot \,|\, s), \bar{\pi}_{p(s)}^{(t)}(\cdot \,|\, s)) \tag{I.2}
$$

As Theorem 4.2, we can show the convergence of this update-rule. Note that in the following theorem, the proportion of $\eta_s$ and $\eta_s^{\mathrm{anc}}$ is now a constant when the feedback is Q-value or counterfactual value, which implies a polynomial dependence on the game size.

**Theorem I.1.** Consider the update rule (I.1) and $q^{(t)}(s, \cdot)$ is chosen to be counterfactual value, trajectory Q-value, or Q-value. When $\frac{\eta_s^{\mathrm{anc}}}{\eta_s} \leq \frac{\tau}{2C_s^-}$ for any $s \in \mathcal{S}$ and **(A)**,**(B)**,**(C)** are satisfied, we have the following guarantee.

$$
\sum_{t=2}^{T} D_{\psi^{\Pi}}(\boldsymbol{\mu}^{(\tau,\gamma),*}, \bar{\boldsymbol{\mu}}^{(t)})
$$

$$
\leq 2 \sum_{s \in \mathcal{S}} C_s' \eta_s^{\mathrm{anc}} \mu^{(\tau,\gamma),*}(\sigma(s)) \sum_{t=1}^{T} \left| \psi_s^{\Delta}(\pi_{p(s)}^{(t)}(\cdot \,|\, s)) - \psi_s^{\Delta}(\bar{\pi}_{p(s)}^{(t+1)}(\cdot \,|\, s)) \right| \tag{I.3}
$$

$$
+ \frac{4}{\tau} \sum_{s \in \mathcal{S}} C_s^{\mathrm{diff}} \mu^{(\tau,\gamma),*}(\sigma(s)) \|\boldsymbol{q}\|_{\infty} \eta_s M_2 T
$$

$$
+ \frac{2}{\tau} \sum_{s \in \mathcal{S}} \frac{M_2}{\eta_s} \mu^{(\tau,\gamma),*}(\sigma(s)) D_{\psi_s^{\Delta}}(\pi_{p(s)}(\cdot \,|\, s), \bar{\pi}_{p(s)}^{(1)}(\cdot \,|\, s)).
$$

**Lemma I.2** (Generalized from Lemma C.2. in Liu et al. (2023)). Consider the update rule in (4.2). When $\psi_s^\Delta$ is strongly convex, then for any $\pi_{p(s)}(\cdot \mid s) \in \Delta^{|\mathcal{A}_s|}$ and $t \geq 1$, we have

$$
\eta_s \frac{\tau}{m_s^{(t)}} \psi_s^\Delta(\pi_{p(s)}^{(t)}(\cdot \mid s)) - \eta_s \frac{\tau}{m_s^{(t)}} \psi_s^\Delta(\pi_{p(s)}(\cdot \mid s))
$$

$$
+ \eta_s \tau \left( \frac{1}{m_s^{(t-1)}} - \frac{1}{m_s^{(t)}} \right) (\psi_s^\Delta(\pi_{p(s)}^{(t)}(\cdot \mid s)) - \psi_s^\Delta(\bar\pi_{p(s)}^{(t+1)}(\cdot \mid s)))
$$

$$
+ \eta_s \langle -q^{(t)}(s, \cdot), \pi_{p(s)}^{(t)}(\cdot \mid s) - \pi_{p(s)}(\cdot \mid s) \rangle
$$

$$
\leq D_{\psi_s^\Delta}(\pi_{p(s)}(\cdot \mid s), \bar\pi_{p(s)}^{(t)}(\cdot \mid s)) - (1 + \eta_s \frac{\tau}{m_s^{(t)}}) D_{\psi_s^\Delta}(\pi_{p(s)}(\cdot \mid s), \bar\pi_{p(s)}^{(t+1)}(\cdot \mid s))
$$

$$
- (1 + \eta_s \frac{\tau}{m_s^{(t-1)}}) D_{\psi_s^\Delta}(\bar\pi_{p(s)}^{(t+1)}(\cdot \mid s), \pi_{p(s)}^{(t)}(\cdot \mid s))
$$

$$
- D_{\psi_s^\Delta}(\pi_{p(s)}^{(t)}(\cdot \mid s), \bar\pi_{p(s)}^{(t)}(\cdot \mid s)) + \eta_s \left\langle q^{(t-1)}(s, \cdot) - q^{(t)}(s, \cdot), \pi_{p(s)}^{(t)}(\cdot \mid s) - \bar\pi_{p(s)}^{(t+1)}(\cdot \mid s) \right\rangle.
$$

The lemma's proof is similar to Lemma E.2.

Multiplying $m_s^{(t)}$ on both sides of Lemma E.2, we have

$$
\eta_s \tau \psi_s^\Delta(\pi_{p(s)}^{(t)}(\cdot \mid s)) - \eta_s \tau \psi_s^\Delta(\pi_{p(s)}(\cdot \mid s)) + \eta_s \tau \left( \frac{m_s^{(t)}}{m_s^{(t-1)}} - 1 \right) (\psi_s^\Delta(\pi_{p(s)}^{(t)}(\cdot \mid s)) - \psi_s^\Delta(\bar\pi_{p(s)}^{(t+1)}(\cdot \mid s)))
$$

$$
+ \eta_s m_s^{(t)} \langle -q^{(t)}(s, \cdot), \pi_{p(s)}^{(t)}(\cdot \mid s) - \pi_{p(s)}(\cdot \mid s) \rangle
$$

$$
\leq m_s^{(t)} D_{\psi_s^\Delta}(\pi_{p(s)}(\cdot \mid s), \bar\pi_{p(s)}^{(t)}(\cdot \mid s)) - (m_s^{(t)} + \eta_s \tau) D_{\psi_s^\Delta}(\pi_{p(s)}(\cdot \mid s), \bar\pi_{p(s)}^{(t+1)}(\cdot \mid s))
$$

$$
- (m_s^{(t)} + \eta_s \tau \frac{m_s^{(t)}}{m_s^{(t-1)}}) D_{\psi_s^\Delta}(\bar\pi_{p(s)}^{(t+1)}(\cdot \mid s), \pi_{p(s)}^{(t)}(\cdot \mid s))
$$

$$
- m_s^{(t)} D_{\psi_s^\Delta}(\pi_{p(s)}^{(t)}(\cdot \mid s), \bar\pi_{p(s)}^{(t)}(\cdot \mid s)) + \eta_s m_s^{(t)} \left\langle q^{(t-1)}(s, \cdot) - q^{(t)}(s, \cdot), \pi_{p(s)}^{(t)}(\cdot \mid s) - \bar\pi_{p(s)}^{(t+1)}(\cdot \mid s) \right\rangle.
$$

By using Property 2, we have

$$
\left( \frac{m_s^{(t)}}{m_s^{(t-1)}} - 1 \right) (\psi_s^\Delta(\pi_{p(s)}^{(t)}(\cdot \mid s)) - \psi_s^\Delta(\bar\pi_{p(s)}^{(t+1)}(\cdot \mid s)))
$$

$$
\geq - C_s^/ \eta_s^{\mathrm{anc}} \left| \psi_s^\Delta(\pi_{p(s)}^{(t)}(\cdot \mid s)) - \psi_s^\Delta(\bar\pi_{p(s)}^{(t+1)}(\cdot \mid s)) \right|.
$$

Furthermore, by using Lemma C.2 and Hölder's Inequality, we have

$$
\left| \left\langle q^{(t-1)}(s, \cdot) - q^{(t)}(s, \cdot), \pi_{p(s)}^{(t)}(\cdot \mid s) - \bar\pi_{p(s)}^{(t+1)}(\cdot \mid s) \right\rangle \right|
$$

$$
\leq \left\| q^{(t)}(s, \cdot) - q^{(t-1)}(s, \cdot) \right\|_\infty \cdot \left\| \pi_{p(s)}^{(t)}(\cdot \mid s) - \bar\pi_{p(s)}^{(t+1)}(\cdot \mid s) \right\|_1 \leq 2 C_s^{\mathrm{diff}} \|\boldsymbol{q}\|_\infty \eta_s.
$$

where $\|\boldsymbol{q}\|_\infty = \max_{t \in [T], s \in \mathcal{S}} \left\| q^{(t)}(s, \cdot) \right\|_\infty$.

By telescoping and non-negativity of Bregman divergence, we have

$$
\sum_{t=1}^T \left( \eta_s \tau \psi_s^\Delta(\pi_{p(s)}^{(t)}(\cdot \mid s)) - \eta_s \tau \psi_s^\Delta(\pi_{p(s)}(\cdot \mid s)) + \eta_s m_s^{(t)} \langle -q^{(t)}(s, \cdot), \pi_{p(s)}^{(t)}(\cdot \mid s) - \pi_{p(s)}(\cdot \mid s) \rangle \right)
$$

$$
\leq \sum_{t=2}^T \underbrace{\left( m_s^{(t)} - m_s^{(t-1)} - \eta_s \tau \right)}_{\textcircled{1}} D_{\psi_s^\Delta}(\pi_{p(s)}(\cdot \mid s), \bar\pi_{p(s)}^{(t)}(\cdot \mid s))
$$

$$
+ C_s^/ \eta_s \tau \eta_s^{\mathrm{anc}} \sum_{t=1}^T \left| \psi_s^\Delta(\pi_{p(s)}^{(t)}(\cdot \mid s)) - \psi_s^\Delta(\bar\pi_{p(s)}^{(t+1)}(\cdot \mid s)) \right| + 2 C_s^{\mathrm{diff}} \|\boldsymbol{q}\|_\infty \eta_s^2 M_2 T
$$

$$
+ m_s^{(1)} D_{\psi_s^\Delta}(\pi_{p(s)}(\cdot \mid s), \bar\pi_{p(s)}^{(1)}(\cdot \mid s)).
$$

①can be upper-bounded by $C_s^- \eta_s^{\text{anc}} - \eta_s \tau \leq -\frac{\eta_s \tau}{2}$ by Property 2 and letting $\frac{\eta_s^{\text{anc}}}{\eta_s} \leq \frac{\tau}{2C_s^-}$. By non-negativity of Bregman divergence, we have

$$\sum_{t=1}^{T} \left( \eta_s \tau \psi_s^\Delta(\pi_{p(s)}^{(t)}(\cdot \mid s)) - \eta_s \tau \psi_s^\Delta(\pi_{p(s)}(\cdot \mid s)) + \eta_s m_s^{(t)} \langle -q^{(t)}(s, \cdot), \pi_{p(s)}^{(t)}(\cdot \mid s) - \pi_{p(s)}(\cdot \mid s) \rangle \right)$$

$$\leq -\frac{\eta_s \tau}{2} \sum_{t=2}^{T} D_{\psi_s^\Delta}(\pi_{p(s)}(\cdot \mid s), \bar{\pi}_{p(s)}^{(t)}(\cdot \mid s)) + C_s' \eta_s \tau \eta_s^{\text{anc}} \sum_{t=1}^{T} \left| \psi_s^\Delta(\pi_{p(s)}^{(t)}(\cdot \mid s)) - \psi_s^\Delta(\bar{\pi}_{p(s)}^{(t+1)}(\cdot \mid s)) \right|$$

$$+ 2 C_s^{\text{diff}} \|q\|_\infty \eta_s^2 M_2 T + m_s^{(1)} D_{\psi_s^\Delta}(\pi_{p(s)}(\cdot \mid s), \bar{\pi}_{p(s)}^{(1)}(\cdot \mid s)).$$

For simplicity, we use $\mu := (x, y)$, $F(\mu^\pi) := (-Ay, A^\top x)$, $\Pi := \Pi_1 \times \Pi_2$, and $\mathcal{S} := \mathcal{S}_1 \times \mathcal{S}_2$. By using Lemma E.3, we have

$$0 \overset{(i)}{\leq} G^{(T),\Pi}(\mu^{(\tau,\gamma),*})$$

$$= \sum_{s \in \mathcal{S}} \mu^{(\tau,\gamma),*}(\sigma(s)) G^{(T)}(h; \pi_{p(s)}^{(\tau,\gamma),*}(\cdot \mid h))$$

$$= \sum_{s \in \mathcal{S}} \mu^{(\tau,\gamma),*}(\sigma(s)) \sum_{t=1}^{T} \left( \tau \psi_s^\Delta(\pi_{p(s)}^{(t)}(\cdot \mid s)) - \tau \psi_s^\Delta(\pi_{p(s)}^{(\tau,\gamma),*}(\cdot \mid h)) \right.$$

$$\left. + m_s^{(t)} \langle -q^{(t)}(s, \cdot), \pi_{p(s)}^{(t)}(\cdot \mid s) - \pi_{p(s)}^{(\tau,\gamma),*}(\cdot \mid h) \rangle \right)$$

$$\leq -\frac{\tau}{2} \sum_{t=2}^{T} \sum_{s \in \mathcal{S}} \mu^{(\tau,\gamma),*}(\sigma(s)) D_{\psi_s^\Delta}(\pi_{p(s)}^{(\tau,\gamma),*}(\cdot \mid h), \bar{\pi}_{p(s)}^{(t)}(\cdot \mid s))$$

$$+ \sum_{s \in \mathcal{S}} C_s' \tau \eta_s^{\text{anc}} \mu^{(\tau,\gamma),*}(\sigma(s)) \sum_{t=1}^{T} \left| \psi_s^\Delta(\pi_{p(s)}^{(t)}(\cdot \mid s)) - \psi_s^\Delta(\bar{\pi}_{p(s)}^{(t+1)}(\cdot \mid s)) \right|$$

$$+ 2 \sum_{s \in \mathcal{S}} C_s^{\text{diff}} \mu^{(\tau,\gamma),*}(\sigma(s)) \|q\|_\infty \eta_s M_2 T$$

$$+ \sum_{s \in \mathcal{S}} \frac{m_s^{(1)}}{\eta_s} \mu^{(\tau,\gamma),*}(\sigma(s)) D_{\psi_s^\Delta}(\pi_{p(s)}(\cdot \mid s), \bar{\pi}_{p(s)}^{(1)}(\cdot \mid s)).$$

$(i)$ is because $\mu^{(\tau,\gamma),*}$ is the NE of the regularized and perturbed EFG. Then, by rearranging the terms, we have

$$\sum_{t=2}^{T} D_{\psi^\Pi}(\mu^{(\tau,\gamma),*}, \bar{\mu}^{(t)})$$

$$= \sum_{t=2}^{T} \sum_{s \in \mathcal{S}} \mu^{(\tau,\gamma),*}(\sigma(s)) D_{\psi_s^\Delta}(\pi_{p(s)}^{(\tau,\gamma),*}(\cdot \mid h), \bar{\pi}_{p(s)}^{(t)}(\cdot \mid s))$$

$$\leq 2 \sum_{s \in \mathcal{S}} C_s' \eta_s^{\text{anc}} \mu^{(\tau,\gamma),*}(\sigma(s)) \sum_{t=1}^{T} \left| \psi_s^\Delta(\pi_{p(s)}^{(t)}(\cdot \mid s)) - \psi_s^\Delta(\bar{\pi}_{p(s)}^{(t+1)}(\cdot \mid s)) \right|$$

$$+ \frac{4}{\tau} \sum_{s \in \mathcal{S}} C_s^{\text{diff}} \mu^{(\tau,\gamma),*}(\sigma(s)) \|q\|_\infty \eta_s M_2 T$$

$$+ \frac{2}{\tau} \sum_{s \in \mathcal{S}} \frac{m_s^{(1)}}{\eta_s} \mu^{(\tau,\gamma),*}(\sigma(s)) D_{\psi_s^\Delta}(\pi_{p(s)}(\cdot \mid s), \bar{\pi}_{p(s)}^{(1)}(\cdot \mid s)).$$

The first line is by Lemma 4.1. Now, we achieved best-iterate convergence to the regularized NE $\mu^{(\tau,\gamma),*}$ in terms of Bregman divergence. $\qquad \square$

## I.1 LAZY QFR WITH STOCHASTIC FEEDBACK

Consider when we can only estimate $q^{(t)}$ at each iteration, we apply the following update-rule to *each individual* infoset $s \in \mathcal{S}$,

$$
\begin{aligned}
\pi_{p(s)}^{(t)}(\cdot \mid s) =& \operatorname*{argmin}_{\pi_{p(s)}(\cdot \mid s) \in \Delta_{|\mathcal{A}_s|}^{\gamma_s, \boldsymbol{\nu}_s}} \left\langle \pi_{p(s)}(\cdot \mid s), -\widetilde{q}^{(t-1)}(s, \cdot) \right\rangle + \frac{\tau}{m_s^{(t-1)}} \psi_s^{\Delta}(\pi_{p(s)}(\cdot \mid s)) \\
&+ \frac{1}{\eta_s} D_{\psi_s^{\Delta}}(\pi_{p(s)}(\cdot \mid s), \bar{\pi}_{p(s)}^{(t)}(\cdot \mid s)) \\
\bar{\pi}_{p(s)}^{(t+1)}(\cdot \mid s) =& \operatorname*{argmin}_{\pi_{p(s)}(\cdot \mid s) \in \Delta_{|\mathcal{A}_s|}^{\gamma_s, \boldsymbol{\nu}_s}} \left\langle \pi_{p(s)}(\cdot \mid s), -\widetilde{q}^{(t)}(s, \cdot) \right\rangle + \frac{\tau}{m_s^{(t)}} \psi_s^{\Delta}(\pi_{p(s)}(\cdot \mid s)) \\
&+ \frac{1}{\eta_s} D_{\psi_s^{\Delta}}(\pi_{p(s)}(\cdot \mid s), \bar{\pi}_{p(s)}^{(t)}(\cdot \mid s)).
\end{aligned}
\tag{I.4}
$$

**Remark I.3.** Note that the update-rule above is not realistic, since even when an infoset is unvisited at timestep $t$, *i.e.* $\widetilde{q}^{(t)} = 0$, we still need to update the strategy $\bar{\pi}_{p(s)}^{(t+1)}(\cdot \mid s)$ and $\pi_{p(s)}^{(t+1)}(\cdot \mid s)$ due to the regularization term. However, we can do a lazy update. We will record the update time $t_h$ of each infoset $s \in \mathcal{S}$. Once we touch a infoset $s \in \mathcal{S}$ at timestep $t$, we will do $t - t_h$ steps update with a zero $q^{(t)}(s, \cdot)$ and update the time stamp $t_h \leftarrow t$.

Then, we have the following theorem. Note that the dependence on game size is still polynomial for Q-value and counterfactual value, since $\frac{\eta_s^{\mathrm{anc}}}{\eta_s}$ is bounded by a constant.

**Theorem I.4.** Consider update-rule (I.4) and $q^{(t)}(s, \cdot)$ is chosen to be counterfactual value, trajectory Q-value, or Q-value. When $\frac{\eta_s^{\mathrm{anc}}}{\eta_s} \leq \frac{\tau}{2C_s}$ for any $s \in \mathcal{S}$ and (**A**), (**B**), (**C**) are satisfied, we have the following guarantee with probability $1 - \delta$.

$$
\begin{aligned}
&\sum_{t=2}^{T} D_{\psi^{\Pi}}(\boldsymbol{\mu}^{(\tau,\gamma),*}, \bar{\boldsymbol{\mu}}^{(t)}) \\
\leq& 2 \sum_{s \in \mathcal{S}} C_s' \eta_s^{\mathrm{anc}} \mu^{(\tau,\gamma),*}(\sigma(s)) \sum_{t=1}^{T} \left| \psi_s^{\Delta}(\pi_{p(s)}^{(t)}(\cdot \mid s)) - \psi_s^{\Delta}(\bar{\pi}_{p(s)}^{(t+1)}(\cdot \mid s)) \right| \\
&+ \frac{4}{\tau} \sum_{s \in \mathcal{S}} C_s^{\mathrm{diff}} \mu^{(\tau,\gamma),*}(\sigma(s)) \|\boldsymbol{q}\|_{\infty} \eta_s M_2 T \\
&+ \frac{2}{\tau} \sum_{s \in \mathcal{S}} \frac{m_s^{(1)}}{\eta_s} \mu^{(\tau,\gamma),*}(\sigma(s)) D_{\psi_s^{\Delta}}(\pi_{p(s)}(\cdot \mid s), \bar{\pi}_{p(s)}^{(1)}(\cdot \mid s)) + \frac{8}{\tau} \|\boldsymbol{q}\|_{\infty} \cdot |\mathcal{S}| \sqrt{2T \log \frac{1}{\delta}}.
\end{aligned}
\tag{I.5}
$$

By Lemma 4.3, once we have an unbiased estimator $\widetilde{q}^{(t)}(s, \cdot)$ for $q^{(t)}(s, \cdot)$ in each infoset $s \in \mathcal{S}$ with the same upper-bound $\|\boldsymbol{q}\|_\infty$, by Lemma 4.3, with probability $1 - \delta$, we have

$$\sum_{t=1}^{T} \left( \tau \psi_s^\Delta(\pi_{p(s)}^{(t)}(\cdot \mid s)) - \tau \psi_s^\Delta(\pi_{p(s)}(\cdot \mid s)) + m_s^{(t)} \langle -q^{(t)}(s, \cdot), \pi_{p(s)}^{(t)}(\cdot \mid s) - \pi_{p(s)}(\cdot \mid s) \rangle \right)$$

$$\leq \sum_{t=1}^{T} \left( \tau \psi_s^\Delta(\pi_{p(s)}^{(t)}(\cdot \mid s)) - \tau \psi_s^\Delta(\pi_{p(s)}(\cdot \mid s)) + m_s^{(t)} \langle -\widetilde{q}^{(t)}(s, \cdot), \pi_{p(s)}^{(t)}(\cdot \mid s) - \pi_{p(s)}(\cdot \mid s) \rangle \right)$$

$$+ 4 \|\boldsymbol{q}\|_\infty \sqrt{2T \log \frac{1}{\delta}}$$

$$\overset{(i)}{\leq} -\frac{\tau}{2} \sum_{t=2}^{T} D_{\psi_s^\Delta}(\pi_{p(s)}(\cdot \mid s), \bar{\pi}_{p(s)}^{(t)}(\cdot \mid s)) + C_s' \tau \eta_s^{\mathrm{anc}} \sum_{t=1}^{T} \left| \psi_s^\Delta(\pi_{p(s)}^{(t)}(\cdot \mid s)) - \psi_s^\Delta(\bar{\pi}_{p(s)}^{(t+1)}(\cdot \mid s)) \right|$$

$$+ 2 C_s^{\mathrm{diff}} \|\boldsymbol{q}\|_\infty \eta_s m_s^{(t)} T$$

$$+ \frac{m_s^{(1)}}{\eta_s} D_{\psi_s^\Delta}(\pi_{p(s)}(\cdot \mid s), \pi_{p(s)}^{(1)}(\cdot \mid s)) + 4 \|\boldsymbol{q}\|_\infty \sqrt{2T \log \frac{1}{\delta}}.$$

$(i)$ follows the discussion in Appendix E.

Then, the proof follows the proof of Theorem 4.2. $\qquad\square$

