# OpenReview forum: "A Policy-Gradient Approach to Solving Imperfect-Information Games with Best-Iterate Convergence"
_ICLR.cc/2025/Conference — ICLR 2025 Poster_

### Official Review · Reviewer_66ub · 2024-11-01

**Soundness:** 3
**Presentation:** 2
**Contribution:** 3
**Rating:** 8
**Confidence:** 3

**Summary:**

The paper introduces a policy gradient method, QFR, for two-player (imperfect-information) zero-sum extensive-form games. The algorithm can approximate best-iterate convergence to a regularized Nash equilibrium. Policy gradient updates are preferred over CFR’s counterfactual updates because they can be computed from rollouts and do not require importance sampling. The paper proves its findings and runs some benchmarks on toy environments.

**Strengths:**

The paper proposes a compelling algorithm and goes to great lengths to prove its convergence properties. Not all competing algorithms in the class have proven convergence properties. Some strengths over CFR:

* Best iterate convergence is more convenient than time-average convergence
* No importance sampling

The paper seems technically sound – I haven’t found a flaw yet – however there are parts that I did not understand (see weakness/questions). I am very open minded to increase the soundness score and overall rating of this paper after discussions with authors and other reviewers.

**Weaknesses:**

Some terms seem to be undefined which compounded confusion on an already complex paper. Please fix the undefined symbols (see questions for concrete examples).

There are many step-jumps in the main text that are not immediately clear to me. Some of these are explained in the appendices, but they are not always well sign-posted. Improved sign-posting may make reading through the paper easier (see questions for concrete examples).

Focuses on two-player zero-sum (a simple game-theoretic setting).

**Questions:**

I am really struggling to understand how Equation 4.3 is derived. Is there somewhere in the appendix that describes it? This confusion snowballed throughout the rest of Section 4.2.

“We can see QFR outperforms BOMD in all games”. Does it in Liar’s Dice?

Minor:

* Line 208: I find the definition of d(h|s) hard to understand. Can another sentence be added, or some more steps shown? Footnote 2 did not seem to enlighten it for me.
* Line 240: The notation of the policies is confusing. And what is pi bar? Has it been defined?
* Line 245: “ad ditional”
* Line 245: What does subscript “bi” mean on the regularizer? Bilinear? (edit: later on line 269, bidilated)
* Line 256: mu should be a psi?
* Line 296: Dodgy line formatting of “else”
* Line 303: Algo line 10. Where did this come from? Add a couple more words explaining the policy gradient update.
* Line 323: “...learning *rate* of inforset…”
* Line 411: +“the”, “and” -> “or”
* Line 417: I am really struggling to see where this inequality comes from.
* Line 469: “... with under …”
* Line 716: “a NE” -> “an NE”
* Line 796: What is alpha_s’? Did it get defined?

---

> ### Author Response · Authors · 2024-11-21
> **Response to Reviewer 66ub**
>
> Thank you very much for your acknowledgment and response, and for making our manuscript better!
>
> ## Two-Player Zero-Sum Game
>
> We focus on two-player zero-sum game because this is one of the most popular and important games. For instance, Go, Chess, StarCraft, Dark Chess, and Stratego all fall into this class.
>
> Moreover, [1][2] showed that computing Nash equilibria in two-player **general-sum** matrix games is PPAD-hard. Therefore, for games beyond two-player zero-sum, computing Nash equilibrium is generally intractable. Moreover, it is still unclear what the suitable solution concept is for superhuman agents in games beyond two-player zero-sum, due to the equilibrium selection issue [11].
>
> ## How Equation 4.3 is derived
>
> Thank you very much for pointing out the typo. We updated it in the latest version.
>
> Equation 4.3 is derived from the three-point identity of Bregman divergence. Its formal proof is in Appendix E (line 1193-1205 are its formal statement, we use the big O notion in the main text for simplicity).
>
> Basically, to prove Equation 4.3, we need to use the three-point identity first (Lemma E.2.). Then, by telescoping and bounding the additional terms given by Lemma E.2. individually, we will get Equation 4.3.
>
> ## Experiments
>
> Thank you very much for pointing this out. In the latest revision, as pointed out by Reviewer ggwY, the algorithms seem not converge in Liar's Dice. Therefore, we run ten times as many iterations in Figure 1 of the latest revision. Now QFR outperforms BOMD in all games but is still inferior to BFTRL in Liar's Dice. The reason may be Liar's Dice is too easy since it can be solved within 50 iterations with full-information feedback (see Figure 2 in Appendix H).
>
> ## $d(h|s)$
>
> $d(h|s)$ is the probability of reaching node $h$ conditioned on reaching infoset $s$. In other words, we can also write it as $d(h|s):=Pr(h|s)=\frac{Pr(h) Pr(s|h)}{Pr(s)}$ by Bayes' Theorem. Moreover, $Pr(s|h)=1$ given that node $h$ is in infoset $s$. Therefore, $d(h|s)$ is proportional to $Pr(h)$, which is $\mu_c(h) \mu_1^{\pi_1}(\sigma_1(h)) \mu_2^{\pi_2}(\sigma_2(h))$.
>
> Let $s(h)$ denote the infoset node $h$ belongs to, then $\mu_1^{\pi_1}(\sigma_1(h))=\prod_{(h'',a'')\sqsubseteq h: h''\in \mathcal{H}_1}\pi_1(a''|s(h''))=\mu_1^{\pi_1}(\sigma_1(h'))$ for any nodes $h,h'\in s$. Because the game is perfect-recall, the infosets of player 1 should be the same along the path from the root to any node $h\in s$. Otherwise, if $h,h'\in s$ has an infoset of player 1 differs along the path, then player 1 can differentiate $h,h'$ because he remembers perfectly how he comes from root to $h$ and $h’$, which conflicts with the definition of infoset.
>
> ## Minor comments
>
> Thank you for flagging the typos and making the new version better. Here are the answers to your questions in the "Minor:" section of your review.
>
> - $\bar \pi_i^{(t)}$ is an auxiliary strategy in the optimistic updates of the algorithm. We define it in Line 14 of Algorithm 1 and explain it in line 274-277.
> - $\alpha_s>0$ is the hyper-parameter in the local regularizer $\psi_s^{\Delta}$ at infoset $s$. The formal definition of $\psi_s^{\Delta}$ can be found in Equation C.1. of the latest revision.
> We add the introduction at the beginning of Appendix C in the latest revision. We include $\alpha_s$ in the regularizer for the generality of the results.

---

> > ### Author Response · Authors · 2024-11-21
> > **References**
> >
> > [1]. Daskalakis, Constantinos, Paul W. Goldberg, and Christos H. Papadimitriou. "The complexity of computing a Nash equilibrium." Communications of the ACM 52.2 (2009): 89-97.
> >
> > [2]. Xi Chen, Shang-Hua Teng, and Paul Valiant. The approximation complexity of win-lose games. In SODA, volume 7, pages 159–168. Citeseer, 2007.
> >
> > [3] Liu, Mingyang, et al. "The power of regularization in solving extensive-form games." arXiv preprint arXiv:2206.09495 (2022).
> >
> > [4] Sokota, Samuel, et al. "A unified approach to reinforcement learning, quantal response equilibria, and two-player zero-sum games." arXiv preprint arXiv:2206.05825 (2022).
> >
> > [5] Perolat, Julien, et al. "Mastering the game of Stratego with model-free multiagent reinforcement learning." Science 378.6623 (2022): 990-996.
> >
> > [6] Lanctot, Marc, et al. "Monte Carlo sampling for regret minimization in extensive games." Advances in neural information processing systems 22 (2009).
> >
> > [7] Bai, Yu, et al. "Near-optimal learning of extensive-form games with imperfect information." International Conference on Machine Learning. PMLR, 2022.
> >
> > [8] Fiegel, Côme, et al. "Adapting to game trees in zero-sum imperfect information games." International Conference on Machine Learning. PMLR, 2023.
> >
> > [9] Brown, Noam, et al. "Deep counterfactual regret minimization." International conference on machine learning. PMLR, 2019.
> >
> > [10] Schulman, John, et al. "Proximal policy optimization algorithms." arXiv preprint arXiv:1707.06347 (2017).
> >
> > [11] Ge, Jiawei, et al. "Towards Principled Superhuman AI for Multiplayer Symmetric Games." arXiv preprint arXiv:2406.04201 (2024).
> >
> > [12] Heinrich, Johannes, and David Silver. "Deep reinforcement learning from self-play in imperfect-information games." arXiv preprint arXiv:1603.01121 (2016).
> >
> > [13] Brown, Noam, et al. "Deep counterfactual regret minimization." International conference on machine learning. PMLR, 2019.
> >
> > [14] Perolat J, De Vylder B, Hennes D, et al. Mastering the game of Stratego with model-free multiagent reinforcement learning[J]. Science, 2022, 378(6623): 990-996.
> >
> > [15] Abe K, Ariu K, Sakamoto M, et al. Adaptively Perturbed Mirror Descent for Learning in Games[C]//Forty-first International Conference on Machine Learning.

---

> > ### Comment · Reviewer_66ub · 2024-11-25
> >
> > Thank you for the clarifications. I have no further questions. I have increased soundness rating.

---

> > > ### Author Response · Authors · 2024-11-25
> > >
> > > We are happy that the clarifications are helpful. Thank you very much for your acknowledgment and for raising the score!

---

### Official Review · Reviewer_XFQu · 2024-11-02

**Soundness:** 3
**Presentation:** 3
**Contribution:** 4
**Rating:** 6
**Confidence:** 3

**Summary:**

The author(s) propose a Q-function based RL approach to solve zero-sum, extensive-form games under imperfect information. The proposed approach only requires sampling randomly generated trajectories and has iterate convergence guarantees.

**Strengths:**

- The contribution of the paper is strong, where the proposed approach only requires sampling of randomly generated trajectories (as opposed to using importance sampling) to estimate value function and compute policy. The approach is proved to have best-iterate convergence guarantees to the Nash equilibria of the regularized game under both full and imperfect information.
- Clear introduction of related works and main obstacles in the field as well as contribution statement. I really enjoyed reading this part.

**Weaknesses:**

1. Line 280: The introduction of the algorithm can be more detailed. Right now, it is only a few lines and assumes the readers are already familiar with the references. For example, the author(s) ca n do a better job walking the readers through their update of the regularizer at Line 8.
2. The exploitability metric used in the experiment section is undefined. The experiment section can be better presented: what makes the proposed approach outperform a certain baseline in a certain setting? Why in certain setting, the proposed approach is inferior? Could the author(s) comment on how many samples are required by their approach vs. the baselines?
3. Completeness of the main text: In my opinion, discussion and conclusion should definitely be in the main text. Right now, the main text is not self-contained. That being said, while I appreciate the completeness of the first two sections, I also feel that they could probably be tightened to make room for other important content without losing much information. If the author(s) finds it impossible to fit the important content all in the main text, this paper might fit better for a journal.
4. Other minor points:
- Introduction (line 35): The statements (I) and (III) overlap with each other; they might be merged into one
- Line 91 “Techniques based on rollouts of …” overlaps the previous paragraph. Author(s) could move this point to above
- line 245: “additional”
- Algorithm 1 line 6: indentation of the if-else statement is broken
- Line 282: “Lastly, all infosets along the trajectory…” - I think the author(s) meant to point to Line 12 rather than Line 11

**Questions:**

1. Eq (2.3): should the direction of the inequality be flipped?
2. Have the author(s) tried to anneal the regularization? Or the results given in the experiments are all subject to regularized objective functions?
3. The proposed approach samples one trajectory using the current policy to compute estimates of value function at each iteration, I feel like this approach might have a high variance for larger games. Did author(s) have any observation and/or explanation regarding this?
4. The method seems to be a value function-based RL approach, as it estimates value functions and computes policies by maximizing the value functions. Why call it “policy gradient” method? Am I missing something?
5. The approach integrates existing approaches, e.g., convex regularizers and mirror descent. Could the author (s) clarify in the text clearly, what are the algorithmic modifications of the proposed approach beyond existing methods that enable the main features? My sense is that weighting of the regularizers considering both players’ paths along the tree is the main stretch, but I would like to know more specifically.

---

> ### Author Response · Authors · 2024-11-21
> **Response to Reviewer XFQu**
>
> ## Expanded introductory description of the algorithm
>
> Thank you very much for the advice. We add more descriptions about the algorithm in the latest revision.
>
> ## Experiments
>
> The exploitability is the left-hand-side of Equation 2.1, and we add the description in the latest revision.
>
> In line 520-528, we have added more discussion about experiment results in the latest revision.
>
> In Figure 1, since we only compare algorithms learning with trajectory rollouts, the sample complexity is the same for all algorithms listed, which is the height of the game tree per iteration.
>
> ## Conclusions and completeness of the body
>
> We added the conclusion in the latest revision.
>
> ## (I) and (III) overlap
>
> (I) and (III) differ from each other since there exist algorithms that can achieve (I) without (III) or achieve (III) without (I).
>
> For instance, previous work such as outcome-sampling CFR [6], Balanced OMD [7], and Balanced FTRL [8], all achieve (I) without (III) since they need importance sampling to get an estimator of the counterfactual values.
>
> Previous work based on external-sampling, such as external-sampling CFR [6] and Deep CFR [9], achieves (III) by sampling approximately the square root of the game tree size at each iteration.
>
> We have briefly discussed it in line 86-96 in the *original* submission.
>
> # Questions
>
> - We have clarified the wording around Equation (2.3).
> - We keep the regularization coefficient $\tau$ unchanged in the experiments. All exploitability is computed in the original game without additional regularization (Equation 2.1). However, it is easy to anneal the regularization as [3].
> - In practice, to control the variance, we need to train another value network to approximate the value function. For instance, Proximal Policy Optimization (PPO) [9], a practical variant of the policy gradient algorithms in RL, also deploys a value network to reduce the variance. In Figure 3 of the latest revision, the implementation of QFR is based on PPO, which utilized value function approximators during training.
> - We call it policy gradient since it resembles some key properties of policy gradient algorithms such as REINFORCE. (i) learning with trajectory rollouts (ii) estimating the Q-value with the reward collected in the rollout.
>
> ## Technical Novelty
>
> The main technical novelty of this paper is in showing that Q-value-based iterations are sound in imperfect-information games. To get there, (i) we extended the dilated regularizer to bidilated regularizer, which can soundly avoid importance sampling while learning from trajectory rollouts.
>
> (ii) We proposed a new learning rate schedule. More specifically, in our algorithm QFR, the learning rate is selected to increase from the top to the bottom of the game tree. The learning rate of each infoset is recursively set to be a constant multiple of its parent nodes.
>
> Intuitively, this enables strategies at the bottom of the game tree, which are inherently easier to learn, to converge more quickly while keeping the parent utilities relatively stable. The fact that bottom strategies are easier to learn can be observed in Texas hold'em. At the end of the game, all public cards are revealed so that it would be easier to decide on the bet than at the beginning of the game.
>
> ## Minor Issues
>
> We fixed all minor issues mentioned in the response. Thank you very much for your suggestions!

---

> > ### Author Response · Authors · 2024-11-21
> > **References**
> >
> > [1]. Daskalakis, Constantinos, Paul W. Goldberg, and Christos H. Papadimitriou. "The complexity of computing a Nash equilibrium." Communications of the ACM 52.2 (2009): 89-97.
> >
> > [2]. Xi Chen, Shang-Hua Teng, and Paul Valiant. The approximation complexity of win-lose games. In SODA, volume 7, pages 159–168. Citeseer, 2007.
> >
> > [3] Liu, Mingyang, et al. "The power of regularization in solving extensive-form games." arXiv preprint arXiv:2206.09495 (2022).
> >
> > [4] Sokota, Samuel, et al. "A unified approach to reinforcement learning, quantal response equilibria, and two-player zero-sum games." arXiv preprint arXiv:2206.05825 (2022).
> >
> > [5] Perolat, Julien, et al. "Mastering the game of Stratego with model-free multiagent reinforcement learning." Science 378.6623 (2022): 990-996.
> >
> > [6] Lanctot, Marc, et al. "Monte Carlo sampling for regret minimization in extensive games." Advances in neural information processing systems 22 (2009).
> >
> > [7] Bai, Yu, et al. "Near-optimal learning of extensive-form games with imperfect information." International Conference on Machine Learning. PMLR, 2022.
> >
> > [8] Fiegel, Côme, et al. "Adapting to game trees in zero-sum imperfect information games." International Conference on Machine Learning. PMLR, 2023.
> >
> > [9] Brown, Noam, et al. "Deep counterfactual regret minimization." International conference on machine learning. PMLR, 2019.
> >
> > [10] Schulman, John, et al. "Proximal policy optimization algorithms." arXiv preprint arXiv:1707.06347 (2017).
> >
> > [11] Ge, Jiawei, et al. "Towards Principled Superhuman AI for Multiplayer Symmetric Games." arXiv preprint arXiv:2406.04201 (2024).
> >
> > [12] Heinrich, Johannes, and David Silver. "Deep reinforcement learning from self-play in imperfect-information games." arXiv preprint arXiv:1603.01121 (2016).
> >
> > [13] Brown, Noam, et al. "Deep counterfactual regret minimization." International conference on machine learning. PMLR, 2019.
> >
> > [14] Perolat J, De Vylder B, Hennes D, et al. Mastering the game of Stratego with model-free multiagent reinforcement learning[J]. Science, 2022, 378(6623): 990-996.
> >
> > [15] Abe K, Ariu K, Sakamoto M, et al. Adaptively Perturbed Mirror Descent for Learning in Games[C]//Forty-first International Conference on Machine Learning.

---

> > ### Comment · Reviewer_XFQu · 2024-11-23
> > **Thank You for the Answers**
> >
> > Thank you for your answers and clarification!

---

> > > ### Author Response · Authors · 2024-11-25
> > >
> > > We are glad to clarify the questions. Thank you very much for your review and acknowledgment of our contribution!

---

### Official Review · Reviewer_msNB · 2024-11-04

**Soundness:** 3
**Presentation:** 3
**Contribution:** 3
**Rating:** 5
**Confidence:** 2

**Summary:**

This paper proposes a policy gradient method, Q-Function based Regret Minimization (QFR) for solving two-player zero-sum extensive-form games, which uses rollouts to estimate values without importance sampling and achieves best-iterate convergence. Experiments on poker and Hex support the proposed method.

**Strengths:**

The idea of using Q-value in regret minimization is reasonable.

The writing is clear and and the paper is easy to follow.

**Weaknesses:**

The proposed algorithm combines optimistic mirror descent updates and estimating Q-values from rollouts, both of which seem to be well known techniques. The technical novelty might be limited.

Motivations seem to be disconnected with later sections, such as experiments.

**Questions:**

The three desirable properties were mentioned at the beginning of the paper and seem to be important. But they seem to be not mentioned in later sections. For example, how can be observe from experiments the advantage of not using importance sampling in QFR? Does it make the proposed method have smaller variance than other methods? And also, how is the using rollout property helpful? Does it make the proposed method more sample efficient than other methods in the experiments?

---

> ### Author Response · Authors · 2024-11-21
> **Response to Reviewer msNB**
>
> Thank you very much for your advice on our paper!
>
> ## Technical Novelty
>
> The main technical novelty of this paper is in showing that Q-value-based iterations are sound in imperfect-information games. To get there, (i) we extended the dilated regularizer to bidilated regularizer, which can soundly avoid importance sampling while learning from trajectory rollouts.
>
> (ii) We proposed a new learning rate schedule. More specifically, in our algorithm QFR, the learning rate is selected to increase from the top to the bottom of the game tree. The learning rate of each infoset is recursively set to be a constant multiple of its parent nodes.
>
> Intuitively, this enables strategies at the bottom of the game tree, which are inherently easier to learn, to converge more quickly while keeping the parent utilities relatively stable. The fact that bottom strategies are easier to learn can be observed in Texas hold'em. At the end of the game, all public cards are revealed so that it would be easier to decide on the bet than at the beginning of the game.
>
> ## Significance of Avoiding Importance Sampling
>
> The most critical issue of importance sampling is that it will cause numerical instability in training targets. Because the denominator is the probability of reaching an infoset, it can be as small as $\frac{1}{\text{game size}}$. Therefore, the feedback (an unbiased estimator of the counterfactual value) fed into the neural network can be as large as the game size. For instance, in Stratego, it can be over $10^{200}$ [5].
>
> To illustrate this, in Figure 3 of the latest revision we present the comparison between importance sampling and no importance sampling when the strategies are approximated by neural networks. We can see from the figure that when using importance sampling, the neural network does not converge. Please also refer to the response to Reviewer ggwY for details.
>
> ## Importance of Trajectory Rollouts
>
> Trajectory rollout is one of the standard components in learning in large games because of its per-iteration sampling efficiency. In each iteration, the number of infosets visited per iteration in trajectory rollouts is bounded by the game tree depth, which is typically the **logarithm of the game tree** size. However, the number of infosets visited by external sampling per iteration is approximately the **square root of the game tree size**.
>
> For instance, in Stratego, trajectory rollouts only need to visit approximately $4\cdot 10^3$ [5] infosets while it is more than $10^{200}$ infosets when using external sampling.
>
> We added some motivation in the latest revision about this on line 85-92.

---

> > ### Author Response · Authors · 2024-11-21
> > **References**
> >
> > [1]. Daskalakis, Constantinos, Paul W. Goldberg, and Christos H. Papadimitriou. "The complexity of computing a Nash equilibrium." Communications of the ACM 52.2 (2009): 89-97.
> >
> > [2]. Xi Chen, Shang-Hua Teng, and Paul Valiant. The approximation complexity of win-lose games. In SODA, volume 7, pages 159–168. Citeseer, 2007.
> >
> > [3] Liu, Mingyang, et al. "The power of regularization in solving extensive-form games." arXiv preprint arXiv:2206.09495 (2022).
> >
> > [4] Sokota, Samuel, et al. "A unified approach to reinforcement learning, quantal response equilibria, and two-player zero-sum games." arXiv preprint arXiv:2206.05825 (2022).
> >
> > [5] Perolat, Julien, et al. "Mastering the game of Stratego with model-free multiagent reinforcement learning." Science 378.6623 (2022): 990-996.
> >
> > [6] Lanctot, Marc, et al. "Monte Carlo sampling for regret minimization in extensive games." Advances in neural information processing systems 22 (2009).
> >
> > [7] Bai, Yu, et al. "Near-optimal learning of extensive-form games with imperfect information." International Conference on Machine Learning. PMLR, 2022.
> >
> > [8] Fiegel, Côme, et al. "Adapting to game trees in zero-sum imperfect information games." International Conference on Machine Learning. PMLR, 2023.
> >
> > [9] Brown, Noam, et al. "Deep counterfactual regret minimization." International conference on machine learning. PMLR, 2019.
> >
> > [10] Schulman, John, et al. "Proximal policy optimization algorithms." arXiv preprint arXiv:1707.06347 (2017).
> >
> > [11] Ge, Jiawei, et al. "Towards Principled Superhuman AI for Multiplayer Symmetric Games." arXiv preprint arXiv:2406.04201 (2024).
> >
> > [12] Heinrich, Johannes, and David Silver. "Deep reinforcement learning from self-play in imperfect-information games." arXiv preprint arXiv:1603.01121 (2016).
> >
> > [13] Brown, Noam, et al. "Deep counterfactual regret minimization." International conference on machine learning. PMLR, 2019.
> >
> > [14] Perolat J, De Vylder B, Hennes D, et al. Mastering the game of Stratego with model-free multiagent reinforcement learning[J]. Science, 2022, 378(6623): 990-996.
> >
> > [15] Abe K, Ariu K, Sakamoto M, et al. Adaptively Perturbed Mirror Descent for Learning in Games[C]//Forty-first International Conference on Machine Learning.

---

### Official Review · Reviewer_ggwY · 2024-11-06

**Soundness:** 3
**Presentation:** 3
**Contribution:** 3
**Rating:** 6
**Confidence:** 3

**Summary:**

The authors propose an algorithm named QFR, which performs policy gradient updates to solve for regularized Nash equilibrium of two-player zero-sum EFGs. Convergence analysis is presented with sublinear rates. Experiment results also show the compatibility of Q-value based policy gradient updates to two-player zero-sum EFGs.

**Strengths:**

The analysis is sound with sublinear iterate average convergence. Proposed idea is novel and easy to follow.

**Weaknesses:**

Analysis section:

- Although it makes sense to enforce strong convexity to the bilinear objective via regularization for easier analysis and stronger convergence guarantee, it also brings a bias to the equilibrium. As the authors are introducing a new regularization as their part of the novelty, it is also expected that the authors show how large this bias is.

Experiment section:

- As the authors mentioned in intro, the motivation behind proposing and proving the convergence of policy gradient based algorithm is to fit better under today's DRL trend. However, the experiment section does not show the advantage of QFR in this regard. As such, I suggest the authors to perform experiment on larger size and compare the solution performance (sample efficiency, computational efficiency etc) across all the different baseline algorithms.

- In figure 1, the convergence seems not reached for Liar's Dice and Leduc Poker. What is the reason behind truncation at the iteration 1000?

**Questions:**

Please see weakness section.

---

> ### Author Response · Authors · 2024-11-21
> **Response to Reviewer ggwY**
>
> Thank you very much for your comments and acknowledgement!
>
> ## Additional Regularization
>
> You are correct: regularization speeds up convergence at the cost of converging to biased solutions. The exploitability (Equation 2.1) of the regularized Nash equilibrium in the original game is $O(\tau)$ (the proof can be found in Lemma D.1. of [3], we have added the link to it in the latest revision on line 330-333), where $\tau$ is the coefficient controlling the scale of regularization.
>
> Additional regularization is popular in modern literature on solving games (e.g., [3] [4] [14] [15]), because intuitively it is often more valuable to moving fast in the right direction than to hit an exact target, especially in large games. To get the best of both worlds, two approaches have been proposed, each of which can be applied as an afterthought. One possibility, employed by R-NaD [14] and APMD [15], is to use "slingshotting": the regularization term is updated periodically to "recenter" around later iterates. Another possibility is to simply anneal the regularization, as done in [3].
>
>
> ## Showing the advantage of QFR
>
> Thank you very much for your comment.
>
> We did an ablation study on importance sampling with the neural network approximated strategy in Leduc Poker, which is a standard benchmark for deep reinforcement learning in games [12][13]. As shown in Figure 3 in the latest revision, the gradient norm blows up when we do importance sampling and thus the network generally learns nothing even though gradient clipping is applied (the right plot shows the gradient **before clipping**). Therefore, with importance sampling, deep reinforcement learning does not work even in a small game, the Leduc Poker, which proves the necessity of property (III).
>
> Additionally, property (I) guarantees efficient sampling (in contrast to external-sampling and full-information, see also the response to Reviewer msNB) and (II) guarantees lower approximation error or storage consumption, which has been extensively discussed in previous work [3][4].
>
> ## On Figure 1
>
> Thank you for your advice! We run experiments in Figure 1 (algorithms with sampling feedback) for ten times as many iterations as the original version and the result is shown in Figure 1 of the new version. We can see that now QFR outperforms outcome-sampling CFR / CFR+, MMD, and BOMD in all games. It is only inferior to BFTRL in Liar's Dice by a small margin (QFR 0.174 v.s. BFTRL 0.167 in exploitability). The reason may be Liar's Dice is too easy since it can be solved within 50 iterations with full-information feedback (see Figure 2 in Appendix H).

---

> > ### Author Response · Authors · 2024-11-21
> > **References**
> >
> > [1]. Daskalakis, Constantinos, Paul W. Goldberg, and Christos H. Papadimitriou. "The complexity of computing a Nash equilibrium." Communications of the ACM 52.2 (2009): 89-97.
> >
> > [2]. Xi Chen, Shang-Hua Teng, and Paul Valiant. The approximation complexity of win-lose games. In SODA, volume 7, pages 159–168. Citeseer, 2007.
> >
> > [3] Liu, Mingyang, et al. "The power of regularization in solving extensive-form games." arXiv preprint arXiv:2206.09495 (2022).
> >
> > [4] Sokota, Samuel, et al. "A unified approach to reinforcement learning, quantal response equilibria, and two-player zero-sum games." arXiv preprint arXiv:2206.05825 (2022).
> >
> > [5] Perolat, Julien, et al. "Mastering the game of Stratego with model-free multiagent reinforcement learning." Science 378.6623 (2022): 990-996.
> >
> > [6] Lanctot, Marc, et al. "Monte Carlo sampling for regret minimization in extensive games." Advances in neural information processing systems 22 (2009).
> >
> > [7] Bai, Yu, et al. "Near-optimal learning of extensive-form games with imperfect information." International Conference on Machine Learning. PMLR, 2022.
> >
> > [8] Fiegel, Côme, et al. "Adapting to game trees in zero-sum imperfect information games." International Conference on Machine Learning. PMLR, 2023.
> >
> > [9] Brown, Noam, et al. "Deep counterfactual regret minimization." International conference on machine learning. PMLR, 2019.
> >
> > [10] Schulman, John, et al. "Proximal policy optimization algorithms." arXiv preprint arXiv:1707.06347 (2017).
> >
> > [11] Ge, Jiawei, et al. "Towards Principled Superhuman AI for Multiplayer Symmetric Games." arXiv preprint arXiv:2406.04201 (2024).
> >
> > [12] Heinrich, Johannes, and David Silver. "Deep reinforcement learning from self-play in imperfect-information games." arXiv preprint arXiv:1603.01121 (2016).
> >
> > [13] Brown, Noam, et al. "Deep counterfactual regret minimization." International conference on machine learning. PMLR, 2019.
> >
> > [14] Perolat J, De Vylder B, Hennes D, et al. Mastering the game of Stratego with model-free multiagent reinforcement learning[J]. Science, 2022, 378(6623): 990-996.
> >
> > [15] Abe K, Ariu K, Sakamoto M, et al. Adaptively Perturbed Mirror Descent for Learning in Games[C]//Forty-first International Conference on Machine Learning.

---

### Meta-Review · Area_Chair_MEir · 2024-12-22

**Metareview:**

Most reviewers agree that this work makes a valuable contribution to policy gradient methods in 2p0s IIEFG with best-iterate convergence. However, concerns were raised regarding the quality of the presentation, which I believe can be addressed in a revised version. I strongly encourage the authors to enhance the presentation to make the work more accessible and coherent.

**Additional Comments On Reviewer Discussion:**

The reviewers raised questions regarding the novelty of the work, certain technical details and experiments, which were thoroughly addressed in the rebuttal. With most reviewers providing positive scores, I recommend acceptance.

---

### Decision · Program_Chairs · 2025-01-22

Accept (Poster)